# The apicoplast link to fever-survival and artemisinin-resistance in the malaria parasite

Min Zhang [1,6], Chengqi Wang[1,6], Jenna Oberstaller [1,6], Phaedra Thomas [1], Thomas D. Otto[2,3], Debora Casandra[1], Sandhya Boyapalle[1], Swamy R. Adapa[1], Shulin Xu[1], Katrina Button-Simons [4], Matthew Mayho[2], Julian C. Rayner [2,5], Michael T. Ferdig[5], Rays H. Y. Jiang[1] & John H. Adams [1✉]

The emergence and spread of *Plasmodium falciparum* parasites resistant to front-line anti-malarial artemisinin-combination therapies (ACT) threatens to erase the considerable gains against the disease of the last decade. Here, we develop a large-scale phenotypic screening pipeline and use it to carry out a large-scale forward-genetic phenotype screen in *P. falciparum* to identify genes allowing parasites to survive febrile temperatures. Screening identifies more than 200 *P. falciparum* mutants with differential responses to increased temperature. These mutants are more likely to be sensitive to artemisinin derivatives as well as to heightened oxidative stress. Major processes critical for *P. falciparum* tolerance to febrile temperatures and artemisinin include highly essential, conserved pathways associated with protein-folding, heat shock and proteasome-mediated degradation, and unexpectedly, isoprenoid biosynthesis, which originated from the ancestral genome of the parasite's algal endosymbiont-derived plastid, the apicoplast. Apicoplast-targeted genes in general are upregulated in response to heat shock, as are other *Plasmodium* genes with orthologs in plant and algal genomes. *Plasmodium falciparum* parasites appear to exploit their innate febrile-response mechanisms to mediate resistance to artemisinin. Both responses depend on endosymbiont-derived genes in the parasite's genome, suggesting a link to the evolutionary origins of *Plasmodium* parasites in free-living ancestors.

[1] Center for Global Health and Infectious Diseases Research and USF Genomics Program, College of Public Health, University of South Florida, Tampa, FL, USA. [2] Wellcome Sanger Institute, Wellcome Genome Campus, Hinxton, Cambridgeshire, UK. [3] Institute of Infection, Immunity and Inflammation, MVLS, University of Glasgow, Glasgow, UK. [4] Eck Institute for Global Health, Department of Biological Sciences, University of Notre Dame, Notre Dame, IN, USA. [5] Cambridge Institute for Medical Research, University of Cambridge, Cambridge Biomedical Campus, Cambridge, Cambridgeshire, UK. [6]These authors contributed equally: Min Zhang, Chengqi Wang, Jenna Oberstaller. ✉email: ja2@usf.edu

Malaria remains a leading infectious disease causing >200 million clinical cases and a half-million deaths every year. *Plasmodium falciparum* is the deadliest malaria parasite by far, with growing parasite resistance to front-line antimalarial artemisinin-combination therapies (ACT) threatening to erase the considerable gains against the disease of the last decade. Alarmingly, data indicate that for the first time since 2010, progress in reducing the global burden of malaria cases and fatalities nearly flatlined between 2015 and 2017[1]. New therapies, ideally informed by an understanding of basic parasite biology, are needed to confront these urgent threats to global malaria control. The study of malaria-parasite biology and gene-function has traditionally been limited, because targeted gene-by-gene approaches are laborious and fraught with difficulty due to an AT-rich (~82%) genome that limits the scalability of specific targeted gene-editing methods (such as CRISPR). Despite the considerable knowledge gene-by-gene studies have enabled, and the approximately two decades that have passed since the *P. falciparum* genome was completed[2], the limited throughput of targeted gene-editing strategies combined with the evolutionary distance of *P. falciparum* from classical model eukaryotes has limited the utility of this powerful tool for broad exploration of *P. falciparum* gene-function, leaving >90% of genes untouched experimentally and ~35% of the parasite's ~5474 genes without meaningful functional annotation (www.plasmodb.org)[3]. High-throughput methods for functionally profiling the malaria-parasite genome can hasten the development of effective interventions to control a parasite proven to be an adaptable foe.

Parasite-specific processes essential for parasite survival are naturally attractive as potential drug targets, given the decreased likelihood of deleterious off-target effects to the host. One such process ripe for interrogation is the parasite's survival response to the extreme conditions of the host's malarial fever. Repeating fever is a hallmark of all types of malaria and the cyclical patterns serve as key diagnostic features of infections. In malignant tertian malaria caused by *P. falciparum*, the 48-h cycle corresponds to the parasite's asexual intraerythrocytic-stage life-cycle, wherein parasites invade, develop, asexually replicate and then rupture their host red blood cell (RBC) to begin the destructive blood-stage cycle anew. Host fever is triggered by a Type I shock-like response of the innate immune-system exposure to extracellular parasite debris released when infected RBCs are lysed during parasite egress[4,5]. Malarial fever concomitantly attenuates and synchronizes the development of blood-stage *P. falciparum* infections, as it is lethal to all parasite stages except for early intraerythrocytic ring stages[6–8]. However, parasite tolerance of febrile temperatures is crucial for its successful propagation in human populations as well as a fundamental aspect of malaria pathogenesis. Previous research suggests parasite-specific factors play a role in modulating this tolerance for febrile temperatures, though the identities of many of these factors or the mechanisms by which they operate remain uncertain[5,9]. In the current study, the in vitro experimental conditions of our experimental method, which was adapted from a previous gene expression study and independently validated[9,10], represent the likely extreme limits of the malarial fever during actual infections.

We previously used random *piggyBac*-transposon insertional mutagenesis to uncover genes essential for *P. falciparum* blood-stage survival, generating a saturation-level *P. falciparum* mutant library containing ~38,000 single-disruption mutants[11]. We defined 2680 genes as essential for asexual blood-stage growth, including ~1000 *Plasmodium*-conserved genes of unknown function. Here we demonstrate the potential of this *piggyBac*-mutant (*pB*-mutant) library to systematically assign functional annotation to the *P. falciparum* genome by genome-wide phenotypic screens.

In this study, we present a large-scale forward-genetic functional screen in *P. falciparum* to identify factors linked to parasite survival of febrile temperatures. Importantly, we functionally annotate hundreds of parasite genes as critical for the parasite's response to heat shock (HS) but dispensable under ideal growth conditions, ~26% of which were previously unannotated with no known function. Expression-profiling the HS-responses in two different heat shock-sensitive (HS-Sensitive) *pB*-mutant clones vs. the wild-type parent NF54 via RNAseq reveals concordance between (1) genes regulated in the parasite's innate response to HS, (2) the processes dysregulated in these mutants vs. wild-type responses to HS, and (3) those mutants we identify as HS-Sensitive in our pooled screens. Together these analyses identify genes and pathways essential in the HS-response, implicating oxidative stress and protein-damage responses, host-cell remodeling, and unexpectedly, apicoplast isoprenoid biosynthesis. Apicoplast-targeted genes, in general, are upregulated in response to HS, as are other *Plasmodium* genes with orthologs in plant and algal genomes. Finally, parallel phenotyping of a mutant library reveals a significant overlap between parasite pathways underlying the response to febrile temperatures and those implicated in the artemisinin mechanism of action (MOA), including oxidative stress, protein-damage responses, and apicoplast-mediated vesicular trafficking[12,13]. Mutants in known protein targets of artemisinin tend to be sensitive to HS[14], and expression data from recent field isolates directly correlates artemisinin resistance with HS tolerance in our pooled screen[15]. Further, we find the key K13-associated parasite endocytosis pathway linked to artemisinin resistance[16,17] is also downregulated in response to HS. Together these data identify an unexpected link between artemisinin MOA, HS-survival, and algal origins of the apicoplast, suggesting the parasite may exploit its innate fever-response mechanisms to gain resistance to artemisinin. This study creates a blueprint for developing a large-scale phenotypic screening pipeline of the *P. falciparum pB*-mutant library to enable high-throughput interrogation of phenotypes of interest to hasten further biological insight that can be weaponized against the parasite.

## Results

**Pooled screens of an extensively characterized pB-mutant clone-library allow robust identification of heat-shock phenotypes.** To interrogate pathways and processes associated with parasite survival at febrile temperatures, we developed a large-scale phenotypic screening pipeline to analyze the phenotypes in pooled *pB*-mutant parasites exposed to HS-induced stress (Supplementary Fig. 1). We previously demonstrated using individual clonal *pB*-mutant parasite lines that mutant growth-phenotypes can be detected and differentiated in pooled screening utilizing QIseq—"Sensitive" mutants with disruptions in genes/genomic features important for growth have lower QIseq-reads, while "Neutral" disruptions in features not vital for growth under the same conditions have higher reads[18]. We, therefore, reasoned that mutants with mutations in genes underlying the HS-response would grow poorly in response to HS compared to mutants in genes not contributing to HS survival.

We used a pool of 128 unique, extensively characterized *P. falciparum pB*-mutant clones reflecting disruptions in genes spanning a range of functional categories, as well as many genes without existing functional information, as a "pilot-library" for initial phenotypic screen-development ([18,19]; "Methods, Generating the pilot-library of pB-mutant parasite clones" section). An in vitro HS-screen of this pilot library, adapted from a phenotype-screen of many *pB*-mutant-clones comprising the pilot-library[10], defined *pB*-mutant HS-response phenotypes to fever-like

temperatures (Fig. 1a–c, Supplementary Data File 1, and "Methods" section). We next calculated a measure of fitness for each mutant in response to HS while also taking into account inherent differences in mutant-growth in ideal conditions, which we termed the Phenotypic-Fitness Score in response to HS (PFS$_{HS}$; "Methods" section). The PFS$_{HS}$ result was consistent with a previously reported flow cytometry-based assay of 25 individual piggyBac-mutant clones in response to heat shock[10] (Wilcoxon $p$ < 0.01, Fig. 1d). We classified 28 mutants of the pilot library as HS-Sensitive (Fig. 1c, d, indicated in red; Supplementary Data File 1). Fourteen mutants performed poorly in both the Growth- and HS-Screens (Fig. 1c, yellow). We classified 28 mutants displaying a slight growth advantage in response to HS (Fig. 1c, d, green) as HS-Tolerant. Mutants exhibiting neither sensitivity nor tolerance to HS were classified as HS-Neutral ($n$ = 49).

QIseq-data resulting from the HS- and Growth-screens allowed robust assignment of mutant-phenotypes for both (see "Methods"

section). We primarily classified mutants sensitive to heat shock alone as HS-Sensitive to avoid possible over-interpretation of generally sick Growth-Sensitive mutants (Fig. 1c and "Methods" section).

**Pooled phenotypic screens scaled up to a 1K $pB$-mutant library enable identification of processes underlying the $P.$ $falciparum$ heat-shock response.** We next scaled our pooled HS-screen to a mutant library of 922 functionally uncharacterized mutants using the methods we established in our pilot-library screens (Supplementary Data File 2). This 1K-library comprised 12 large mixed-population pools of uncloned mutants randomly selected from our saturation library and subjected to phenotypic screens in parallel. Insertion-sites were unknown until the 1K-library HS-Screen and QIseq were completed. Mutants were ranked by fold-change growth in response to HS from HS-Sensitive to HS-Tolerant, as per cut-offs determined from our pilot-library screens.

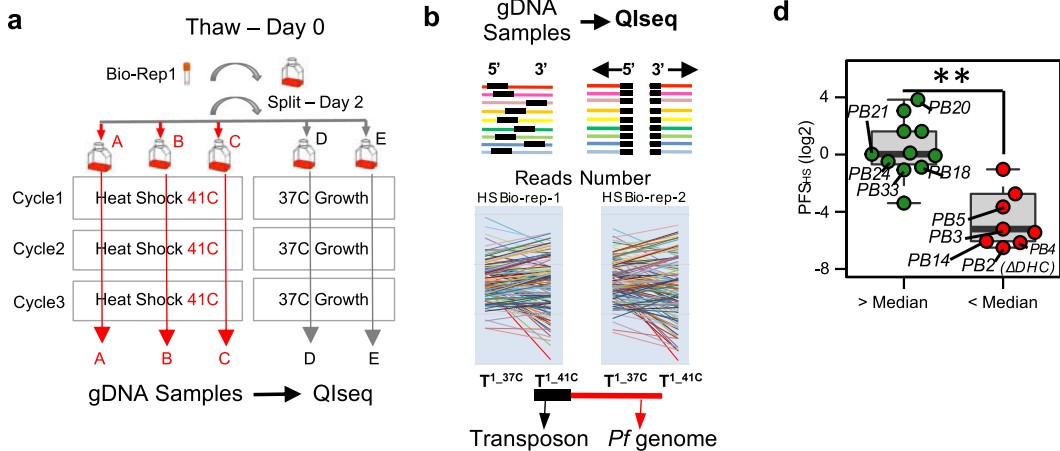

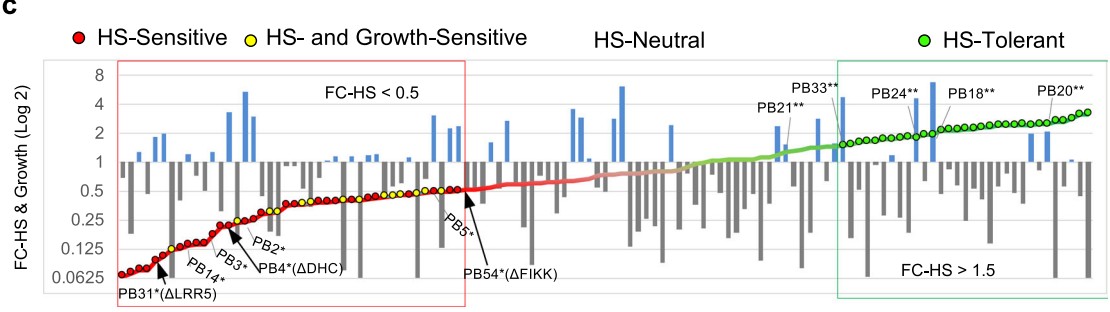

**Fig. 1 Pooled screens of $P.$ $falciparum$ piggyBac mutants allow robust identification of heat-shock phenotypes. a** Experimental design for pooled heat shock (HS) phenotypic screens. The pilot library of $pB$-mutant clones ($n$ = 128) was exposed to three rounds of temperature-cycling (41 °C for 8 h) to simulate malarial fever. A pilot-library control concurrently grown continuously at 37 °C established the inherent growth of each $pB$ mutant. HS screens of the pilot library were conducted in biological duplicate and technical triplicate and were highly correlated, indicating high accuracy and reproducibility (see Supplementary Fig. 5a; "Methods, Pooled-screen assay-design, HS-Screen" section). **b** QIseq quantifies each $pB$-mutant in the pilot library from sequence-reads of the 5′ and 3′ ends of each $pB$ insertion-site. Colored lines represent genes. Black boxes indicate transposon location (Supplementary Fig. 5b; "Methods, QIseq" section). **c** HS- and Growth-phenotypes of the pilot-library mutants. HS-phenotype of each mutant (displayed as line-graph) is superimposed on its corresponding Growth-phenotype (bar graph). Mutants are ordered by fold-change in response to HS (FC-HS) from Sensitive (red) to Tolerant (green). Mutants with inherently slower or faster growth under ideal conditions (FC-Growth) are shown in gray and blue, respectively. *Known HS-Sensitive and **HS-Tolerant pB-mutant clones served as benchmarks in the pilot-library HS-Screen for identifying sensitive/tolerant mutants[10]. See Supplementary Data File 1. **d** Phenotype comparison between mutants characterized in both individual HS-assays[10] and pooled HS-screening ($n$ = 20). Mutant clones without an observed phenotype in individual HS-assay as determined by above-average growth via flow cytometry (green) also had significantly higher Phenotypic Fitness Scores in response to HS (PFS$_{HS}$) in pooled screening, while mutant clones characterized as HS-Sensitive in individual assays (red) also had significantly lower PFS$_{HS}$ in pooled screening (**$p$-value < 0.01, two-tailed Wilcoxon test). Boxplots are drawn to present an interquartile range of values (IQR). Lower bound of each box = 1st quartile, middle line = median, upper bound = 3rd quartile, and whiskers extend to at most 1.5× IQR.

Our analysis distinguished 149 mutants growing well in ideal growth conditions but poorly in response to HS as HS-Sensitive (Fig. 2a), while 91 mutants performed poorly in both the Growth- and HS-screens. Of the remaining mutants, 139 HS-Tolerant mutants had slightly better growth in HS than ideal growth-conditions, while 543 classified as HS-Neutral were neither sensitive nor tolerant.

This larger scale of screening covering genes annotated to diverse GO-categories, as well as many genes of unknown function, allowed us to assess gene functional-enrichment in HS-Sensitive and Growth-Sensitive phenotypic categories vs all other mutants in the 1K-library. HS-Sensitive mutants were enriched in GO terms associated with HS-response such as protein folding, response to DNA-damage, DNA-repair, and regulation of vesicle-

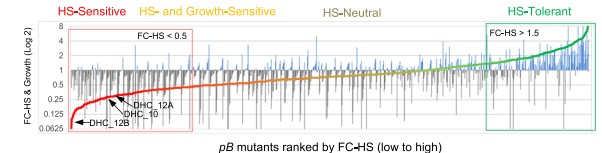

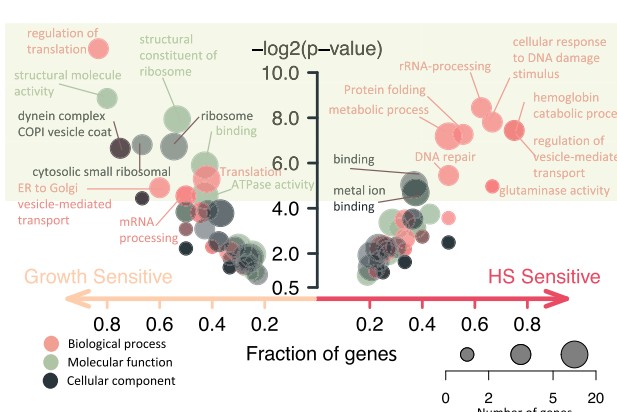

**Fig. 2 Large-scale pooled phenotypic screens enable identification of processes underlying the *P. falciparum* heat-shock response. a** HS-Sensitive mutations identified in pooled screens of the 1K-library of *p*B-mutants (Supplementary Data File 2). The uncloned, large mixed-population pools comprising the 1K-library ($n = 12$) were parallelly screened in both ideal growth conditions and under HS. Mutants were assigned phenotypes as per methods established in the pilot-library screens (Fig. 1 and Supplementary Data File 2). Mutants are ranked by fold-change from HS-Sensitive (red; $n = 149$, FC-HS < 0.5 and PFS$_{HS}$ < 0.25) to HS-Tolerant (green; $n = 139$, FC-HS > 1.5). Mean mutant fold-change in ideal growth (FC-Growth) is superimposed as a bar plot (gray, FC-Growth <1.0; blue, FC-Growth > 1.0). Mutants performing poorly in both screens (yellow; $n = 91$, FC-HS < 0.5, PFS$_{HS}$ > 0.25) were classified as HS- and Growth-Sensitive and were not considered further (Supplementary Fig. 9). Mutations neither HS-Sensitive nor HS-Tolerant were classified as HS-Neutral (taupe, $n = 543$). Internal redundancy within the 1K- and pilot clone libraries allowed assessment of correlation between identical or same-gene mutants across pools. Phenotypes were highly reproducible. See Supplementary Figs. 6a–c, 7a–b, 8a, b. **b** Functional enrichment of GO terms for HS-Sensitive or Growth-Sensitive *p*B-mutants vs all other mutants in the 1K-library. HS-Sensitive mutants were enriched in terms associated with HS-response such as protein folding, response to DNA-damage, DNA-repair, and regulation of vesicle-mediated transport. Growth-Sensitive mutants tended to be enriched for more general categories broadly important for survival in all conditions, such as translation- or mRNA-metabolism-related terms. Circles represent the GO category, circle color represents ontology, and circle size represents the number of significant genes annotated to that category. Significant terms (two-tailed Fisher/elim-hybrid test p-value ≤ 0.05) fall within the light-green box.

mediated transport, broadly in agreement with processes identified to underlie the HS-response by more conventional gene expression-based methods[5,9]. Growth-Sensitive mutants tended to be enriched for more general categories broadly important for survival in all conditions, such as translation- or mRNA-metabolism-related terms (Fig. 2b), as might be expected given the high essentiality of these processes in ideal growth[11,20].

**Increased transcription of the unfolded-protein response (UPR), organelle-targeted stress-response pathways, and host-cell remodeling characterize the parasite HS-response.** We first characterized the wild-type parent-NF54 transcriptome in response to HS to establish a baseline for comparison using an experimental design similar to a prior study assessing transcriptional changes in response to febrile temperatures via microarray[9]. The HS assay design-mimicking parasite exposure to malarial fever was modeled after conditions we established for our pooled screens (see "Methods" section). RNAseq was performed on heat-shocked parasites vs. a non-heat-shocked control. Genes identified as differentially expressed in response to febrile temperatures vs. 37 °C were classified into three different categories based on the direction of response in the wild-type parasite: (1) increased abundance in response to HS (upregulated); (2) decreased abundance in HS (downregulated), and (3) neutral in HS (Fig. 3a, b and Supplementary Data File 3). The majority of genes expressed above threshold in our analysis were HS-neutral (1541 genes out of 2567, or ~60%) and were enriched for genes involved in general housekeeping functions such as the proteasome core complex (ubiquitin-proteasome system), (the ubiquitin-dependent ERAD-pathway, and regulators thereof), RNA metabolism (RNA-binding, mRNA-splicing) and transport functions (e.g., protein import into nucleus, vesicle-mediated transport).

Genes upregulated in HS (↑, $n = 415$) were enriched for processes such as protein folding, unfolded-protein-binding, response to heat, mitochondrial processes, and host-cell remodeling-associated exported proteins localizing to the Maurer's clefts (Fig. 3b and Supplementary Data File 3). Genes downregulated in HS (↓, $n = 611$) were enriched for pathogenesis-related functions and components of the parasite invasion machinery, such as entry/exit from the host cell and cell-cell adhesion, and organelles including the inner-membrane pellicle complex, micronemes, and rhoptries. These data are in general agreement with previously reported processes expected to drive the parasite HS-response[5,9].

We reasoned that genes dysregulated in HS-Sensitive mutants compared to wild-type underlie the HS-response. We chose two individual HS-Sensitive mutant clonal lines satisfying several careful criteria for additional profiling via RNAseq to identify dysregulated genes responsible for this sensitivity: ΔDHC and ΔLRR5 (dynein heavy-chain gene PF3D7_1122900 and leucine-rich repeat protein PF3D7_1432400). Criteria for selection were: (i) Specificity of phenotype. Both mutants are highly sensitive to heat shock (PFS < 0.1), but under ideal culture, conditions grow better than most other mutants in the pilot library (exhibiting higher fold-change than 95.3% and 83.6% mutants, respectively). (ii) Clear functional consequences of disruption. Each mutant has a single disruption in the coding region of a gene determined to be dispensable for asexual blood-stage growth under ideal culture conditions[11]. (iii) GO classification. GO classifications of LRR5 and DHC are representative of the broad functional categories we found to be associated with heat response in our earlier small screen[10] and other reports (regulating gene expression and intracellular vesicular transport, respectively), yet interactions between these pathways are undefined. Finally, (iv) Clonal phenotype validation. Both mutant lines were validated in a

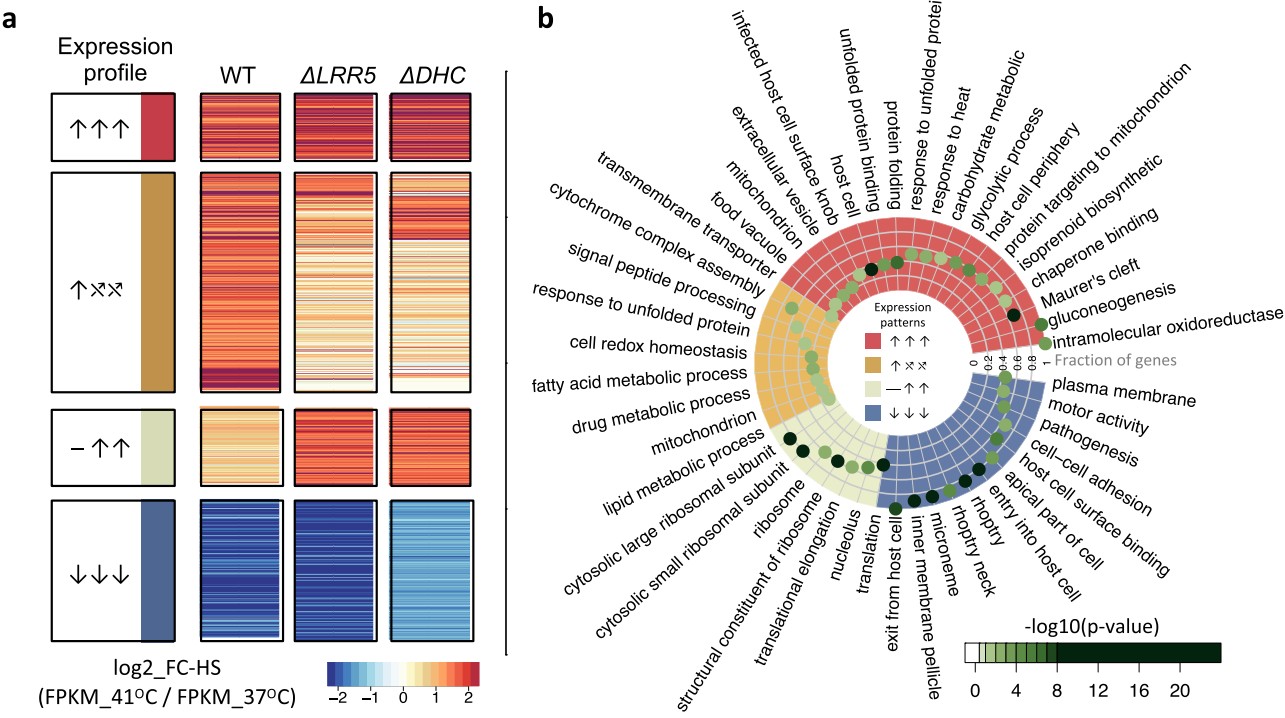

**Fig. 3 Unfolded-protein response, apicoplast-targeted, and mitochondria-targeted stress-response pathways are critically dysregulated in functionally unrelated HS-Sensitive mutant clones. a** Genes were classified into three different categories based on NF54-expression with and without HS-exposure across all three parasite lines (Supplementary Data File 3, "Methods" section): upregulated in HS (↑, $n = 415$), downregulated in HS (↓, $n = 611$), or not regulated by HS (–, $n = 1541$). Genes expressed above threshold in NF54 and both HS-Sensitive mutants ($n = 1298$) were then assigned into six HS expression categories based on phenotype in NF54 vs. mutants Δ*LRR5* and Δ*DHC*. HS-regulated genes shared between NF54 and both mutants are indicated in red (↑↑↑, $n = 94$) or blue (↓↓↓, $n = 205$) for upregulated and downregulated genes, respectively. Genes dysregulated in one or both HS-Sensitive mutants fell into two main expression-profile categories: those upregulated in NF54 that failed to be regulated in the mutants (↑⤬⤬, $n = 83$), and genes not regulated in response to HS in NF54 that were inappropriately upregulated in the mutants (– ↑↑, $n = 74$). Most remaining genes were not regulated in response to HS in any parasite line ($n = 615$). **b** Functional enrichment analyses between wild-type/mutant HS-expression profiles. Red: Shared upregulated HS-responsive GO-terms between NF54 and the two HS-Sensitive *pB*-mutants (↑↑↑). Blue: Shared downregulated GO-terms (↓↓↓). Ocher: GO-terms upregulated in NF54 but dysregulated in both mutants (↑⤬⤬). Tan: GO-terms enriched in genes not regulated in the wild-type HS-response but upregulated in the mutants (– ↑↑). Only enriched GO-terms are shown (two-tailed Fisher/elim-hybrid test *p*-value ≤ 0.05), with the highest significance indicated in dark green. Fraction of significant genes mapping to a GO-term in an HS expression-profile category vs. genes mapping to that GO-term in the entire analysis is indicated by distance to the center of the circle. Source data are provided as a Source Data file. **c** Several apicoplasts and isoprenoid biosynthesis-related genes were upregulated in the wild-type response to HS and were dysregulated in one or both HS-Sensitive *pB*-mutant clones (↑⤬⤬). *Isoprenoid biosynthesis-related genes upregulated by HS confirmed in the pooled HS-Screen.

heat-shock assay of individual clones[10], but otherwise, these genes were not previously implicated in the HS-response of malaria parasites.

The 1298 genes which could be classified into HS-response categories across all three parasites were analyzed for functional enrichment (Supplementary Data File 3). The majority of genes were HS-neutral across all three parasites and were enriched for essential housekeeping functions ($n = 615$; Supplementary Data File 3). We reasoned these non-HS-regulated genes have functions too important for basic survival to tolerate large stress-associated expression changes, and these genes were not considered drivers of the HS-response. We identified 94 genes significantly upregulated in HS across all three parasites (↑↑↑), which were functionally enriched for protein folding, chaperone-related processes, and other processes related to heat-stress and the UPR, in agreement with previous expression-based studies[9], as well as enrichment results from HS-Sensitive mutants in our pooled screening, indicating the parasite increases the production of heat-shock proteins (HSPs) and associated chaperones to repair the glut of proteins damaged/misfolded by heat-stress (Supplementary Data File 3). Energy-producing processes (gluconeogenesis, glycolysis) were also upregulated, suggesting the parasite reroutes anabolic metabolism to increase energy production to support ATP-dependent processes such as protein-refolding to correct heat-damaged proteins. Genes upregulated in HS were further enriched for processes involved in host-cell remodeling, including genes targeted to the Maurer's clefts, the host cell, and intracellular vesicles—all known to be important for parasite-remodeling of the host cell to promote structural reinforcement against heat-shock damage to ensure its own survival[5,9]. Organellar targeting to the mitochondria and apicoplast are also enriched in upregulated HS-responsive genes. The parasite's increased utilization of mitochondrial stress-response pathways may aid in degrading heat-damaged proteins that cannot be correctly refolded. Increased activity in the digestive vacuole may allow the parasite to phagocytose and eliminate toxic misfolded protein aggregates. The apicoplast involvement, particularly the isoprenoid biosynthesis pathway, has not been previously implicated in the HS-response.

Genes downregulated in all three parasites in response to HS (↓↓↓, $n = 205$) were enriched for virulence-factor and invasion machinery-associated GO terms, suggesting the parasite decreases the production of transcripts associated with pathogenesis, invasion, and egress, lengthening its intracellular recovery-time to address global protein damage.

Both HS-Sensitive mutants share many characteristic features of the wild-type response to febrile temperatures, which likely enabled their survival (Fig. 3a, b, red, blue; Supplementary Data File 3). We identified two primary expression categories of genes dysregulated in the HS-Senstive mutants: (1) genes upregulated in the wild-type HS-response that were otherwise dysregulated in the HS-Sensitive mutants, which we interpreted as loss-of-function changes (↑✗✗, $n = 83$), and (2) genes that were not regulated in response to HS in the wild-type but were upregulated in the HS-Sensitive mutants (- ↑↑, $n = 74$), presumably equivalent to dominant-negative gain-of-function changes (Fig. 3a, b, ocher and tan, respectively). This first category of mutant-dysregulated genes (↑✗✗) was enriched for the UPR, as well as mitochondrial and apicoplast-localized pathways (cytochrome oxidase-assembly and fatty-acid biosynthesis, respectively). Several apicoplast isoprenoid biosynthesis-related genes upregulated in the wild-type HS-response were additionally dysregulated in one or both HS-Sensitive pB-mutant clones (Fig. 3c). The second category of mutant-dysregulated genes (- ↑↑), those that are not HS-responsive in wild-type, were enriched for translation-associated processes.

These data taken together suggest underlying mechanisms responsible for the HS-response. Critically, HS-Sensitive mutants fail to upregulate mitochondrial and apicoplast stress-response pathways, as well as signal peptide-processing pathways that might enable appropriate activation of those pathways. Mutants do not increase the production of transcripts associated with responding to unfolded proteins. HS-Sensitive mutants additionally upregulate translation-related processes in response to HS when translation should be paused or neutral. This increase may overwhelm the parasite's capacity to repair or degrade heat-damaged proteins, exacerbating the formation/accumulation of toxic misfolded-protein aggregates that increase parasite sensitivity to HS.

**Apicoplast isoprenoid biosynthesis is critical for *P. falciparum* survival of febrile temperatures**. We examined our RNAseq data more closely to discern contributions of the apicoplast to HS-survival. We found that apicoplast-targeted genes tended to be increased in response to HS as compared to all non-apicoplast-targeted genes (Fig. 4a), were more likely to be essential during ideal blood-stage growth conditions (Fig. 4b), and were enriched for stress-response processes such as the UPR and oxidative stress, and less expectedly, isoprenoid biosynthesis (Fig. 4c). As a major function of isoprenoid biosynthesis is in protein-prenylation—an important post-translational modification that regulates protein-targeting and function throughout the cell—we hypothesized that mutants in known-prenylated proteins[21,22] would also have a phenotype in HS. We examined our 1K mutant-library for representation of isoprenoid biosynthesis, its immediate upstream-regulators (proteins responsible for modulation and import of glycolytic intermediates that serve as pathway substrates), and immediate downstream-effector proteins, and found that 10 of the 11 isoprenoid biosynthesis-related pB-mutants included in the pooled screen were also HS-Sensitive (10 mutants in 9 unique genes, Fisher test p-value < 0.01; Fig. 5a).

Based on these data we further hypothesized that proteins or pathways allowing *P. falciparum* survival of febrile temperatures would be absent or otherwise divergent in *Plasmodium* species whose hosts do not mount fever responses. We, therefore, compared the apicoplast isoprenoid biosynthesis pathway between *P. falciparum* and two rodent-infective species, *P. berghei* and *P. yoelii*. We found key thiamine-synthesis enzymes directly upstream of the pathway missing in the rodent-infective malaria parasites, including hydroxyethylthiazole kinase (ThzK); ThzK is upregulated in the canonical parasite response to febrile temperatures and dysregulated in HS-Sensitive mutants (Fig. 5b). Perhaps most importantly, DOXP-Synthase (DXS), the critical enzyme marking the first step in isoprenoid biosynthesis, is upregulated in HS, dysregulated in HS-Sensitive mutants, and was HS-Sensitive in pooled screening, as were all five members of the prenylated blood-stage proteome represented in our screen (Fig. 5b). These data taken together strongly implicate isoprenoid biosynthesis in the HS-response.

Though the apicoplast has not previously been implicated in parasite survival of febrile temperatures, there is extensive literature on the ability of plants to mount effective defenses against heat as well as other external stressors, particularly critical for non-motile organisms at the mercy of their environments. We investigated the relationship between the parasite's HS-response and plant-like stress responses by evaluating the phyletic distribution of parasite HS-response genes in representative plant and algal genomes. *P. falciparum* genes with plant orthologs indicating potential endosymbiont-ancestry tended to be increased in response to HS vs. genes that do not have plant orthologs (Fig. 5c). These lines of evidence considered together

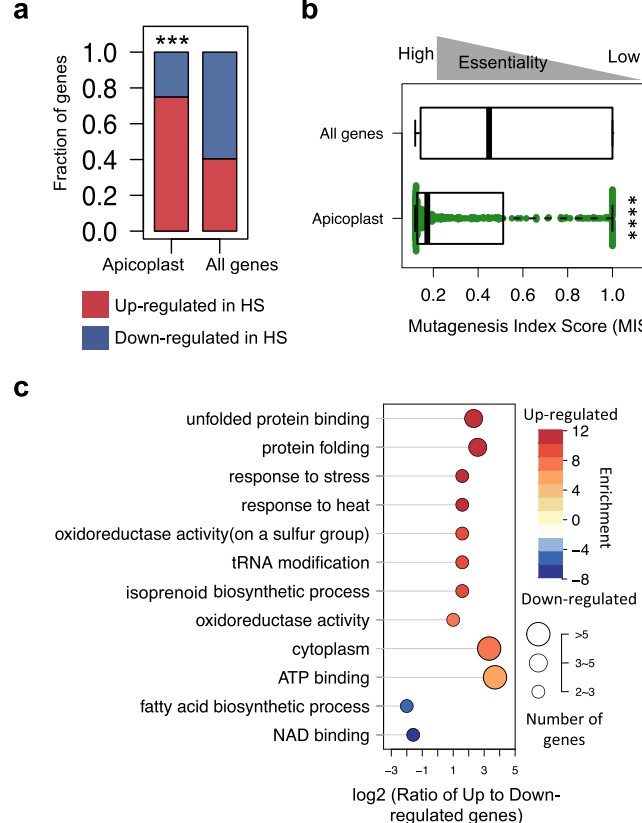

**Fig. 4 Apicoplast-targeted genes are upregulated in response to heat shock. a** Apicoplast-targeted genes tend to be increased in response to HS ($n = 39$ upregulated vs. 12 downregulated genes) as compared to all non-apicoplast-targeted genes detected above threshold in RNAseq ($n = 415$ upregulated vs. 611 downregulated genes). Apicoplast-targeted genes are as defined in ref. [56] (***two-tailed FDR-corrected Fisher test $p$-value < 1e −5). **b** Apicoplast-targeted genes tend to be highly essential during blood-stage vs. all other non-apicoplast-targeted genes detected above threshold in RNAseq. Median Mutagenesis Index Score (MIS) for apicoplast-targeted genes is much lower than median MIS for all other genes, indicating a lower tolerance for disruption and thus higher likely essentiality during blood-stage development than non-apicoplast-targeted genes (****two-tailed Wilcoxon test $p$-value < 1e−15). Boxplots are drawn to present an interquartile range of values (IQR). Lower bound of each box = 1st quartile, middle line = median, upper bound = 3rd quartile, and whiskers extend to at most 1.5× IQR. **c** Apicoplast pathways regulated in response to HS. GO categories enriched in upregulated and downregulated apicoplast genes are shown on a scale from red to blue, respectively. The horizontal direction indicates the log-ratio between upregulated and downregulated apicoplast genes in each category. Circle size represents the number of genes per category. Source data are provided as a Source Data file.

present an evolutionary explanation that endosymbiosis of the apicoplast's algal progenitor enabled parasite survival of extreme temperatures.

**Processes enabling parasites to survive fever also drive resistance to artemisinin.** We noted similarities between processes we identified to be driving the parasite HS-response and altered sensitivity to ART from our previous analysis using a chemogenomic profiling method[23]. In addition, similarities extended to pathways implicated in parasite resistance to artemisinin, including protein-damage and oxidative stress responses[12,13,15]. Therefore, we adapted the chemogenomic profiling method to use QIseq analysis for a

series of parallel phenotype-screens of our $pB$-mutant pilot-library using sublethal concentrations of two artemisinin compounds (dihydroartemisinin, DHA; artesunate, AS), heightened conditions of oxidative stress of RBCs, and exposure to a proteasome inhibitor (Bortezomib; BTZ) to investigate the possible relationship between HS-response and artemisinin MOA, as well as Oxidative-Screens (Fig. 6a, Supplementary Fig. 2a, and "Methods" section). We found that HS-Sensitive mutants tended to be sensitive to both artemisinin derivatives and $H_2O_2$-induced oxidative stress, while HS-Tolerant mutants were less sensitive to either condition (Fig. 6a). HS-Sensitive mutants also shared an increased sensitivity to the proteasome inhibitor BTZ, consistent with laboratory observations connecting artemisinin MOA to the proteasome and clinical data that proteasome-inhibitors act synergistically with artemisinins[13,24–26]. Overall, the correlation of mutant phenotypic profiles across screens varied, with 16–45% having correlating phenotypes in at least one additional screen (Fig. 6b and Supplementary Fig. 2b, Supplementary Data File 4).

We next assessed whether these laboratory-based experimental findings corresponded to changes associated with $P.$ $falciparum$ in artemisinin-resistant (ART-R) clinical isolates[15]. Consistent with our laboratory findings linking HS-sensitivity and ART-sensitivity, we found that mRNA levels of HS-Sensitive genes are significantly positively correlated with parasite clearance half-life under treatment with artemisinin-based combination therapies in recent field isolates compared with HS-Tolerant genes[15] (Fig. 6c and Supplementary Data File 2). We also compared genes by HS-response expression category to mRNA expression levels in these field isolates, finding that genes upregulated in response to heat stress are significantly positively correlated with parasite clearance half-life, while genes downregulated in response to heat stress are more likely to be negatively correlated (Fig. 6d and Supplementary Data File 3). Therefore, we conclude the parasite's responses to heat shock mirror the responses to artemisinin as both are similar types of cellular stress on the parasite. Both of these stressors induce unfolded-protein responses, which include both upregulation and downregulation of metabolic activities that enable the parasite to tolerate the toxic effects of accumulating damaged proteins. The upregulated processes include the proteasome core and chaperones to degrade or refold damaged proteins, while many other aspects of metabolism, including growth-related anabolic processes, are downregulated to prevent the build-up of new proteins that may be damaged.

Artemisinin is activated by the degradation of host hemoglobin. Recent evidence has suggested two key, temporally distinct ART-R mechanisms: (1) a multi-functional protein long associated with resistance in field isolates, $kelch13$ (K13) confers resistance upstream of hemoglobin degradation by modulating an associated endocytosis pathway; and (2) downstream of hemoglobin degradation through the ubiquitin-proteasome system (UPS), where K13 may function as, or regulate, a ubiquitin ligase[15–17,27–30]. In upstream resistance, endocytic transport of hemoglobin to the digestive vacuole (DV) is downregulated as this is the key process through which the parasite ingests, degrades, and then releases hemoglobin. K13 mutant-isolates appear to downregulate processes along this endocytosis pathway, decreasing parasite hemoglobin digestion and release of heme to activate artemisinin, thereby increasing parasite survival. We found that K13-defined endocytosis is also downregulated in response to HS (Fig. 6e). As the K13-mediated endocytosis pathway culminates in host hemoglobin-cargo being degraded in the DV, we further assessed our 1k HS-screen for DV-associated proteins. We found several DV-associated proteins were sensitive to heat shock, including key DV resident-proteases Plasmepsin I and M1-family alanyl aminopeptidase (10 of 22 mutants, Supplementary Fig. 3a)[31]. We next evaluated our 1K-library

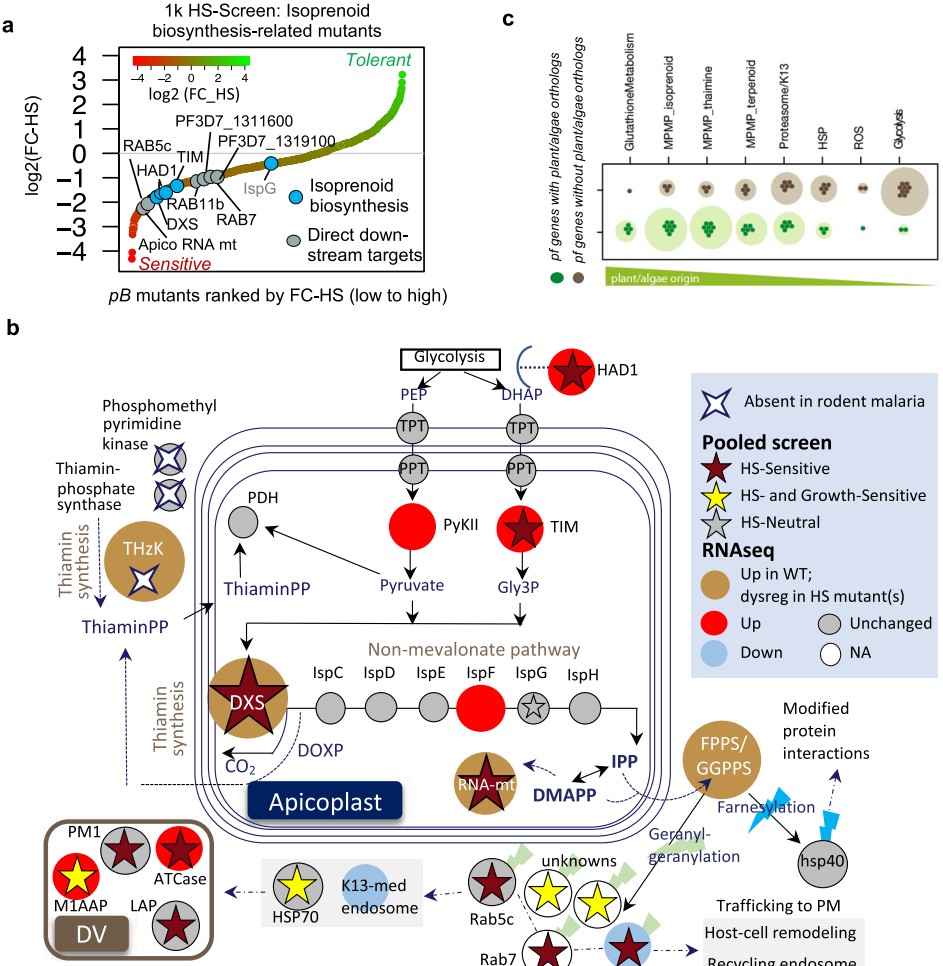

**Fig. 5 Apicoplast isoprenoid biosynthesis is critical for *P. falciparum* survival of febrile temperatures. a** Ten of 11 *pB*-mutants in genes related to apicoplast isoprenoid biosynthesis represented in the 1K-library were HS-Sensitive. Mutants are ranked by phenotype from HS-Sensitive (red) to HS-Tolerant (green). Circles indicate each HS-Sensitive mutant related to isoprenoid biosynthesis. *The isoprenoid biosynthesis genes we identified as directly upregulated in response to HS via RNAseq (DXS, PF3D7_1337200; apicoplast RNA methyltransferase, PF3D7_0218300). **b** Key enzymes in the *P. falciparum* isoprenoid biosynthesis pathway are upregulated in response to heat-shock (red circle), dysregulated in HS-Sensitive mutants (ocher), and absent in malaria-parasites of hosts that do not present fever. Pathway diagram modeled from[57]. Isoprenoid biosynthesis genes upregulated in HS include DXS, 2-C-methyl-D-erythritol 2,4-cyclodiphosphate synthase (IspF, PF3D7_0209300), pyruvate kinase II (PyKII, PF3D7_1037100), phosphoenolpyruvate/phosphate translocator (PPT, PF3D7_0530200), triosephosphate isomerase (TIM, PF3D7_1439900), triosephosphate transporter (TPT, PF3D7_1218400), and upstream-regulator of MEP-pathway substrates HAD1-phosphotase (HAD1, PF3D7_1033400)[58]. All direct downstream-targets prenylated by bifunctional farnesyl/geranylgeranyl diphosphate synthase (FPPS/GGPPS, PF3D7_1128400) with products of the MEP-pathway (zigzag) represented in pooled screening were HS-Sensitive, including the Rab-family vesicular trafficking proteins (Rab5c, PF3D7_0106800; Rab7, PF3D7_0903200; Rab11b, PF3D7_1340700), as were several digestive vacuole proteases and proteins involved in hemoglobin digestion (PM1, PF3D7_1407900; ATCase, PF3D7_1344800; M1AAP, PF3D7_1311800; LAP, PF3D7_1446200; HSP70, PF3D7_0818900). The key thiamin-synthesis enzyme hydroxyethylthiazole kinase (ThzK, PF3D7_1239600) is absent in *P. berghei* and *P. yoelii*, malaria-parasites whose rodent hosts do not present fever. **c** *P. falciparum* genes with plant orthologs (green circles) indicating potential endosymbiont-ancestry tend to be increased in response to HS vs. genes that do not have plant orthologs (gray circles). *P. falciparum* genes with potential endosymbiont-ancestry were derived from 1919 ortholog-pairs between *Arabidopsis thaliana* and *P. falciparum* (data from OrthoMCLv5.0). The listed processes are sorted based on the ratio of plant-like to non-plant-like orthologs. Source data are provided as a Source Data file.

HS-Screen for direct K13-interacting partner-proteins recently identified via immunoprecipitation[30], and found that 12 of 30 mutants in putative K13-partner-proteins represented in the screen were sensitive to HS. Further, 6 of 10 mutants in predicted alkylation targets of artemisinin represented in our screen had sensitivity to HS[14,31] (Supplementary Fig. 3b). We noted significant overlap in each of these categories of ART MOA-related genes and isoprenoid biosynthesis-related genes (Supplementary Fig. 3c). Of the ~300 genes experimentally implicated

across each of these ART-associated processes and isoprenoid biosynthesis, 47 unique genes were represented in our 1K-library ($n = 58$; Source Data). Altogether, these mutants were significantly more sensitive to heat stress than non-associated mutants (Wilcoxon *p*-value < 0.03; Supplementary Fig. 3d). There was no difference between the groups in ideal growth conditions (Wilcoxon *p*-value > 0.5).

In a second downstream step post-activation of artemisinin, the parasite engages the UPS to further mitigate artemisinin-

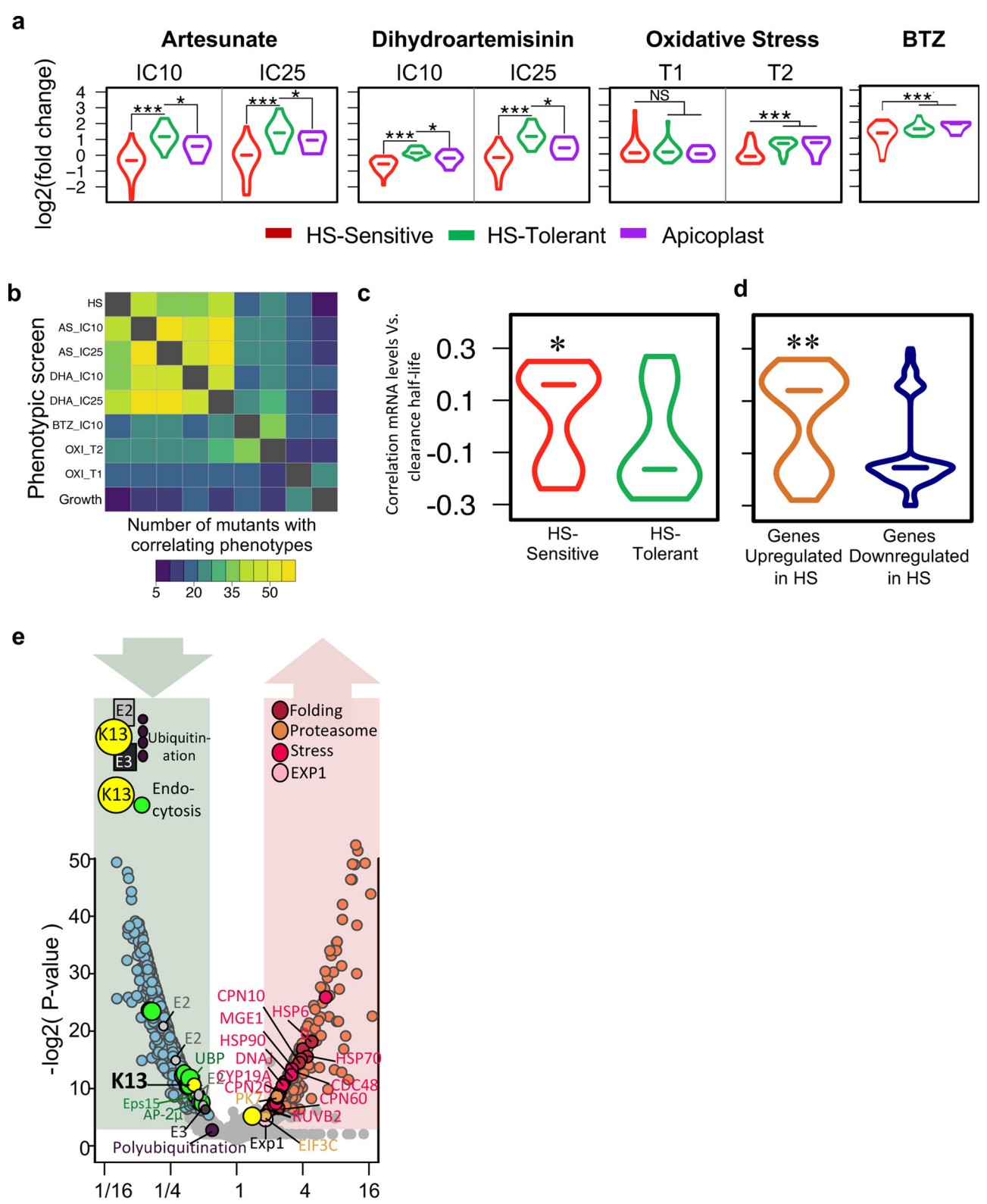

induced damage. Artemisinins mount a multi-pronged attack against the parasite by causing a global, non-specific accumulation of damaged parasite proteins, which are then polyubiquitinated/marked for degradation, while also inhibiting proteasome-function. These polyubiquitinated proteins ultimately overwhelm the parasite's decreased capacity for UPS-mediated protein degradation[13]. Key ubiquitinating components of this system, including E2/E3 ligases and K13, are downregulated in response to HS, while key components of the UPR and protein folding are increased (Fig. 6e). In contrast, components of the core proteasome were universally increased in response to HS when considered in aggregate, although the change did not meet our fold-change criteria for being HS-regulated (Supplementary Fig. 4a).

**Fig. 6 Increased sensitivity to fever is directly correlated with increased sensitivity to artemisinin in the malaria parasite. a** HS-Sensitive *pB*-mutants (red) are more sensitive to artemisinin derivatives AS and DHA, proteasome-inhibitor BTZ, and conditions of heightened oxidative stress (T1 and T2, "Methods" section) than HS-tolerant parasites (green) in all pooled screens of the pilot library. HS-Sensitive mutants tended to be sensitive to both artemisinin derivatives and $H_2O_2$-induced oxidative stress, while HS-Tolerant mutants were less sensitive to either condition. HS-Sensitive mutants also had increased sensitivity to BTZ (Supplementary Data File 4 and Supplementary Figs. 2, 10). Biological replicates ($n = 2$) were highly correlated for all screens (Pearson correlation >0.94; Supplementary Figs. 2 and 10). Mutants in apicoplast-targeted genes (purple; $n = 5$) and HS-Sensitive mutants ($n = 28$) had similar phenotypes to artemisinin derivatives, but not to proteasome inhibitors or oxidative stress (*two-tailed Wilcoxon $p < 0.05$; ***two-tailed Wilcoxon $p < 1e-10$). **b** Correlation between mutant phenotypes in all pooled screens of the pilot library. Mutants performing in the bottom 25% or top 25% of each screen were classified as Sensitive and Tolerant, respectively. Mutants falling into the same category in pairwise comparisons between screens were considered to have correlating phenotypes. **c** Compared with HS-Tolerant genes (green, $n = 16$ mutants), mRNA levels of HS-Sensitive genes (red, $n = 29$ mutants) are positively correlated with parasite clearance half-life under ACT pressure in field isolates[15]. (*two-tailed Wilcoxon $p$-value < 0.05). **d** Genes upregulated in HS (red, $n = 67$) are positively correlated with parasite clearance half-life under ACT in field isolates[15]. Downregulated genes are more likely to be negatively correlated with parasite clearance half-life (green, $n = 114$). (**two-tailed Wilcoxon $p$-value < 1e−3). **e** Both K13-mediated mechanisms of artemisinin resistance (endocytosis, ubiquitin-proteasome system) are similarly regulated in HS. RNAseq data are plotted by average log2 fold-change in HS and significance (−log2 of FDR-corrected two-tailed Fisher test). Circles in shades of blue and pink indicate genes downregulated or upregulated in HS, respectively. Source data are provided as a Source Data file.

Synthesizing these data, we present a model for the relationship between what is currently understood of artemisinin MOA and HS-response (Fig. 7). The canonical parasite-response to fever is to increase protein folding and UPR while inhibiting ubiquitination to prevent the accumulation of toxic, polyubiquitinated protein aggregates. The parasite simultaneously increases its capacity for proteasome-mediated degradation—ultimately enabling it to resolve HS-instigated stress and thus survive febrile temperatures (Supplementary Fig. 4b). As heat-stress is also injurious to the host RBC, the parasite diverts resources to stabilize the host cell—increasing export and trafficking of proteins involved in host-cell remodeling that support fortification of the host-cell membrane, as well as decreasing uptake of host-cell hemoglobin through the K13-mediated endocytosis pathway—processes which are ultimately driven by prenylation downstream of apicoplast isoprenoid biosynthesis. Artemisinins kill by overwhelming these same pathways: damaging and unfolding proteins, preventing folding of newly synthesized proteins, and inhibiting the proteasome, while at the same time activating ubiquitination machinery to ensure the accumulation of toxic polyubiquitinated proteins that eventually cause cell death. ART-R-associated mutations allow the parasite to constitutively activate unfolded-protein response mechanisms which increase its capacity for refolding or degrading those toxic proteins[32]. The overall increase in damaged-protein degradation capacity allows ART-R parasites to keep up with the influx of artemisinin-induced protein damage, clearing the waste and enabling parasite survival. This direct inverse relationship in activation of endocytosis, the ubiquitin-proteasome system, and other pathways underlying DHA-mediated killing and febrile-temperature survival, supports a shared mechanism for artemisinin resistance and HS-response, suggesting that ART-R parasites evolved to harness canonical HS-survival mechanisms to survive artemisinin.

## Discussion
Our data indicate that the parasite crisis response to HS is multi-faceted to relieve the build-up of heat-damaged proteins before it is overwhelmed by toxic, misfolded-protein aggregates. Responding to or perhaps preventing a build-up of potentially toxic heat-damaged proteins, the parasite upregulates the expression of chaperones to stabilize and detoxify them, down-regulating ubiquitinating enzymes to discourage their aggregation while upregulating the core proteasome and vesicular trafficking to degrade and eliminate proteins which cannot be repaired. Equally important in the survival response change in redox homeostasis, lipid metabolism, cellular transport, and metabolic processes associated with the endosymbiont-derived organelles.

The parasite requires increased energy to mount this febrile response, which it provides by redirecting its own internal biosynthetic pathways to produce glucose. Interestingly, we confirm the parasite's protective response mechanisms include proteins exported into the erythrocyte, suggesting that the parasite's metabolic processes exported to remodeled cytoplasm of the parasitized host cell are equally vulnerable and vital to malaria parasite survival.

Apicoplast-targeted genes tend to be upregulated in HS and also tend to be essential under ideal growth conditions (Fig. 4a, b). The apicoplast isoprenoid biosynthesis pathway's critical involvement in the survival of febrile temperatures is nevertheless a surprise, as it has not been implicated before in the *Plasmodium* HS-response. We consider it unlikely that observed differences between isoprenoid biosynthesis-gene expression in wild-type and HS-Sensitive mutants are an artifact of cell-cycle differences, as both HS-Sensitive mutant clones selected for RNAseq have no observed phenotype in ideal growth conditions (e.g., no cell-cycle defects). As an additional control for any potential differences in cell-cycle length, all mutant and WT cultures were highly synchronous and harvested for RNAseq based on morphology, rather than precise time point post-exposure. Additionally, though isoprenoid biosynthesis genes are highly essential (and were not disrupted in the coding regions in our saturation mutagenesis)—the vast majority of isoprenoid biosynthesis-related mutants represented in pooled screening were non-coding and had no phenotype in ideal growth conditions, indicating no appreciable difference in cell-cycle length and supporting a specific role for isoprenoid biosynthesis in the heat-shock response. Isoprenoids are required for myriad functions across the tree of life—plant chloroplasts, algae, some parasitic protozoa, and bacterial pathogens utilize a specialized form of this pathway absent from all metazoans (also called the MEP or DOXP non-mevalonate pathway), which has made isoprenoid biosynthesis an attractive target for intervention against a range of pathogens[33,34]. Most studied organisms make wide use of protein-prenylation and have large prenylated proteomes; malaria parasites, in contrast, have a very small prenylated blood-stage proteome (~20 proteins) consisting primarily of vesicular trafficking proteins, notably the Rab-family GTPases[21,22]. Recent studies indicate the key essential function of isoprenoids in the parasite blood stage is in their roles as substrate for protein-prenylation—specifically, in prenylating proteins driving vesicular transport to the digestive vacuole[35,36]. In the absence of prenylation, Rab5 trafficking is disrupted, which leads to digestive vacuole-destabilization and parasite death[36]. Notably, artemisinin also disrupts digestive vacuole-morphology, resulting in a very similar phenotype as a consequence of its activation via hemoglobin digestion[37,38]. Intriguingly, recent data confirm the association of key resistance-mediator K13 with Rab-

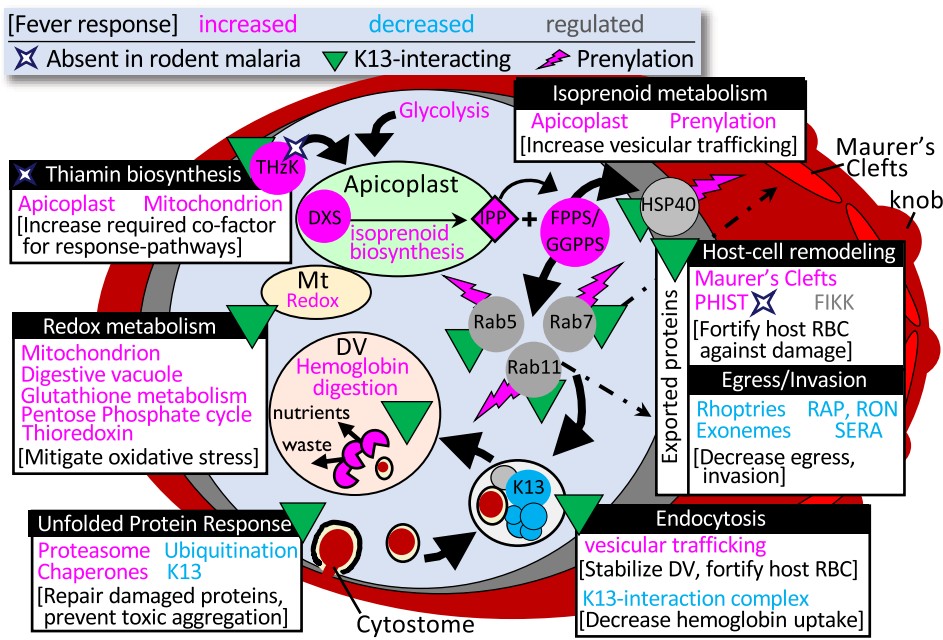

**Fig. 7 Proposed model integrating key pathways underlying the parasite survival of host fever and artemisinin resistance.** All pathways were identified via pooled phenotypic screening. The direction of regulation in response to HS is informed by comparative RNAseq data where available (pink = increased; blue = decreased). Pathways/proteins previously identified as interacting with K13 are indicated (green triangle). Prenylated proteins are indicated with a pink lightning bolt. Proteins without orthologs in rodent malaria parasites (whose hosts are non-febrile) are indicated with a white diamond. FV food vacuole. Modified from Oberstaller et al.[59]. Source data are provided as a Source Data file.

GTPases[30], adding to the repertoire of proteins comprising K13-mediated endocytic vesicles, and by extension supporting the role of prenylation in K13-mediated processes associated with ART MOA.

Another key parasite-defense against oxidative stress induced by pro-oxidant compounds (such as artemisinin) includes increased vitamin E biosynthesis—another exclusive function of the MEP isoprenoid biosynthesis pathway, whose stress-related regulation has been extensively studied in plants[39,40]. Further insights to the role isoprenoids play in the HS-response may be gleaned from plants and pathogenic bacteria, where research suggests key branchpoint-enzyme DXS, which catalyzes the first and rate-determining step of the MEP pathway[41], has a role in sensing and then facilitating adaptation to ever-changing environmental conditions, including temperature, light-exposure, chemical compounds, and oxidative stress (for example, refs. [42,43]). Elevated levels of isoprenoids have been found to correlate with plant exposure to drought and other stressors and are considered a key component of plant defenses against abiotic stress[44]. The DXS ortholog may play a similar role in *P. falciparum*, enabling the parasite to mount quick responses to unfavorable conditions in the host environment, such as fever.

Interestingly, concurrent studies now provide mechanistic insights illuminating the biochemical relationship between apicoplast isoprenoid biosynthesis and the parasite febrile-temperature survival response[45]. Farnesylation of HSP40 (PF3D7_1437900), a type of prenylation mediated by the MEP pathway, is critical for *P. falciparum* survival of thermal stress. In this study, inhibition of isoprenoid biosynthesis ultimately resulted in the reduced association of HSP40 with critical components of the cytoskeleton, protein-export, and vesicular transport pathways—without which *P. falciparum* could survive neither heat nor cold stress. Suppression of these cellular processes by loss of HSP40-farnesylation directly corresponds to HS-sensitive pathways identified via both our forward-genetic screen and our gene-expression analyses of the HS-Sensitive Δ*LRR5*- and Δ*DHC4* mutant clones.

Few eukaryotes are known to be able to thrive in extreme-heat environments; most are unable to complete their lifecycles above 40 °C[46]. The survival mechanism of malaria parasites could be attributed to the algal ancestral lineage of the apicoplast. Some extant red algal-lineages (genus *Cyanidioschyzon*) are extremophilic inhabitants of acidic hot-springs and are remarkably resistant to heat shock up to 63 °C; green-algae *Chlamydamonas reinhardtii* was also able to survive to 42 °C[47]. Responsibility for this extreme resistance to transient exposure to high temperatures was attributed to two genes of the small heat shock protein (sHSP) family (CMJ100C and CMJ101C). The *P. falciparum* ortholog for these genes (PF3D7_1304500) was upregulated in the wild-type HS-response and dysregulated in both our HS-Sensitive mutants, indicating its contribution to parasite survival in extreme temperatures. Mutations in this gene were not represented in our pooled screens.

It is tempting to speculate that presence of the endosymbiont cyanobacterium-related ancestral genes and its associated plant stress-response mechanisms is what enabled the ancestral parasite to survive host-fever, likely an important and early step leading to successful infection of hominid hosts. Our findings of significant overlap between parasite-responses to three disparate stressors (HS, artemisinin, oxidative stress) offers insight into how *P. falciparum* exhibited artemisinin resistance even in the initial clinical trials[48], and then further evolved resistance relatively quickly after mass-introduction of the drug by hijacking and repurposing the parasite's in-built fever-response pathways.

Deeper knowledge of parasite biology is expected to enable more effective and likely longer-lasting antimalarial interventions. Similarly, a better mechanistic understanding of artemisinin MOA will lead to better combination therapies to combat emerging resistance. With this large-scale forward-genetic screen in *P. falciparum*, we revealed the parasite's survival responses to malarial fever and artemisinin chemotherapy share common underpinnings that heavily depend on metabolic processes of plant origin. Our data support an increasingly prevalent

appreciation that parasite stress-response mechanisms are complex—resistance phenotypes cannot be fully explained through characterization of any single gene, nor can the complete catalog of epistatic interactions that ultimately results in survival of any stress condition be detailed mechanistically from genome-scale approaches alone. These data should be viewed as a systems-level map guiding the further mechanistic study of implicated genes, pathways, and points of interaction between those pathways.

ART-R ultimately hinges on highly efficient protein-degradation mechanisms. This mechanistic knowledge allows for the application of intelligently considered counters to ART-R, such as combinatorial therapy with proteasome-inhibitors, which has experimentally shown great promise[49]. Our current study highlights the potential of forward-genetic screens to elucidate unexpected processes and pathways, such as DOXP and iso-prenoid biosynthesis, that are associated with the artemisinin MOA which may serve as synergistic druggable targets[50]. Future studies can exploit a genome-wide screening approach to iteratively ascribe function to every part of the malaria-parasite genome to support targeted development of new, more efficacious antimalarial combination therapies to limit and potentially reverse artemisinin resistance.

## Methods

**Pilot-library of pB-mutant clones: characteristics and validation.** The single piggyBac-transposon insertion sites of each pB-mutant-clone in the pilot library were verified as previously described[18,19]. Briefly, we demonstrated growth rates of individual pB-mutant clones were highly reproducible between biological replicates, as well as between pools with different compositions; growth for all 128 clones confirmed in subsequent growth screens across 12 asexual intraerythrocytic development cycles (24 days, with samples collected in biological duplicate at cycles 3, 6, 9, and 12). Additionally, whole-genome sequencing performed on 29 of the 128 pB-mutant-clones in the pilot library verified that no major genomic changes occurred aside from the piggyBac insertion, ensuring any detected phenotypes are attributable to the single disruption[10]. The pilot library was generated in a manner to ensure approximately equal representation of each of the 128 clones at thaw[18].

**Generating the pilot library of pB-mutant parasite clones.** The pilot library was built as described in our previous QIseq methods-development study[18] and data are available in PlasmoDB (RRID:SCR_013331). Aliquots of the pilot library were generated by first growing each of the 128 extensively characterized mutant clones individually in T25-flasks to 1–2% parasitemia. All clones were then combined equally into one large flask and gently mixed. One-hundred equal-volume aliquots of the pilot library were then cryopreserved according to standard methods, providing enough biological-replicate samples for use in the parallel phenotype screens of the pilot library.

**Pooled-screen assay-design**

*HS-screens.* The pooled phenotypic screen-design pipeline has three important steps to ensure quality control and scalability: (1) protocols are tested using individual pB-mutant clones; (2) methods are adapted for pooled-screening using the well-characterized pilot-library; (3) methods developed using the pilot-library are applied to 1K-library screens (Supplementary Fig. 1). We exposed pools of pB-mutant parasites to three rounds of temperature-cycling to simulate the cyclical pattern of fever characteristic of human malaria (Fig. 1a). Parasites under phenotypic selection (heat-shock) and ideal-growth controls originated from the same thaw, grown at 37 °C for one cycle then split equally into five flasks (three flasks A, B, and C for exposure to heat-shock, samples were harvested from these three flasks at the same time as three technical-replicates for HS-Screens; two flasks C and D for the ideal-growth controls). Experimental and control flasks were maintained in parallel to minimize potential batch effects. Parasites were grown for one cycle at 37 °C until they reached the ring-stage of development as verified by Giemsa smear (Time point 0; $T^0$), at which point the experimental group were exposed to febrile temperatures (41 °C) for 8 h. Post-heat-shocked parasites were then returned to 37 °C for the remainder of the 48-h window until they again reached ring-stage. Parasite-gDNA was harvested for QIseq after two more rounds of temperature-cycling in successive growth cycles to ensure enough parasite material was available for QIseq (Time point 1; $T^1$). Control-parasites were harvested for gDNA before and after three cycles of pooled growth at 37 °C ($T^0$ and $T^1$, respectively) for quantification via QIseq in technical triplicate. We used QIseq-reads obtained for each mutant after the same number of cycles of pooled growth at 37 °C as our $T^0$ control as previously reported[18]. Pilot-library screens were performed in biological duplicate. As the 1K-library consists of multiple randomly selected, uncloned, large mixed-population pools and direct biological

replication are not feasible, we leveraged insertions duplicated across pools as internal controls. FC-HS for 15 insertion-sites represented in at least two different pools of the 1 K library allowed evaluation of consistency across pools. FC-HS was highly correlated between duplicate insertion sites regardless of the pool in which they were screened (Pearson correlation = 0.806; Supplementary Fig. 7a, b). We further evaluated reproducibility between the pilot library and the 1K library using mutants in genes represented in both the pilot library and the 1K-library (n = 16 genes; max distance between the pilot library and 1K-library insertion < 1 kb). FC-HS was again highly correlated across pools (Supplementary Fig. 8a, b, Pearson correlation = 0.702).

*Drug-screens.* As with the HS-screen, parasites were split from the same thaw of the pilot library after one cycle of growth into experimental flasks and control flasks. Experimental flasks were exposed to three cycles of continuous drug pressure at two different concentrations (IC10, IC25) of each artemisinin-compound (AS, DHA). Proteasome-inhibitor BTZ-experiments were performed at IC10. Control flasks were cultured continuously in parallel at 37 °C without drug. Parasites were harvested immediately at the conclusion of three growth cycles for gDNA-extraction and phenotype analysis via QIseq.

*Oxidative stress screens.* Parasites were split after one cycle of growth from the same thaw of the pilot library as the HS-screen. Parasites were grown one more cycle, then split into four flasks: two control flasks to be cultured with standard, washed human red blood cells (hRBC), and two experimental flasks to be cultured with $H_2O_2$-treated hRBCs to mimic conditions of oxidative stress. Experimental flasks ($H_2O_2$-treated hRBC) and control flasks (untreated-hRBC) were cultured continuously in parallel at 37 °C. Parasites were harvested immediately after three growth cycles (T1), then again after an additional three growth cycles (T2) for gDNA-extraction and phenotype analysis by QIseq.

Methods for oxidative pre-treatment of hRBCs were as published previously[51]. Briefly, O + hRBCs (Interstate blood bank, packed, 100% hematocrit) were incubated with 1 mM $H_2O_2$ (Sigma-Aldrich, Cat. no. H1009-100ML) for 1 h at room temperature. After treatment, cells were washed three times with phosphate-buffered saline (PBS) before dithiothreitol (DTT) was added to a final concentration of 1 mM to heal any reversible oxidative damages. Cells were then treated with menadione sodium bisulfite for one hour at room temperature (Sigma-Aldrich Cat. no. M5750-100G) and washed five times. A volume of 3–4 ml of AB medium (RPMI 1640 medium supplemented with 2 mM L-glutamine, 25 mM HEPES, 100 μM hypoxanthine, and 20 μg ml$^{-1}$ gentamicin) was added on top of the cell pellet after discarding the final wash. Pre-treated erythrocytes were stored at 4 °C before use in parasite culture.

All pooled phenotypic screens of pilot-library (AS, DHA, BTZ, oxidative stress, ideal growth) were performed in biological duplicate (Supplementary Fig. 10).

*QIseq.* QIseq, which uses Illumina next-gen sequencing technology and custom library-preparation to enable sequencing from both the 5′ and 3′ ends of the piggyBac transposon out into the disrupted genome-sequence, allows quantitative identification of each pB-mutant line by its unique insertion-site within mixed-population pools of pB-mutants[18] (Fig. 1b). The anatomy of the piggyBac transposon and its distinct 5′ and 3′ inverted terminal-repeat sequences (ITRs) allows double-verification of insertion-sites; both 5′ and 3′ QIseq libraries were therefore generated and sequenced for each sample. Counts per insertion site were determined as described previously[18]. We observed a high correlation between biological replicates at 41 and 37 °C, respectively (Pearson correlation = 0.964 at 41 °C and 0.967 at 37 °C, Supplementary Fig. 5a). We observed a lower correlation between Growth (37 °C) and HS (41 °C) assays (Supplementary Fig. 5b, average Pearson correlation = 0.723), suggesting that our heat-shock exposure conditions are sufficient to allow reproducible detection of mutants with specific selection response phenotypes from pooled screening.

*Calculating mutant fold-change in pooled screening to assign HS- and Growth-phenotypes.* We defined FC-Growth by pB-mutant fold-change after three cycles of growth at ideal temperatures ($T^{1-37C}/T^{0-37C}$). FC-HS was defined as pB-mutant fold-change after exposure to heat-shock vs. the non- heat-shocked control ($T^{1-41C}/T^{1-37C}$). We used changes in reads-number detected for each pB-mutant in the Growth-Screen and the HS-Screen as compared to reads-number detected for that mutant in the respective control-screen to calculate mutant Fold Change (FC) in both screens (see "Methods" section). We then ranked mutants from lowest to highest FC, with the lowest FC indicating the highest sensitivity to the screened condition.

We developed a scoring system to distinguish mutants with phenotypes specifically in the condition under selection (HS) vs. those with inherently compromised growth in ideal conditions, called the Phenotypic Fitness Score (PFS). $PFS_{HS}$ is the mutant fold-change in response to heat-shock (FC-HS, 41 °C/37 °C) multiplied by the ratio of FC-HS to mutant fold-change under ideal growth-conditions (FC-HS/FC-Growth), with the smallest and largest values indicating the largest mutant growth-differentials between the two screens (smallest $PFS_{HS}$ indicating worse mutant-fitness in the HS-Screen than the Growth-Screen, and largest $PFS_{HS}$ indicating better mutant-fitness in the HS-Screen than the Growth-screen). Mutants exhibiting (1) poor growth in the HS-Screen (i.e., low FC-HS of <0.5 based

on the performance of *known HS-Sensitive *pB*-mutant-clones), and (2) comparatively much better growth in the Growth-Screen (i.e., low PFS$_{HS}$ of <0.25) were classified as HS-Sensitive in pooled phenotypic screens (indicated in red in Fig. 1c, d). Mutants exhibiting poor fitness in both the Growth- and HS-Screens (FC-HS < 0.5 and PFS$_{HS}$ > 0.25) are indicated in Fig. 1c, d in yellow ($n = 14$). These double-sensitive mutants were not included in our HS-Sensitive classification to avoid overinterpretation of possibly confounding phenotypes. We classified mutants displaying a slight growth advantage in response to heat shock (FC-HS > 1.5, $n = 28$, indicated in the green box, Fig.1c, d) as HS-Tolerant. Mutants exhibiting neither sensitivity nor tolerance to heat shock were classified as HS-Neutral ($n = 49$).

*Assigning drug- and oxidative stress-screen phenotypes.* Mutant fold-change in response to the given condition was calculated against an ideal-growth control as above. Mutants in the top 25% of reads recovered in QIseq in the screened condition were classified as Tolerant, while mutants in the bottom 25% were classified as Sensitive.

*Comparative RNAseq between wild-type NF54 and two HS-Sensitive mutant parasite lines in response to heat shock.* RNAseq experimental design is outlined in Supplementary Fig. 11a. Briefly, cultures of wild-type NF54 and HS-Sensitive mutants Δ*LRR5* and Δ*DHC* were sorbitol-synchronized 3× to highly synchronous rings, then split equally into four T75 flasks each. All parasites were grown at the normal human body temperature (37 °C) to early ring-stage. Two flasks of each parasite-line were then exposed to febrile temperatures (41 °C) for 8 h, while the remaining two flasks were allowed to continue to grow at 37 °C for 8 h without exposure to heat-stress. This temperature-cycling was repeated three times, just as we allowed for the pooled HS-Screen. After the third round of heat shock (Time 1, T$^1$), RNA was harvested simultaneously from both conditions for RNAseq as in[24]. Parasite staging was verified by Giemsa smear at every time point. Parasite fold-change in response to HS was calculated at the time of sample collection and verified mutant defects in response to HS as compared to NF54 (Supplementary Fig. 11b). RNAseq was performed in-house on an Illumina MiSeq using a 300-cycle V2 MiSeq reagent kit.

*RNAseq data-analysis.* RNAseq reads from each sample were aligned to the *P. falciparum* reference genome (PlasmoDB version 28, RRID:SCR_013331). A maximum of one mismatch per read was allowed. The mapped reads from TopHat[52] were used to assemble known transcripts from the reference and their abundances were estimated using Cufflinks[53]. The expression level of each gene was normalized as FPKM (fragments per kilobase of exon per million mapped reads). We defined expressed genes as those having FPKM > 20 for at least one biological replicate at either 37 or 41 °C. The fold-change of normalized gene expression between 41 and 37 °C was calculated for every biological replicate. Fold-change for genes not expressed in both temperatures was set equal to one. We conservatively filtered out genes in the top and bottom 10% of fold-change to remove outliers. We then fit a Gaussian model to the log2-fold change (*log2FC*) for every biological replicate using maximum log-likelihood estimation to assess the fold-change distribution The *p*-value is calculated as the probability of estimated gaussian distribution higher than the observed *log2FC* (when observed *log2FC* > the expectation of estimated Gaussian distribution), or lower than the observed *log2FC* (when observed *log2FC* ≤ the expectation of estimated Gaussian distribution). The false discovery rate (FDR) was calculated for each replicate. We defined genes for which FDR < 0.1 in both biological replicates as having significant fold-change in response to HS. Genes were assigned HS phenotype categories based on significance and direction of HS-response. We assigned HS phenotype categories for 2567 genes using these criteria (Supplementary Data File 3). Heat-shock phenotypes as identified via pooled phenotypic screening and comparative RNAseq were highly associated (Supplementary Fig. 12a, b), supporting our methodology.

*GO-term enrichment analyses.* All GO-enrichment analyses were performed testing GO-terms mapped to genes in the category of interest against a background of GO-terms mapped to all other genes in the analysis. The GO-term database was created from the latest curated *P. falciparum* ontology available at the time of analysis, downloaded from GeneDB (accessed May 2, 2019)[54]. For enrichment-analysis in the 1K-library screens: Mutants were divided into HS-phenotype categories, and each category was tested for enrichment against a background of GO-terms mapped to the genes represented by the remainder of the 922 mutants in the screen using the weighted-Fisher/elim-hybrid-method of the TopGO package (v 1.0) available from Bioconductor[55] (Fig. 2b). For enrichment-analysis in comparative RNAseq data: a database of all GO-terms mapped to the 1298 genes which could be assigned an HS-phenotype in all three parasites was assembled. Genes were divided into HS phenotype-categories based on the direction of fold-change (Up, Down, Unchanged) in response to HS in all three parasites, then evaluated for GO-term enrichment against the background GO-term database of all other genes in the analysis using the weighted-Fisher/elim-hybrid-method of the TopGO package (Fig. 3b and Supplementary Data File 3). For enrichment of apicoplast-targeted genes by RNAseq HS-phenotype category: enrichment for each investigated GO-term *g* (The *x* axis in Fig. 4c, the ratio of up to downregulated genes) was calculated as the ratio ($C_r$) of upregulated vs. downregulated genes mapped to GO-term *g* among all differential expressed apicoplast genes. This ratio (*C*) was also calculated for the genes mapped to GO-term *g* in the whole genome (the background distribution). The GO

annotation for each gene was downloaded from GeneDB (accessed May 2, 2019). The fraction of HS-regulated apicoplast genes to non-HS-regulated apicoplast genes ($C_r/C$) was assessed for significance using the Fisher exact test (Fig. 4c).

**Reporting summary.** Further information on research design is available in the Nature Research Reporting Summary linked to this article.

## Data availability

Raw QIseq data sets generated for this study were deposited to the European Nucleotide Archive under project accession code PRJEB31716 (sample accession numbers ERS571589, ERS571592, ERS571594, ERS571599, ERS571602, ERS571612, ERS571615, ERS571617, ERS571620, ERS801326, ERS801327, ERS801328, ERS801329, ERS801330, ERS801331, ERS801335, ERS801336, ERS801337, ERS801338, ERS801339, ERS801340, ERS801342, ERS801343, ERS801344, ERS801345, ERS801346, ERS801347, ERS801351, ERS801352, ERS801353, ERS801354, ERS801355, ERS801356, ERS801358, ERS801359, ERS801360, ERS801361, ERS801362, ERS801363, ERS801366, ERS801367, ERS801368, ERS801369, ERS801370, ERS801371, ERS3340779, ERS3340780, ERS3340781, ERS3340782, ERS3340784, ERS3340785, ERS3340787, ERS3340788, ERS3340789, ERS3340790, ERS3340792, ERS3340793, ERS3340795, ERS3340802, ERS3340803, ERS3340810, ERS3340829, ERS3340830, ERS3340831, ERS3340832, ERS3340833, ERS3340834, ERS3340835, and ERS3340852). Accession numbers and descriptions are listed in Supplementary Table 1. Processed QIseq data are provided in Supplementary Data File 1 (HS screens of the pilot library), Supplementary Data File 2 (HS screens of the 1K-library), and Supplementary Data File 4 (all other reported phenotypic screens of the pilot library). RNAseq data generated for this study have been deposited to the NCBI Gene Expression Omnibus (GEO) database under the accession number GSE177479. Processed RNAseq data are provided in Supplementary Data File 3. Previously published TRAC I microarray data were accessed through GEO accession number GSE59099. Unique biological materials used in the study, such as transfection plasmids and parasite clones, are available upon request and/or through MR4. Large-scale mutant libraries are provided by the authors when unused aliquots remain available. Source data are provided with this paper.

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

## Acknowledgements

We appreciate the Wellcome Sanger Institute (UK) for performing QIseq and Xiangyun Liao, Suzanne Li, and Kenneth Udenze for support of parasite cell culture. We thank the USF Genomics Program (Equipment Core and Omics Hub) for support of RNAseq and productive discussions. This work was supported by the National Institutes of Health grants R01 AI094973 and R01 AI117017 (J.H.A.), F32 AI112271 (J.O.), and the Wellcome Trust grant 098051 (J.C.R.).

## Author contributions

Conceptualization, M.Z., C.W., J.O., T.D.O., J.C.R., M.T.F., R.H.Y.J., and J.H.A.; methodology, M.Z., C.W., J.O., K.B.-S., R.H.Y.J., and J.H.A.; software, J.O. and C.W.; validation, M.Z., C.W., and J.O.; formal analysis, M.Z., C.W., J.O., S.R.A., and R.H.Y.J.; investigation, M.Z., C.W., J.O., P.T., D.C., S.B., S.X., M.M., and R.H.Y.J.; data curation, M.Z., C.W., and J.O.; writing—original draft, J.O., M.Z., and J.H.A.; writing—review and editing, J.O., M.Z., T.D.O., J.C.R., and J.H.A.; visualization, C.W., M.Z., J.O.v and R.H.Y.J.; supervision, M.Z., J.O., T.D.O., J.C.R., M.T.F., R.H.Y.J., and J.H.A.; project administration, M.Z.; funding acquisition, J.C.R. and J.H.A.

## Competing interests

The authors declare no competing interests.
