## [Peer Review File · Nature Communications]

REVIEWER COMMENTS

Reviewer #1 (Remarks to the Author):

This is an important study that illustrates the general power of libraries of gene-disrupted parasites now available for some apicomplexan parasites, and the more specific utility to address mechanisms of parasite survival of the heat-shock that is central to malaria fever. This makes a strong contribution to our understanding of this important topic and I support its publication here. There are a few areas where I believe the manuscript could be improved

Major points

- There is somewhat of a conundrum in the link between ART resistance and HS. The authors describe that genes that are mutated in ART resistance (mutations elsewhere shown to downregulate Hb endocytosis) are more likely to be required for Heat shock survival, but they also show that genes involved in ART resistance are downregulated upon heat shock survival. That is to say, a family of genes is downregulated in the same condition that they are required to be present for survival. There is no direct contradiction in this relationship per se – the gene may need to be both present and transcribed at lower abundance, but this needs some more exploration and explanation.

- I struggled with the rationale of the comparison of the dynein and LRR5 mutants to the wild type parasites as a basis for identifying dysregulated genes. Neither of these mutants has an obvious causal link to transcriptional regulation, which means that transcriptional changes in those mutants in response to heat shock are either the product of some other non-detected mutations in those parasite clones that do initiate a transcriptional change, or are instead the product of the parasite experiencing less heat-shock stress, and as a result exhibit altered abundance of some transcripts. Can the authors explain mechanistically why those mutations should result in transcriptional dysregulation? I'm not much convinced that the transcriptional profiling of those two extra parasite lines adds much that wasn't revealed by the detection of genes identified as upregulated in the wt parasites. I didn't find the authors' description of how we should interpret these data very clear – and indeed I don't see any clear mechanistic connection to the HS phenotype for those genes that were presented as being detected on the basis of difference between the wt and the dynein and LRR5 mutant parasites. Nor do I see why transcriptional dysregulation in those two lines would be in any way representative of the large number of other HS mutants that could have been profiled. Indeed the real insights from this paper, which are rightly highlighted in the abstract, are the link to apicoplast isoprenoid anabolism, and subsequent prenylation, and these insights are quite clearly demonstrated from the data in figure 4, which does not draw on the transcriptional comparison between the wt and mutants. My interpretation of the data presented in figure 3 is that the genes that are worth paying attention to here are those that are upregulated in the wt, irrespective of their behaviour in the mutants (and it's quite dicey to speculate on why some of them seem dysregulated in the mutants). Other than the very useful identification of genes generically upregulated in response to heatshock, I am quite unclear on the value and meaning of the data presented in figure three and am unconvinced that it adds to this manuscript. This all needs some clearer explanation.

Minor points

1. Figure 5F superimposes a model for survival in Heat shock that doesn't really draw much from the data from this study. The key genes identified through the RNAseq experiments and the piggybac screen highlight genes in apicoplast isoprenoid synthesis, then deployment of those isoprenoids in prenylated proteins that are involved in protein trafficking. In contrast, figure 5F presents a model centred around ubiquitination – which isn't really consistent with the data produced in this paper. Indeed the more recent data on the connection between K13 and ART-R is much more consistent with a role for K13 in trafficking than in bulk turnover of proteins through Ubiquitination. I would be much

more interested in seeing a how the authors think these data fit into an integrated model of protein trafficking.

2 Figure 4B could be easier to understand for a reader if it had a short label above the panel explaining if the high mutagenesis index correlates with more essential or less essential.

3. Some of the genes in figure 5 have multiple genes that correspond to that gene name. All need the appropriate PlasmoDB id in the figure legend. The PPT and TPT in figure 5 are drawn as transporters in the same membrane potentially importing different substrates – guessing from the names they are given, these names are presumably referring to the proteins we know now to be the transporters of the inner and outer membranes respectively (and presumably sequentially transport the same metabolites through different steps), and this should be redrawn

4. The focus of the title on the “ancient endosymbiotic origins” and the line “the most ancient genes in the parasite’s genome” are not useful foci – all of the genes in the parasite genome can be considered to realistically to be the same age, with the possible exception of those genes gained by lateral gene transfer, which are more newly arrived in the genome. It isn’t the ancientness of the endosymbiont that generates this link – instead it is the newer, eukaryotic process (prenylation) that is central to this. I suggest rewording the title and removing the reference to the “most ancient” genes.

5 The authors describe genes in the abstract and at line 111 as “apicoplast-associated” this needs to be more specific. Do they mean genes encoded in the apicoplast or (nuclear) genes for apicoplast targeted proteins? – I suspect in both cases it’s the latter, and the authors could make this more precise.

Reviewer #2 (Remarks to the Author):

The submission makes use of a panel of previously published insertion mutants in the human malaria parasite *Plasmodium falciparum* to screen for genes with important roles in parasite survival within febrile hosts. A major conclusion is that isoprenoid precursors synthesized by the apicoplast and used to prenylate proteins are an essential capacity for survival of heat shock.

A key flaw with the interpretation of the results, which is encapsulated in the title, is the claim that the acquisition of the apicoplast “enabled parasite-survival of extreme temperatures”.

This is not a reasonable conclusion. The apicoplast existed long before homeotherms arose. It is clear that the common ancestor of dinoflagellates and Phylum Apicomplexa had already acquired a plastid [1]. How old is that plastid/apicoplast-containing common ancestor? We have no dates for the origin of Apicomplexa, but there are robust fossil records for the sister lineage dinoflagellates. For instance, a recent analysis demonstrates that dinoflagellates unequivocally go as far back as the Triassic [2], and depending on whether one considers achritarchs to be dinoflagellates, the group could go as far back as the Proterozoic [3]. Therefore, the apicoplast is very ancient and undoubtedly predates the origin of the febrile response by mammals. Indeed, it possibly predates animals.

What the data do show is that protein prenylation is essential for surviving heat shock. Apicomplexa happen to use isopentenyl diphosphate made by the apicoplast to build prenyl chains to attach to proteins, but all eukaryotes make prenyl chains by one means or another, so the apicoplast is not a ‘solution’ to a parasite surviving febrile defence by a host.

The above considerations render Fig 4F (heat shock genes shared with plants and algae) not relevant.

smaller issues:

The manuscript needs to make clear whether or not these are loss of function mutants or knockdowns. Several of the genes are considered essential in blood stage growth, so I presume the insertions don't totally abrogate expression.

line 48 running title malaria is a disease not an organism

line 45 ".....responses depend on some of the most ancient genes in the parasite's genome"

This statement is fraught. Do the authors mean the absolute antiquity of the genes (i.e. how old is a given gene?), or how long the genes have been in the parasite genome? Given that the apicoplast/plastid was acquired by secondary endosymbiosis, the bulk of apicoplast related genes were acquired by the common ancestor of Apicomplexa and dinoflagellates at the time of the secondary endosymbiotic event. This ancestral, pre-secondary endosymbiosis organism already had a full complement of genes in its nucleus and mitochondrion. Those genes are therefore 'older' to the lineage than the genes picked up during secondary endosymbiosis of a plastid containing organism. However, from a broader perspective, one could argue that the plastid related genes, which ultimately derive from a primary endosymbiotic cyanobacterium, are indeed amongst the oldest of all genes since cyanobacteria are the oldest known organisms, as represented by stromatolite microfossils. Categorising genes as 'ancient' or 'recent' is not simple.

ref 45 is wrong. The authors appear to have confused Boucher's other preprint from the same year. In any case, the work is now published in a refereed journal [4], and that citation should be used for the apicoplast proteome.

The authors have focused on protein prenylation as the key role in surviving heat shock, but the isopentenyl diphosphate precursors made by the apicoplast are almost certainly used to make dolichols for glycosylation of proteins and ubiquinone tails for mitochondrial electron transport. While it is apparent from the phenotype data that protein prenylation is crucial for heat shock survival, did the authors tease out any requirement for dolichol-related processes or mitochondrial electron transport?

[1]. Janouskovec, J., Horak, A., Obornik, M., Lukes, J., and Keeling, P.J., (2010) A common red algal origin of the apicomplexan, dinoflagellate, and heterokont plastids. *Proc Natl Acad Sci U S A.* 107: 10949-54

[2] Janouskovec, J. et al (2017) Major transitions in dinoflagellate evolution unveiled by phylotranscriptomics. *Proc Natl Acad Sci U S A.* 114 (2) E171-E180

3]. Meng, F., (2005) The oldest known dinoflagellates: Morphological and molecular evidence from Mesoproterozoic rocks at Yongji, Shanxi Province. *Chinese Sci Bull.* 50: 1230-1234

[4] Boucehr, MJ and Yeh, E (2019) Disruption of Apicoplast Biogenesis by Chemical Stabilization of an Imported Protein Evades the Delayed-Death Phenotype in Malaria Parasites. *mSphere* 4: e00710-18

Reviewer #3 (Remarks to the Author):

This is a potentially interesting manuscript that seeks to identify new genes involved in the *Plasmodium falciparum* febrile heat shock response and suggests a tie to artemisinin resistance in malaria parasites. The idea sounds compelling and timely and the authors have used a powerful piggybac transposon system that they have developed—the authors' previous study on gene essentiality using the piggybac system shows the system is robust and has provided useful data to the community. In theory, the authors' collection of transposon mutants could be a powerful toolbox for functional annotation of the *Plasmodium* genome. However, as presented, the work is very difficult to evaluate. First, as written it appears to be quite a bit of "cherry-picking" going on in terms of piggybac library creation and validation. In addition, the supplemental data is not well annotated or not provided in a way that allows one to cross check the authors' work. Instead of providing exact numbers, the authors often make vague statements about tendencies which cannot be readily evaluated. In addition, there seem to be so many items that are poorly labeled or missing in the supplements and figure legends that this reviewer feels that this manuscript cannot be effectively reviewed as is. Finally, the authors should provide either A. better transparency and statistical support for their claims, especially for the piggybac library screens B. orthogonal validation (e.g. testing claims using independent gene disruptions, possibly created using CRISPR-Cas9 or C. more biological replicates (e.g. a second, independent biorep for the heat shock experiment described in Table S2). Obviously a great deal of careful work went into the study and it would be great to see this published, so it is a shame that it can't be given the evaluation it deserves because of presentation issues.

Major points:

Additional mutations that randomly arise during long term culture may affect the authors' conclusions about phenotypes. The authors should ideally use either a second independent mutant or use edited lines to check that the phenotypes are the same for their heat shock sensitive mutants (DHC and the LRR mutants)

As a genomics paper there are some statistical issues. First, the authors have done two different screens, a pilot, with a small number of genes (~200 mutants) and multiple bioreps, and a larger screen (~1000 mutants) with no bioreps. The authors then choose to validate several genes using RNASeq. However, the mutants that they choose to validate come from the small pilot screen and were previously known, and not the larger screen (Table S2) so you can't tell how good the larger screen actually was. Is this data useable or just too noisy? It seems many repetitions may be needed to confidently call sensitivity? Are the libraries really random, or were they constructed to be enriched for mutants in genes that authors find interesting? For example, kelch13?

Some parts of the manuscript are misleading and much of the work/validation is done with piggybac transposon mutants whose phenotypes were not actually discovered in this library screen (Pb4 and P31). The authors write "We chose two individual HS-Sensitive mutant clones for additional profiling via RNAseq to identify dysregulated genes responsible for this sensitivity. Both mutant lines have a single disruption in the coding region of a gene not previously implicated in the HS-response: Δ DHC (dynein heavy-chain gene PF3D7_1122900[pb4]), and Δ LRR5 (leucine-rich repeat protein PF3D7_1432400 [pb31]). Based on the positioning of the paragraph it makes it seem like this came from the larger library screen. However, these genes are only listed in the pilot screen, not the whole library screen. In addition, it says in the notes that these were actually controls that were added in. The authors write in supplemental table 1 "Known HS-Sensitive' pB-mutant clones (PB31, PB14, PB3, PB4, PB2 and PB5) were validated by individual heat shock assay (Thomas et. al., 2016)"

Minor points:

Many abbreviations are used and sometimes it is not clear what things are. This makes it hard to review. Add a table of abbreviations?

The authors often make vague unsubstantiated statements that are very difficult to review. They write "A high proportion of essential genes that are upregulated in response to heat stress are targeted to the apicoplast." I can't evaluate this statement. What is high? It would be much easier for me to understand if the authors wrote something like "Of the 45 essential genes that are upregulated in response to heat stress ($FC > 2$), 7 are targeted to the apicoplast. Given that there are only 12 genes that are targeted to the apicoplast and which were detected in our RNAseq analysis, this enrichment is very significant (p value =) or something like this... The word "tend" is used often, often without statistical support.

Supplemental tables. Some descriptions of the table calculations are found in methods but would be much nicer for the reviewer if this was placed next to the actual data as part of a legend. For example, "We defined expressed genes as those having FPKM > 20 for at least one biological replicate at either 37°C or 41°C. The fold change of normalized gene expression between 41°C and 37°C was calculated for every biological replicate." Would be more helpful as legend to supplemental table S3 tab 2. For the GO analysis, what does "annotated" mean? What does "significant" mean. For annotated do we mean number of genes with this annotation in the genome, or in Table S3? For significant, is this based on expression levels and upregulation and a statistical test?

There is no supplemental file to support Figure 4 either—you just need to trust the authors. While maybe one could go through and redo their analyses (e.g. looking up individual genes and reading other papers that are referenced) using what they have provided this would require quite a bit of reviewer time. Most good genomic papers provide more intermediate files allowing the authors' work to be checked. What genes are considered the "downstream targets."

There are no common names for genes listed in supplemental table 4. Also, incomplete column headings. What does the asterisk mean? Also, again data would be easier to interpret if the authors included legends in the tables.

In supplemental files the authors have included mutants with transposons in intergenic regions but they only list one gene that might be affected—typically an intergenic mutation can affect two genes? This issue should be addressed.

A table listing the number of different pb mutants and the types should be provided for the piggybac libraries that are screened. How often do the authors actually get phenotypes for mutants that are classified as intergenic?

Some table summarizing the quality of the RNAseq would also be useful. How many reads? How many genes detected?

Supplemental files. What do the negative distances mean? I assume this is number of bases upstream of the start?

Names of 42 isoprenoid genes and all downstream targets in the entire genome need to be provided somewhere. Likewise, the authors state "We next evaluated our 1K HS-sScreen for direct K13-interacting partner-proteins recently identified via immunoprecipitation [25], and found that mutants in 10 of the 24 unique putative K13-partner-proteins represented in the screen were sensitive to HS." What are the identities of these 24 genes? Expecting the reviewer to go read the other paper [citation 25] is burdensome. The authors could add a column to their supplemental tables in which they indicate genes that are annotated isoprenoid genes, or k13 partner genes or essential genes.

The authors write "Measures indicating reproducibility for QIseq-data for both the pilot-library and 1K-

library screens are shown in Fig. S5A-B and Fig. S6, respectively. We observed high correlation between 5' and 3' QIseq-libraries from each pool (Fig. S5A, Fig. S6). Resulting Growth and HS phenotype-assignments were highly reproducible across three technical and two biological replicates (HS-Screen, $R=0.94$; Growth-Screen, $R=0.89$; Fig. S5B).” making it seem like the large library screen has been validated, but S5 just shows the pilot data, and S6 seems is not understandable...there are no axis labels, for example. The figure legend for S6 states “1K-Library screens were as robust and reproducible as the pilot library screens. QIseq data correlations within and between HS-screens and Growth screens for a representative pool of the 1K-mutant library are shown” which doesn't exactly describe what is actually plotted.

Methods: The authors should briefly describe how the small pilot library was created and not require the reviewer to go read other manuscripts. Its composition is critical to understanding results.

REVIEWER COMMENTS

Reviewer #1 (Remarks to the Author):

This is an important study that illustrates the general power of libraries of gene-disrupted parasites now available for some apicomplexan parasites, and the more specific utility to address mechanisms of parasite survival of the heat-shock that is central to malaria fever. This makes a strong contribution to our understanding of this important topic and I support its publication here. There are a few areas where I believe the manuscript could be improved

Response

- We appreciate the reviewer's comments and helpful suggestions. The manuscript has been modified in accordance with these comments along with specific responses below (in blue) to the individual points in the critique. In addition to the revisions of the text, there are edits to areas of the main figures and additions to the supplemental files to improve statistical analysis.
- *Major points*
 - *There is somewhat of a conundrum in the link between ART resistance and HS. The authors describe that genes that are mutated in ART resistance (mutations elsewhere shown to downregulate Hb endocytosis) are more likely to be required for Heat shock survival, but they also show that genes involved in ART resistance are downregulated upon heat shock survival. That is to say, a family of genes is downregulated in the same condition that they are required to be present for survival. There is no direct contradiction in this relationship per se – the gene may need to be both present and transcribed at lower abundance, but this needs some more exploration and explanation.*

Response

- The reviewer raises an important point that we were seeking to highlight from our analysis. The parasite's responses to heat shock mirror the responses to artemisinin as both are similar types of cellular stress on the parasite. Both of these stressors induce unfolded protein responses (UPR), which include both upregulation and down regulation of metabolic activities that enable the parasite to tolerate the toxic effects of accumulating damaged proteins. The upregulated processes include the proteasome core and chaperones to degrade or refold damaged proteins, while many other aspects of metabolism are down regulated to prevent a build-up of damaged proteins.
- Figures 5C & 5D were intended to demonstrate the correlation between the genes identified in this screen and the genes with altered expression profiles in ART-R isolates of the TRAC-1 study. Unfortunately, the Y-axis labels on the figure 5C and 5D were incorrect and we apologize for the errors, which have all been corrected. We also revised the legends for Fig. 5C and 5D to improve clarity.
- We have included an additional Table S6, which provide data with figure 5C and 5D. Also, we modified the text on pp 15-16 to provide a clearer explanation that highlights the correlation of differential gene regulation in response to heat shock and observed to be associated with increased clearance times of ART-R field isolates.

5C. mRNA levels of HS-Sensitive genes identified in pooled screening ($n = 29$ *pB* mutants; red), but not HS-Tolerant genes ($n = 16$ *pB* mutants; green), positively correlated with genes up-regulated in ART-R field-isolates exposed to ACT (4) (* Wilcoxon p -value < 0.05). mRNA levels of HS-Tolerant genes trend towards inverse correlation with ART-R parasite-clearance half-life that does not rise to significance.

5D. Genes up-regulated in the wildtype-response to heat-stress are positively correlated with genes differentially expressed in ART-R field-isolates exposed to ACT (4) ($n = 67$ genes, orange; ** Wilcoxon p -value $< 1e-3$). Genes down-regulated in the wildtype-response to heat-stress trend towards inverse correlation with ART-R parasite-clearance half-life that does not rise to significance ($n = 114$ genes; blue).

I struggled with the rationale of the comparison of the dynein and LRR5 mutants to the wild type parasites as a basis for identifying dysregulated genes. Neither of these mutants has an obvious causal link to transcriptional regulation, which means that transcriptional changes in those mutants in response to heat shock are either the product of some other non-detected mutations in those parasite clones that do initiate a transcriptional change, or are instead the product of the parasite experiencing less heat-shock stress, and as a result exhibit altered abundance of some transcripts. Can the authors explain mechanistically why those mutations should result in transcriptional dysregulation? I'm not much convinced that the transcriptional profiling of those two extra parasites lines adds much that wasn't revealed by the detection of genes identified as upregulated in the wt parasites. I didn't find the authors' description of how we should interpret these data very clear

Response

- Indeed, these genes are not transcriptional regulators, but the phenotypes indicate that their functions are important in the parasite's response to heat stress. Since the parasite's metabolic functions involve an integrated network of interactions, disrupting the functions of LRR5 and DHC is expected to destabilize this network leading to a ripple effect of altered expression profiles within interaction network. Since LRR5 and DHC are functionally unrelated, dysregulated genes common to both mutants provide strong evidence for underlying components of this heat shock interaction network.
- We selected these two HS-Sensitive mutants over other candidates for phenotypic transcriptional profiling via RNAseq following strict criteria, which we have clarified in the revised manuscript (pp 8-9, lines 258-278):
 - i. Both mutants are highly sensitive to heat shock, but under ideal culture conditions grow better than most other mutants in the pilot library.
 - ii. Both are presumed loss-of-function mutants since the protein-coding sequences for both were disrupted by a *piggyBac* insertion.
 - iii. Gene functions are unrelated and not essential for parasite growth under ideal culture conditions.
 - iv. The HS-Sensitive phenotype was previously validated in a heat shock assay of individual clones.
 - v. The GO classifications of the leucine-rich repeat-5 (LRR5) and dynein heavy chain (DHC) are

representative of the broad functional categories associated with heat response, regulating gene expression and intracellular vesicular transport, respectively.

- Both DHC and LRR5 are involved in dynamic metabolic processes active in the heat shock stress response. In addition, these processes identified as being essential for the heat shock response were otherwise dispensable. As noted above, the selection of these mutants was partly based on the GO classification being generally representative of the type of processes we identified as important for HS survival in our earlier small screen and other published reports, but at the time we did the experiments the selection of DHC could be considered a bit speculative. However, from recent reports there is now little doubt of the importance of vesicular transport (e.g., K13, coronin) in the parasite's response to ART. In addition to this dynein (PF3D7_1122900) from the pilot library, we modified Figure 2A to highlight three more dynein mutants that were included in the 1k library-screen (two different mutants of PF3D7_1202300 and one mutant of PF3D7_1023100). All three of these additional dynein-mutants were also heat shock-sensitive, supporting the relative importance of this functional process in the parasite's stress response.
- The purpose of the differential expression analysis by RNAseq was to reveal the global effects on the linked network of interacting partners caused by disrupting the functions of these genes. These mutants were chosen because their functions are unrelated and not essential for parasite growth under ideal culture conditions, but both were experimentally validated to be essential for parasite survival of heat shock stress. Therefore, while some of the altered expression in each mutant is unique to their respective functions, the interconnections revealed by our comparative analysis is expected to identify the shared stress-response mechanisms.

– and indeed I don't see any clear mechanistic connection to the HS phenotype for those genes that were presented as being detected on the basis of difference between the wt and the dynein and LRR5 mutant parasites. Nor do I see why transcriptional dysregulation in those two lines would be in any way representative of the large number of other HS mutants that could have been profiled.

Indeed the real insights from this paper, which are rightly highlighted in the abstract, are the link to apicoplast isoprenoid anabolism, and subsequent prenylation, and these insights are quite clearly demonstrated from the data in figure 4, which does not draw on the transcriptional comparison between the wt and mutants. My interpretation of the data presented in figure 3 is that the genes that are worth paying attention to here are those that are upregulated in the wt, irrespective of their behaviour in the mutants (and it's quite dicey to speculate on why some of them seem dysregulated in the mutants). Other than the very useful identification of genes generically upregulated in response to heat shock, I am quite unclear on the value and meaning of the data presented in figure three and am unconvinced that it adds to this manuscript. This all needs some clearer explanation.

Response

- We agree with the reviewer that the differential expression analysis of the wild-type parasites does provide the prime list of candidate genes involved in the parasite's heat shock response. However, we disagree that nothing is gained from analyzing the defects of the heat shock-sensitive mutants. As we noted above, both DHC and LRR5 are dispensable for growth under ideal culture conditions but are essential for survival of febrile temperatures. Similar to the analysis of ART-R field isolates to identify changes associated with resistance, our RNAseq analysis of these two isogenic *piggyBac* mutants enables elucidation of genome-wide functional association networks linked to the heat-shock phenotype. Since these are single insertion mutants isogenic with the wild-type parent, we can conclude with some confidence that the observed essentiality of LRR5 and DHC is due to the functions of these individual protein products. Furthermore, by examining the functional overlap of two distinct types of processes, which are both uniquely essential for heat shock survival, we can define some of the core set functional properties, or mechanisms, the parasite uses to survive the toxic effects of elevated temperature.

- Importantly, we also expected and did observe a significant correlation in the genes dysregulated in these mutants and the types of genes mutated in the mutants of the 1K library. The consistency of these results enhances the veracity of conclusions drawn from our study.
- We modified the title of Figure 3 to better convey the conclusions enabled/corroborated by the comparative mutant transcriptional profiling to “Unfolded protein response, apicoplast-targeted and mitochondria-targeted stress-response pathways are critically dysregulated in functionally unrelated HS-Sensitive mutant clones.”
- We have included an additional summary table in the supplemental Tables S3 to give exact number of genes and expression fold change in six transcript-expression categories. Also, a summary table is added into Tables S3.
- We added a summary-figure (Figure 5F) to highlight the network of major processes driving the HS-response suggested by our data, as well as to suggest a systems-level model integrating our data with several other studies that have advanced the field’s understanding of component-pathways driving artemisinin-response.

Minor points

1. *Figure 5F superimposes a model for survival in Heat shock that doesn’t really draw much from the data from this study. The key genes identified through the RNAseq experiments and the piggybac screen highlight genes in apicoplast isoprenoid synthesis, then deployment of those isoprenoids in prenylated proteins that are involved in protein trafficking. In contrast, figure 5F presents a model centred around ubiquitination – which isn’t really consistent with the data produced in this paper. Indeed the more recent data on the connection between K13 and ART-R is much more consistent with a role for K13 in trafficking than in bulk turnover of proteins through Ubiquitination. I would be much more interested in seeing a how the authors think these data fit into an integrated model of protein trafficking.*

Response

- We replaced the UPS-focused summary-figure with a systems-level model of the multiple prongs of the HS/ART-R response suggested by our data (Figure 5F), as well as integrating our data with several other studies that have advanced the field’s understanding of component-pathways driving artemisinin-response.

2 *Figure 4B could be easier to understand for a reader if it had a short label above the panel explaining if the high mutagenesis index correlates with more essential or less essential.*

Response

- We appreciate reviewer’s suggestion. We have included a label of the gene’s essentiality on the top of the figure 4B, which shows apicoplast-targeted genes tend to be highly essential during blood-stage vs. all other non-apicoplast-targeted genes detected above threshold in RNAseq. Lower Mutagenesis Index Score (MIS) indicating higher essentiality, the median Mutagenesis Index Score (MIS; [6]) for apicoplast-targeted genes is much lower than median MIS for all other genes (**** Wilcoxon-test p-value < 1e-15).
- We added a new table S4 providing expression data and GO-terms mapping to the apicoplast-targeted genes differentially expressed in HS-response used in Fig. 4A-C.

3. *Some of the genes in figure 5 have multiple genes that correspond to that gene name. All need the appropriate PlasmoDB id in the figure legend. The PPT and TPT in figure 5 are drawn as transporters in the same membrane potentially importing different substrates – guessing from the names they are given, these names are presumably referring to the proteins we know now to be the transporters of the inner and outer membranes respectively (and presumably sequentially transport the same metabolites through different*

steps), and this should be redrawn

Response

- We modified the figure 4D-E legends to include the appropriate gene IDs available from *PlasmoDB*.

4. The focus of the title on the “ancient endosymbiotic origins “ and the line “the most ancient genes in the parasite’s genome” are not useful foci – all of the genes in the parasite genome can be considered to realistically to be the same age, with the possible exception of those genes gained by lateral gene transfer, which are more newly arrived in the genome. It isn’t the ancientness of the endosymbiont that generates this link – instead it is the newer, eukaryotic process (prenylation) that is central to this. I suggest rewording the title and removing the reference to the “most ancient” genes.

Response

- The title was revised to “The apicoplast link to fever-survival and artemisinin-resistance in the malaria parasite”

5 The authors describe genes in the abstract and at line 111 as “apicoplast-associated” this needs to be more specific. Do they mean genes encoded in the apicoplast or (nuclear) genes for apicoplast targeted proteins? – I suspect in both cases it’s the latter, and the authors could make this more precise.

Response

- All *piggyBac* mutants are in nuclear-encoded genes, and these lines in the manuscript refer to nuclear genes targeted to the apicoplast. We have therefore corrected “apicoplast-associated” to “apicoplast-targeted” throughout the manuscript. We clarify that our later reference to “apicoplast-related” genes (e.g., Figures 4D – 4E) refers to apicoplast-targeted genes, as well as genes encoding the direct downstream-targets of apicoplast prenylation (such as the Rab-family vesicular trafficking proteins).

Reviewer #2 (Remarks to the Author):

The submission makes use of a panel of previously published insertion mutants in the human malaria parasite Plasmodium falciparum to screen for genes with important roles in parasite survival within febrile hosts. A major conclusion is that isoprenoid precursors synthesized by the apicoplast and used to prenylate proteins are an essential capacity for survival of heat shock.

Response

- Thank you for summarizing one of key findings of our study. Apicoplast metabolism is indeed important for the parasite survival under stress conditions such as the responses to febrile temperatures and exposure to artemisinin.

A key flaw with the interpretation of the results, which is encapsulated in the title, is the claim that the acquisition of the apicoplast “enabled parasite-survival of extreme temperatures”.

This is not a reasonable conclusion. The apicoplast existed long before homeotherms arose. It is clear that the common ancestor of dinoflagellates and Phylum Apicomplexa had already acquired a plastid [1]. How old is that plastid/apicoplast-containing common ancestor? We have no dates for the origin of Apicomplexa, but there are robust fossil records for the sister lineage dinoflagellates. For instance, a recent analysis demonstrates that dinoflagellates unequivocally go as far back as the Triassic [2], and depending on whether one considers achritarchs to be dinoflagellates, the group could go as far back as the Proterozoic [3].

Therefore, the apicoplast is very ancient and undoubtedly predates the origin of the febrile response by mammals. Indeed, it possibly predates animals.

Response

- Use of “acquired” was a poor word choice on our part and we did not mean to imply endosymbiosis of the algal ancestor and the emergence of *Plasmodium* parasitism were concurrent. Indeed, our understanding is similar to what the reviewer states and we agree with the reviewer, and with the prevailing body of literature, that the apicomplexan plastid is an ancient organelle and predates animals and the febrile response. The ancestral apicomplexan had already inherited/acquired the unique metabolic pathways from the endosymbiont and because of this then the ancestral malaria parasites had a survival mechanism enabling them to adapt or at least tolerate the extreme febrile temperatures that are lethal to most mammalian cells.
- We modified the title as already noted and replaced “acquired” with “presence” in our final paragraph before the conclusion: “It is tempting to speculate that presence of the red-algal endosymbiont and its associated plant stress-response mechanisms is what enabled the ancestral parasite to survive host-fever...”. [lines 649-650]
- Inclusion of the published research about apicoplast evolutions were also included in order to reinforce this more parsimonious perspective on the evolutionary history of the apicoplast endosymbiosis.

What the data do show is that protein prenylation is essential for surviving heat shock. Apicomplexa happen to use isopentenyl diphosphate made by the apicoplast to build prenyl chains to attach to proteins, but all eukaryotes make prenyl chains by one means or another, so the apicoplast is not a 'solution' to a parasite surviving febrile defence by a host.

Response

- We agree that protein prenylation is a ubiquitous eukaryotic post-translational protein modification process. However, the enormous functional diversity of prenylated proteins gives rise to a wide range of roles in cell signaling, homeostasis and coping with stress. Our study showed the important aspect of isopentenyl diphosphate in the parasite for dealing with cyclical febrile stress encountered during the malaria paroxysm.
- We are not suggesting that parasites invented prenylation to survive heat stress, particularly not to the exclusion of other eukaryotes. As the reviewer has noted, however, we and others have demonstrated that prenylation is critical for parasite survival of febrile temperatures—and phyletic analyses have long suggested the pathway through which these parasites produce isoprenoid precursors necessary for that prenylation was acquired via the secondary endosymbiotic event giving rise to the apicoplast.
- While all eukaryotes do employ prenylation—plants/algae, bacteria, and apicomplexan parasites are the only ones that do so via the non-mevalonate isoprenoid biosynthesis pathway. Stress responses, including those induced in response to heat stress, have been thoroughly studied in these lineages. It is also notable that prenylation (which includes geranyl-geranylation as well as farnesylation) in *Plasmodium* during the blood stage is restricted to an uncharacteristically small list of proteins (~15 to 20), the vast majority of which are members of the Rab-family proteins with key roles in vesicular trafficking. The lone farnesylated protein during the parasite blood-stage is HSP40—and incidentally, this farnesylation of HSP40 is absolutely critical for parasite survival of thermal shock (see Mathews et al., citation).

The above considerations render Fig 4F (heat shock genes shared with plants and algae) not relevant.

- This figure is intended for the broader readership of the journal and we think it is a useful way to highlight the blended ancestry of the apicoplast metabolic pathways important in the fever stress response. Our

results from the forward genetic screen showed that the genetic basis of heat shock response specifically involves distinct GO processes, such as those in isoprenoid biosynthesis and thiamine metabolism. And these pathways are of plant/algae origin, in contrast to other pathways. We highlight the 'green' origin of these pathways because it is absent in the mammalian cells and offers therapeutic targets for future interventions. Already DXS is considered a potential druggable target and our study enhances its value.

smaller issues:

The manuscript needs to make clear whether or not these are loss of function mutants or knockdowns. Several of the genes are considered essential in blood stage growth, so I presume the insertions don't totally abrogate expression.

Response

- We agree with the reviewer that the functional impact of the transposon insertion is important to know. One of the primary reasons for the differential expression analyses used the LRR5 and DHC mutants is the disruption of their ORFs results in a loss of function and the resulting interpretation is more straightforward, which is noted in the manuscript. However, the potential impacts of the transposon insertions go beyond just loss of function or knockdown. Our previous profiling studies of a K13 mutant, in which the transposon integrated into the promoter region, demonstrated that even a small perturbation of the wild-type expression pattern of a gene is sufficient to link a gene to a phenotype. This is fortunate since most mutants created by *piggyBac* mutagenesis in *P. falciparum* are not loss of function, since most genes in the genome are essential for asexual growth. The supplemental table S1 and S2 provide data on the location of every insertion of the mutants used in this study and this will enable readers to interpret the results for their genes of interest. And we have included additional Tables in Table S1 and Table S2, which provide the distance to gene, gene IDs, gene names and functions for the neighboring genes on both sides of *piggy-bac* insertion for pilot-library and 1K-library *pB* insertions.

line 48 running title malaria is a disease not an organism

Response

- We've revised the running title to "Plastid metabolism enables malaria parasites to survive fever and artemisinin".

line 45 ".....responses depend on some of the most ancient genes in the parasite's genome"

This statement is fraught. Do the authors mean the absolute antiquity of the genes (i.e. how old is a given gene?), or how long the genes have been in the parasite genome? Given that the apicoplast/plastid was acquired by secondary endosymbiosis, the bulk of apicoplast related genes were acquired by the common ancestor of Apicomplexa and dinoflagellates at the time of the secondary endosymbiotic event. This ancestral, pre-secondary endosymbiosis organism already had a full complement of genes in its nucleus and mitochondrion. Those genes are therefore 'older' to the lineage than the genes picked up during secondary endosymbiosis of a plastid containing organism. However, from a broader perspective, one could argue that the plastid related genes, which ultimately derive from a primary endosymbiotic cyanobacterium, are indeed amongst the oldest of all genes since cyanobacteria are the oldest known organisms, as represented by stromatolite microfossils. Categorising genes as 'ancient' or 'recent' is not simple.

Response

- Again, we concede our choice of words needs refinement to better convey the complicated evolutionary history of the apicoplast-targeted genes involved in the febrile response. Indeed, categorizing genes as recent vs ancient is not simple, although we did attempt to address the complex nature of this

evolutionary history in the discussion already. We agree the point here is that endosymbiotic cyanobacterium-related genes associated with these apicoplast metabolic pathways are among the most ancient. We have revised the origin statement by putting it more into an evolutionary context.

- The abstract Conclusion is revised to: “*Plasmodium falciparum* parasites appear to exploit their innate febrile-response mechanisms to mediate resistance to artemisinin. Both responses depend on endosymbiotic cyanobacterium-related ancestral genes in the parasite’s genome, suggesting a link to the evolutionary origins of *Plasmodium* parasites in free-living ancestors”.

ref 45 is wrong. The authors appear to have confused Boucher's other preprint from the same year. In any case, the work is now published in a refereed journal [4](reference #45 in revised ms), and that citation should be used for the apicoplast proteome.

Response

- The reference has been updated.

The authors have focused on protein prenylation as the key role in surviving heat shock, but the isopentenyl diphosphate precursors made by the apicoplast are almost certainly used to make dolichols for glycosylation of proteins and ubiquinone tails for mitochondrial electron transport. While it is apparent from the phenotype data that protein prenylation is crucial for heat shock survival, did the authors tease out any requirement for dolichol-related processes or mitochondrial electron transport?

Response

- We agree with the reviewer that the isoprenyl diphosphate precursors synthesized by the apicoplast are used in multiple pathways. The key enzyme 1-deoxy-D-xylulose 5-phosphate synthase (DXS) is at a metabolic branch-point to supply essential vitamins and isoprenoids. Regarding stress-resilience, there could indeed be involvement in other pathways such as ubiquinone synthesis, vitamin K production, and tRNA prenylation. However, the role of many of these pathways in malarial blood-stage infection remain under investigation and this study perhaps serves as a gateway to research other pathways branching off these crucial apicoplast-generated precursor-molecules.

Reviewer #3 (Remarks to the Author):

This is a potentially interesting manuscript that seeks to identify new genes involved in the Plasmodium falciparum febrile heat shock response and suggests a tie to artemisinin resistance in malaria parasites. The idea sounds compelling and timely and the authors have used a powerful piggybac transposon system that they have developed—the authors’ previous study on gene essentiality using the piggybac system shows the system is robust and has provided useful data to the community. In theory, the authors’ collection of transposon mutants could be a powerful toolbox for functional annotation of the Plasmodium genome.

Response

- We thank the reviewer for careful evaluation and helpful comments. We really appreciate the reviewer’s effort for a thorough review of our results and methods for the complex study. The comments have been taken into account to improve the manuscript’s clarity – below we reply to each comment separately.

However, as presented, the work is very difficult to evaluate. First, as written it appears to be quite a bit of “cherry-picking” going on in terms of piggybac library creation and validation.

Response

- We appreciate the comment regarding the *piggyBac* library creation and validation and understand the concern we were selective in creation of the library screened. An important aspect of our approach is the random nature of *piggyBac* mutagenesis and consequently the collection of mutants screened. While we are in the process of creating libraries of selected mutants, that is not the case for the libraries used in the heat shock screens.
- The *piggyBac* transposon preferentially inserts at the tetranucleotide target sequence ‘TTAA’ and *piggyBac* transposase mediates this process. Our methods use the native transposase under control of *Plasmodium* regulatory elements and each transposition has nearly equal probability to utilize any TTAA in the genome. The *P. falciparum* genome has a large number of evenly distributed target sequence ‘TTAA’ sites (>328,000 ‘TTAA’, averaging one site per 70 bp over both coding and noncoding regions). Our methods are optimized to create one insertion per organism and mutants created in each transfection are unique and pre-selection of specific mutants as suggested by the reviewer is not possible.
- Comment on the origins of the pilot and 1K libraries. Since our early efforts to develop *piggyBac* mutagenesis for *P. falciparum* were not efficient, mutants from each transfection were relatively few and cloned easily. After numerous transfection experiments finally surpassed 100 unique mutants, we aggregated all of these and that is the origin of the pilot library – it literally represented all the mutants created at the time. Of course, it was not possible to scale that approach to large-scale or whole genome screens. Consequently, once transfection was optimized, we began a collaboration with Sanger Institute to develop a more efficient and accurate method to identify inserts (we shared a common problem as the mouse genome *piggyBac*-mutagenesis studies suffered from the same limitation). QIseq analysis was then developed over several years of collaborative effort and this allowed subsequent libraries to be created that could be left as large mixed mutant pools, which led to creation of the 1K library. As described in the supplemental figure S1, the design of the phenotypic screening pipeline is: 1) to test the protocols using individual pB-mutant clones; 2) to develop a reliable pooled phenotypic screening method with pilot-library aliquots; 3) then we scale-up the phenotypic screen using 1K-library.
- In the current study, we used the pilot-library created from the early clones to establish and validate the methods for the heat shock screen. Although the pilot-library is small, it has been extremely informative to give a ‘thumbnail’ view of the genome and is infinitely reproducible from the original clones. Periodic genetic analysis/sequencing insures the authentication of these clones. Large batches of up to 100 aliquots can be generated at a time, with each aliquot a nearly identical collection of 128 pB mutant clones, to provide enough biological replicate samples for experimental parallel phenotypic screens as used in this study. While this may seem trivial, the approach represents a major breakthrough in experimental genetic studies of *P. falciparum* – an organism well-known for its refractoriness to genetic manipulation. In the methods, we described how to grow 128 mutants individually and then pool together an equal amount of each. Once prepared we make as many cryopreserved aliquots as possible. This collection of mutant clones used for the pilot library are available to the community through MR4/BEI Resources.
- In contrast, creating biological replicates of the 1K-library is not really feasible as it is created from mixing 12 un-cloned mixed population pools (estimate ~100 unique mutants per pool). Each time the smaller pools are thawed and put in culture to amplify, the relative abundance of each mutant changes due to the differences in inherent growth rates. As noted above, the pB insertions in each mixed population pool comprising the 1K-library were generated randomly via transfection. We did not know the insertion sites in each pool until we completed the 1K-library screens, using QIseq to identify each insertion site.
- To demonstrate the random nature of the 1K library, we added new comparative analysis of the *piggyBac* mutants’ distribution patterns in coding vs noncoding regions between the pilot-library and 1K-library in the revised manuscript. We show the distribution patterns between intergenic regions and CDS are almost equal and also reflect the distribution of the saturation mutagenesis-library as a whole (below, new figure S6A).

- Previous publications (references cited #6 & #13; PMID:19422698, PMID:27197223) established the unbiased nature of *piggyBac* mutagenesis for large-scale mutant screen, the randomness of *piggyBac* insertion, and the accuracy and sensitivity of QIseq.

- We compared the *pB*-mutants' distribution patterns for HS-Sensitive and HS-Tolerant mutants in the 1K-library, finding HS-Tolerant mutants were more likely associated with dispensable genes (exonic insertions) than HS-Sensitive genes (left below, new figure S6B).
- To further demonstrate the randomness of *piggyBac* system, the distance between each pair of neighboring *piggyBac* insertions in our 1K library was calculated. Compared with random insertion-sites generated by sampling, the *piggyBac* insertions of the 1K library did not show any difference (p -value = 0.787, Mann-Whitney U test). The sampling sites were randomly selected to follow the same number of *piggyBac* insertions in each chromosome. The sampling procedure was repeated 100 times (right below, new figure S6C).

HS-Sensitive and HS-Tolerant *pB*-mutants' distribution patterns in 1K-library

In addition, the supplemental data is not well annotated or not provided in a way that allows one to cross check the authors' work. Instead of providing exact numbers, the authors often make vague statements about tendencies which cannot be readily evaluated. In addition, there seem to be so many items that are poorly labeled or missing in the supplements and figure legends that this reviewer feels that this manuscript cannot be effectively reviewed as is.

Response

- We apologize for missing some labels and exact numbers in the supplemental data, which have been corrected in the revised version. All the items in the supplements were double-checked, and we added additional columns and tables, provided exact numbers and statistical analysis for more precise description in the revised version. We listed the specific items that we added and revised in following response.

Finally, the authors should provide either A. better transparency and statistical support for their claims, especially for the piggybac library screens B. orthogonal validation (e.g. testing claims using independent gene disruptions, possibly created using CRISPR-Cas9 or C. more biological replicates (e.g. a second, independent biorep for the heat shock experiment described in Table S2). Obviously a great deal of careful work went into the study and it would be great to see this published, so it is a shame that it can't be given the evaluation it deserves because of presentation issues.

Response

We thank the reviewer for providing many detailed suggestions to strengthen the rigor of our data and the analyses. In response, we have added statistical data as suggested. We also discuss the validation of our assay under the following points.

In response to “A”. To improve clarity and statistical support:

- we rewrote the legend of figure S5 - QIseq data-correlations within and between Pilot-Library Screens;
- modified figure S6 - Randomness of *pB*-mutants' insertion-site distribution in the pilot-library and 1k-Library
- added two additional supplemental figures S7 and S8 to check reproducibility in the 1K-library HS-Screen and consistency of fold change comparing pilot-library and 1K-library HS-Screen.

In response to “B”. Orthologous validation:

- In our first HS screen study of individual mutant clones published in *mSphere*, a phenotype rescue experiment was included (C9/mkp1 rescue) and the current study added a genetic complementation/phenotype rescue of FIKK9.3.
- All data from the pilot library screens presented for this study were based on at least 2 biological replicates. Furthermore, each of these replicates for the pilot screens included 3 technical replicates at 41C and 2 technical replicates for the control-parasites grown in parallel at 37C.
- We've provided an additional detailed evaluation of phenotype-reproducibility utilizing performance of distinct mutants of the same gene in pooled screening in this revision (see below).

Major points:

Additional mutations that randomly arise during long term culture may affect the authors' conclusions about phenotypes. The authors should ideally use either a second independent mutant or use edited lines to check that the phenotypes are the same for their heat shock sensitive mutants (DHC and the LRR mutants)

As a genomics paper there are some statistical issues. First, the authors have done two different screens, a pilot, with a small number of genes (~200 mutants) and multiple bioreps, and a larger screen (~1000 mutants) with no bioreps.

Response

- This is a very good point. We addressed this concern in the published data as part of developing the QIseq method (references cited #13; Genome Res. 2016 Jul;26(7):980-9. doi: 10.1101/gr.200279.115; PMID:27197223). Initial tests of this pooled screening approach using the pilot-library, published data

showed that growth rates of individual mutants were highly reproducible between biological replicates, and even between pools with different compositions. All of those 128 extensively characterized *P. falciparum* *pB*-mutant clones in the pilot-library were repeatedly confirmed in subsequent growth screens in 12 asexual intra-erythrocytic development cycles (24 days), biological replicate samples were collected in subsequent cycles at 3, 6, 9 and 12.

- While developing the QIseq method, we also investigated accuracy and sensitivity in order to set cut-offs for insertion site identification. A series of *P. falciparum* lines were used for single insertion validation and whole-genome sequencing (WGS), including NF54-parent-strain and 29 *piggyBac* mutants previously cloned and insertion sites identified by QIseq. Transposon insertion sites were mapped using our NF54 reference genome. WGS of the NF54-parent-strain and 29 *pB* mutants completed at Sanger Institute confirmed the lack of significant genomic differences in long term culture and only a single *pB* insertion in each *pB* mutant line was identified (WGS data unpublished).
- We revised some of the writing and clarified that two biological replicates were performed in pilot-library screens including heat shock, DHA, AS, BTZ and Oxidative phenotypic screens, statistical analysis has been provided in the supplemental files. As noted above, the pilot-library is periodically prepared batchwise from separate cryopreserved aliquots of 128 mutant clones. After short term culture each clone is combined and mixed thoroughly to create the library followed cryopreservation ~100 aliquots as needed. *PB* mutants' distribution appears identical in the aliquots used for the biological replicates in the parallel phenotypic screens of the pilot-library.
- For the 1K-library screens, we used 12 uncloned large mixed population pools (~100 unique mutants per pool representing different QIseq libraries). Insertion sites in each pool were unknown until the 1K-library screen was completed. Biological replicates of the 1K-library are not feasible as it was created from uncloned mixed population pools. Nonetheless, in the 1K-library we identified 15 *piggyBac* mutants that appeared multiple times (i.e., at least twice in 2 different pools). We used these mutant replicates to check reproducibility and validate the performance of the 1K-library by comparing correlation of these 15 *piggyBac* mutants in different pools. Below, the new figure S7 indicates the distribution of fold changes in HS-Screen (FC-HS) for 15 mutants replicated in different QIseq pools of the 1K-library. The FC-HS of 15 repeated mutants are represented as a color-coded dot and the shape of each dot indicates the type of the mutant of *piggyBac* insertion location category (e.g. the blue square, mutant PfNF54_08_m1::1374031, gene PF3D7_0831800, HRPII, appeared in 4 pools (1, 2, 6, and pool 12), these repeated mutants showing similar FC-HS in different pools in 1K-library.

- To further quantify the consistency, we checked the correlation of FC-HS for these repeated mutants between two (or 4) pools, the Pearson correlation is 0.806.

The Pearson correlation of 15 *piggyBac* mutants appear multiple times in 1K-library (cor = 0.806).

- We also checked consistency of fold change comparing the pilot-library and 1K-library (new figure S8, below). Some genes/gene-families have several distinct mutants appearing in the pilot-library and multiple large mixed-population pools of the 1K-library, such as dynein heavy chain (DHC) and FIKK-family genes, with FC values across pools in close agreement.

PB Pool ID	PB Mutant ID	Insertion Site	Distance to gene	Type of Insertion	GeneID	Gene Function	Fold Change (41C/37C)
Pilot-library	PB4_DHC_11	PfNf54_11_m1::879695	0	exon	PF3D7_1122900	dynein heavy chain, putative	0.21
1K-library-Pool_14	14-016_DHC_12B	PfNf54_12_m1::135712	0	exon	PF3D7_1202300	dynein heavy chain, putative	0.05
1K-library-Pool_4	4-054_DHC_10	PfNf54_10_m1::980010	0	exon	PF3D7_1023100	dynein heavy chain, putative	0.28
1K-library-Pool_14	14-015_DHC_12A	PfNf54_12_m1::121344	0	exon	PF3D7_1202300	dynein heavy chain, putative	0.32
Pilot-library	PB-54_FIKK9.3	PfNf54_09_m1::99320	0	exon	PF3D7_0902200	FIKK family (FIKK9.3)	0.52
1K-library-Pool_4	4-043_FIKK9.1	PfNf54_09_m1::92390	1095	intergenic	PF3D7_0902000	FIKK family (FIKK9.1)	0.23
1K-library-Pool_13	13-028_FIKK9.2	PfNf54_09_m1::93353	-477	3UTRup	PF3D7_0902100	FIKK family (FIKK9.2)	0.41
1K-library-Pool_12	12-040_ETRAMP	PfNf54_10_m1::82063	324	5UTRup	PF3D7_1001500	early transcribed membrane protein 10.1	0.310
1K-library-Pool_1	1-055_ETRAMP	PfNf54_10_m1::82163	424	5UTRup	PF3D7_1001500	early transcribed membrane protein 10.1	0.470
1K-library-Pool_4	4-048_HAD1	PfNf54_10_m1::1339529	-238	5UTRup	PF3D7_1033400	haloacid dehalogenase-like hydrolase (HAD1)	0.320
1K-library-Pool_12	12-037_HAD1	PfNf54_10_m1::1340834	201	3UTRup	PF3D7_1033400	haloacid dehalogenase-like hydrolase (HAD1)	0.410

Mutants in several genes/gene-families (above) were consistently identified as HS-Sensitive in the pilot-library as well as across multiple pools of the 1K-library.

Fig. S8. FC-HS of insertional mutants in genes represented in both the pilot library and the 1K-library (n = 16 genes; colored points) are highly correlated (Pearson correlation = 0.702). Insertion coordinate in the pilot library is indicated on the left of the '|', while insertion coordinate of the mutant in the same gene in the 1K-library is to the right. Distance between the pilot-library insertion and the 1K-library insertion is indicated by shape (maximum distance = 1kb).

The authors then choose to validate several genes using RNASeq. However, the mutants that they choose to validate come from the small pilot screen and were previously known, and not the larger screen (Table S2) so you can't tell how good the larger screen actually was. Is this data useable or just too noisy? It seems many repetitions may be needed to confidently call sensitivity? Are the libraries really random, or were they constructed to be enriched for mutants in genes that authors find interesting? For example, *kelch13*?

Response

- The response to the concern on the selection of the LRR5 and DHC mutants is addressed above in response to the comments of reviewer 1 (pages 3-4), and details have been added to the revised manuscript (lines 247 – 253). As noted in the figure S8 at the top of this page, the different DHC mutants in the 1K library had similar phenotypes as the cloned DHC mutant analyzed from the pilot library. The exception is for the 1K mutant with the insertion nearer the end of its large ORF of the PF3D7_1202300 DHC.
- We have included an additional supplemental figure S12 (“QIseq data-correlations within and between Pilot-Library phenotypic screens”) to show that the pilot-library screens were highly reproducible between biological replicates in all phenotypic screens, and even between pools with different compositions. We previously cloned 128 mutants before we generated the pilot-library aliquots, and we characterized these mutants by QIseq in pooled screens. We also previously performed individual phenotypic assays for 20 of the pB-mutant clones, which served as benchmarks in the pilot-library screen for defining HS-Sensitive and HS-Tolerant mutants.
- *Can we tell how good the 1K screen actually was?* Yes, we found 15 *piggyBac* mutants appeared multiple times (at least appear in 2 pools) in 1K-library, correlation of these mutants >0.8 (Fig. S7). And we compared pilot-library and 1K-library for those mutants relatively close to each other, correlation is >0.7 (Fig. S8). We use same fold change cut-off to identify the mutants as HS-Sensitive or HS-Tolerant. Three pB-mutants carry a single insertion in the exon associated with dynein heavy chain (DHC) genes were consistently identified as HS-Sensitive in multiple pools in 1K-library. Which proved the consistency and robustness of pooled screens using 1K-library.
- *Are the libraries really random?* Yes. Previously (BMC Microbiol. 2009 May 7;9:83. doi: 10.1186/1471-2180-9-83, PMID: 19422698), we performed *piggyBac* mutant screen and highlighted in the analysis the randomness of *piggyBac* insertions. We subsequently confirmed this randomness along with the accuracy and sensitivity of QIseq (Genome Res. 2016 Jul;26(7):980-9. doi: 10.1101/gr.200279.115; PMID:27197223). In response to the reviewers’ concerns in our current study, we checked the pB-mutants’ distribution patterns in coding vs noncoding regions for pilot-library and 1K-library compared with whole-genome saturation-mutagenesis library. The distribution patterns between intergenic regions and CDS are almost equal between the pilot-library, 1K-library and saturation-library. These data show that the distance of *piggyBac* insertions in the 1K-library are no different compared with random chromosomal sites generated by sampling (Fig. S6).
- The generation of a K13 mutant with an insertion in the promoter region occurred in one of our early transfections and predated its identification as a resistance marker for artemisinin resistance. The randomness and luckiness of creating this mutant is evident from the results of the saturation mutagenesis (with 38,000 new mutants) that failed to produce another mutant with an insertion any closer to the transcriptional unit of *pfkelch13*.

*Some parts of the manuscript are misleading and much of the work/validation is done with piggybac transposon mutants whose phenotypes were not actually discovered in this library screen (Pb4 and P31). The authors write “We chose two individual HS-Sensitive mutant clones for additional profiling via RNAseq to identify dysregulated genes responsible for this sensitivity. Both mutant lines have a single disruption in the coding region of a gene not previously implicated in the HS-response: ΔDHC (dynein heavy-chain gene PF3D7_1122900[pb4]), and ΔLRR5 (leucine-rich repeat protein PF3D7_1432400 [pb31]). Based on the positioning of the paragraph it makes it seem like this came from the larger library screen. However, these genes are only listed in the pilot screen, not the whole library screen. In addition, it says in the notes that these were actually controls that were added in. The authors write in supplemental table 1 “*Known HS-Sensitive’ pB-mutant clones (PB31, PB14, PB3, PB4, PB2 and PB5) were validated by individual heat shock assay (Thomas et. al., 2016)”*

Response

- As noted above in our response to reviewer 1, we chose the DHC and LRR5 heat shock sensitive mutants for phenotypic transcriptional profiling by RNAseq because they were well validated in both the screen of individual clones and as mutants within a pooled screen. Both mutants have disrupted ORFs and are representative of the major GO processes critical in the survival response to heat shock. Since the types of forward genetic screens are novel for *P. falciparum* experimental research, the inclusion of the subset of mutants analyzed as clones in the previous small study was intentional to provide improved rigor for the new pooled screen approach. In addition, it also important to note that the results of the first heat shock screen were determined by Giemsa-stained blood smears of individual mutant clones, which has severe limits on scalability, throughput, and accuracy.
- We previously performed individual phenotypic assays for part of the *pB*-mutant clones to test the heat shock protocol, these mutants served as benchmarks in the pilot-library screen and assisted in determining cut-offs to demark sensitive/tolerant mutants. We used same cut-off lines for pilot-library and 1K-library screens. In our revision, we clarified that we chose DHC and LRR5 heat shock sensitive mutants mainly for RNAseq analysis based on the results in pilot-library screens.

Fold change of parasitemia under 41 °C vs. 37 °C
(Parasitemia was determined via flow cytometry)

This figure shows the pooled pilot-library HS-Screen results vis QIseq comparison with previous published individual HS assay results which used flow cytometry (PMID: 27830190). 20 *piggyBac* mutants are available in pilot-library. After group them based on lower (red dots) or higher (green dots) than the median value of heat response (x-axis), we observed consistent results in pooled pilot-library HS-Screen (** indicates p-value < 0.01 Mann-Whitney U test).

- An important advantage of the pilot library is the availability of the clones for follow-on analyses, whereas the 1K mutant library is comprised of uncloned mutants. Cloning is a slow, resource-intensive process not scalable to a whole-genome level.

Minor points:

Many abbreviations are used and sometimes it is not clear what things are. This makes it hard to review. Add a table of abbreviations?

Response

- We thank the reviewer for the helpful suggestions. We revised the manuscript to cut down on abbreviations. In revision, we added a list of abbreviations as table S8.

The authors often make vague unsubstantiated statements that are very difficult to review. They write “A high proportion of essential genes that are upregulated in response to heat stress are targeted to the apicoplast.” I can’t evaluate this statement. What is high? It would be much easier for me to understand if the authors wrote something like “Of the 45 essential genes that are upregulated in response to heat stress (FC > 2), 7 are targeted to the apicoplast. Given that there are only 12 genes that are targeted to the apicoplast and which were detected in our RNAseq analysis, this enrichment is very significant (p value =) or something like this... The word “tend” is used often, often without statistical support.

Response

- We added new table S4 providing expression data and GO-terms mapping to the apicoplast-targeted genes differentially expressed in HS-response used in Fig. 4A-C.
 - Table S4A, apicoplast-targeted genes regulated in response to heatshock (related to Fig. 4A-B)
 - Table S4B, GO-terms mapping to HS-regulated apicoplast-targeted genes (related to Fig. 4C)

Supplemental tables. Some descriptions of the table calculations are found in methods but would be much nicer for the reviewer if this was placed next to the actual data as part of a legend. For example, "We defined expressed genes as those having FPKM > 20 for at least one biological replicate at either 37°C or 41°C. The fold change of normalized gene expression between 41°C and 37°C was calculated for every biological replicate." Would be more helpful as legend to supplemental table S3 tab 2. For the GO analysis, what does "annotated" mean? What does "significant" mean. For annotated do we mean number of genes with this annotation in the genome, or in Table S3? For significant, is this based on expression levels and upregulation and a statistical test?

Response

- The legends for supplemental tables S3A-D were provided in lines 986 – 1000 of the initial submitted manuscript and are copied below for convenience. We have added a "ReadMe" tab containing legends to the table-files to ease review.
- Table S3. Comparative RNAseq-results between NF54 and HS-Sensitive mutant clones Δ LRR5 and Δ DHC in response to heat-shock.
 - S3A. All genes classified into HS response-categories in NF54 with or without exposure to heat-shock using RNAseq data (n = 2567). HS-classifications for each gene in two HS-Sensitive mutant-lines are indicated where available. Criteria for inclusion: NF54 expression above threshold (FPKM > or = 20 for at least one replicate in at least one temperature-condition) and FC-HS supported by two biological replicates.
 - S3B. Genes included in functional enrichment-analyses. Criteria for inclusion: all genes with expression above threshold AND agreement between replicates as to HS fold995 change classification for all three parasite lines (n = 1298).
 - S3C. Enriched GO-terms for specified HS-response-categories as included in Figure 3B. "Annotated": the number of genes annotated to a given GO-term included in the analysis for all HS response-categories. "Significant": the number of genes annotated to a given GO-term in the HS response-category being tested for enrichment.
 - S3D. Full functional enrichment-results for all HS response-categories.

There is no supplemental file to support Figure 4 either—you just need to trust the authors. While maybe one could go through and redo their analyses (e.g. looking up individual genes and reading other papers that are referenced) using what they have provided this would require quite a bit of reviewer time. Most good genomic papers provide more intermediate files allowing the authors' work to be checked. What genes are considered the "downstream targets."

Response

- All common gene-symbols for genes involved in isoprenoid biosynthesis and immediate downstream targets of prenylation are labeled in Figure 4 (as are all genes evaluated in categories of interest provided in Supplemental Figures S3A-S3C). Symbols for all these genes where available are provided with the respective gene ID in Table S2.
- We have included additional Tables S4C and S4D explicitly highlighting genes from categories of interest and referencing the publication from which they were identified, as well as the figures of this manuscript featuring those data.

There are no common names for genes listed in supplemental table 4. Also, incomplete column headings. What does the asterisk mean? Also, again data would be easier to interpret if the authors included legends in the tables.

Response

- We’ve moved the legends from the end of the text-file/below the table in the initial submission to a “ReadMe” tab in the table-files for this revision to ease review. We have added both gene product and gene name or symbol where available.

In supplemental files the authors have included mutants with transposons in intergenic regions but they only list one gene that might be affected—typically an intergenic mutation can affect two genes? This issue should be addressed.

Response

- We have included additional Tables S1C and Tables S2C providing the gene IDs, gene names and functions for the neighboring genes on both sides of *piggy-bac* insertion for pilot-library and 1K-library *pB* insertions.
- We added discussion concerning the possibility of *piggyBac* insertions in intergenic regions disrupting the function of neighboring genes.

A table listing the number of different *pb* mutants and the types should be provided for the *piggybac* libraries that are screened. How often do the authors actually get phenotypes for mutants that are classified as intergenic?

Response

- We have included an additional supplemental figure S6 showing *pB*-mutants’ insertion distribution in the pilot-library and 1K-library, as well as distance between neighboring *piggyBac* insertions in the 1K-library.
- The fraction of insertions in intergenic regions is 75.7% in the pilot-library and 75.0% in the 1K-library. To check the fraction of intergenic mutants exhibiting phenotypes, we picked top and bottom 20% of *pB*-mutants for each assay by comparing the growth fold change. The fraction of intergenic *pB*-mutants is shown in black below.

Some table summarizing the quality of the RNAseq would also be useful. How many reads? How many genes detected?

Response

- We have added a summary-table for genes detected above threshold for each comparative RNAseq analysis in the “ReadMe” tab of Table S3.

Supplemental files. What do the negative distances mean? I assume this is number of bases upstream of the start?

Response

- In our original manuscript, the minus distance indicated the insertion had a smaller coordinate than the nearest gene, while positive distance indicated the insertion had a larger coordinate than the nearest gene. In this revision, we have removed the minus notation and all distances are instead represented as absolute values. We have also added the nearest gene in both directions (upstream and downstream) for each insertion (Tables S1C and S2C).

Names of 42 isoprenoid genes and all downstream targets in the entire genome need to be provided somewhere. Likewise, the authors state “We next evaluated our 1K HS-sScreen for direct K13-interacting partner-proteins recently identified via immunoprecipitation [25], and found that mutants in 10 of the 24 unique putative K13-partner-proteins represented in the screen were sensitive to HS.” What are the identities of these 24 genes? Expecting the reviewer to go read the other paper [citation 25] is burdensome. The authors could add a column to their supplemental tables in which they indicate genes that are annotated isoprenoid genes, or k13 partner genes or essential genes.

Response

- Gene IDs or gene-symbols where available are indicated for these analyses in Supplemental Figures S3A – S3C. We’ve added additional tables providing HS pooled-screen phenotype data (Table S4C) and comparative HS RNAseq phenotype-data (Table S4D) considered for each analysis the Reviewer mentions, including the reference-publication where the gene was implicated and the figure in the current study incorporating the data.

The authors write “Measures indicating reproducibility for Qlseq-data for both the pilot-library and 1K-library screens are shown in Fig. S5A-B and Fig. S6, respectively. We observed high correlation between 5’ and 3’ Qlseq-libraries from each pool (Fig. S5A, Fig. S6). Resulting Growth and HS phenotype-assignments were highly reproducible across three technical and two biological replicates (HS-Screen, $R=0.94$; Growth-Screen, $R=0.89$; Fig. S5B).” making it seem like the large library screen has been validated, but S5 just shows the pilot data, and S6 seems is not understandable...there are no axis labels, for example. The figure legend for S6 states “1K-Library screens were as robust and reproducible as the pilot library screens. Qlseq data correlations within and between HS-screens and Growth screens for a representative pool of the 1K-mutant library are shown” which doesn’t exactly describe what is actually plotted.

Response

- We thank the reviewer pointing out the ambiguous description regarding Fig. S5 and S6. In our revision, we have updated the correlation plots between biological replicates of the pilot library at 41C and 37C (Fig. S5 in revised manuscript). We updated the description in the manuscript regarding reproducibility of the pilot screen and updated Fig. S6 to explain the randomness of 1k library generation. We’ve also added Fig. S7 and Fig. S8 supporting the reproducibility of 1K-library phenotypes and the consistency between 1K-library and pilot library results. The manuscript is revised to incorporate these supporting analyses as well.

Methods: The authors should briefly describe how the small pilot library was created and not require the reviewer to go read other manuscripts. Its composition is critical to understanding results.

Response

- We have added the description of how to generate the pilot library in the methods section.

** See Nature Research's author and referees' website at www.nature.com/authors for information about policies, services and author benefits.

REVIEWER COMMENTS

Reviewer #1 (Remarks to the Author):

The authors have addressed all of the points I raised appropriately.

One error remains in the modified figure 4E. The PPT and TPT proteins PF3D7_1218400 and PF3D7_0530200 are drawn as two different apicoplast transporters that import different metabolites, where in fact they are separate transporters in the outer and inner membranes of the apicoplast, which must import the same metabolites (see <https://pubmed.ncbi.nlm.nih.gov/16760253/>). Both PEP and DHAP are presumably imported into the apicoplast, but through concerted action of the two transporters, rather than separately.

Reviewer #3 (Remarks to the Author):

In their revised manuscript the authors present a genomic study in which they perform a small scale screen of *P. falciparum* piggybac mutants to identify those that cannot tolerate "heat shock". The authors then perform an RNAseq expression analysis of two *Plasmodium* piggybac mutants that show enhanced sensitivity to heat shock. The authors have thoughtfully responded to comments and the authors' logic is easier to follow, although this remains rather difficult to review because of the cross-checking that is often needed. The data seems to be robust even if this reviewer might not entirely agree with interpretation.

The manuscript might be more useful to the reader, especially the reader not so familiar with genomics, if the authors present the limitations of their study or discuss alternative interpretations. For example, the authors conclusions about the role of the apicoplast in heat stress are largely built on enrichment analysis, which is a reasonable approach. However, the authors' gene enrichment calculations may not have been corrected for multiple hypothesis testing. Finding 2 of an expected 10 in a set size of X (not sure what X is) may give a p-value of 0.008, but could be found by chance when the entire genome is examined because of multiple hypothesis testing. So, it is not entirely clear if some of the patterns that the authors discuss really are statistically significant. I would recommend that they make this clearer to the reader and provide both corrected and non-corrected p-values in the discussion of the results (if not in figures).

The authors might also consider the alternative explanation for their RNA seq data—that there are minor differences in progression through the cell cycle in the mutants and thus after 3 cycles of heat shock for 8 hours each, the cycles are no longer in sync between the mutants and wildtype and the authors just identify cell-cycle regulated genes. Ribosomal protein genes are under strong cell cycle control and they do show strong differences after one parasite culture has been exposed to heat cycling. There are many ribosomal protein genes so p-values are impressive. Proteosome genes are under cell cycle control. Genes with a role in invasion also change in the authors' study (entry into host, enrichment by chance $p = 1 \times 10^{-23}$). Apicoplast genes are also under cell cycle control but there are fewer, so p-values are less impressive. Such phasing problems also complicate gene expression profiling of synchronized artemisinin resistant mutants as well. It is very difficult to compare gene expression patterns of different mutant lines because of the cell cycle differences that inevitably exist between two different mutant lines. Just changing temperature could hasten the cell cycle if you do it for 24 total hours. This possibility might be discussed.

Minor points:

The manuscript would be more impressive and stronger if the authors added more p-values to the text. What is the probability of finding 10 of 24 mutants mapping to k13 interacting proteins are heat sensitive by chance? This could be just mentioned in the text (no additional figure needed).

The authors could explain that what they call heat shock is not what is classically considered a heat shock. Classical heat shock causes gene expression changes within minutes of changing the temperature in many species—here the authors are studying the effects 2 days afterwards.

The authors should also mention that some of the piggybac insertion events were not in the actual gene that is named in the figure (e.g. in S3), but rather in the intergenic region near the gene.

Some points are still rather difficult to follow.

We examined our 1K mutant-library for representation of isoprenoid biosynthesis, its immediate upstream-regulators (proteins responsible for modulation and import of glycolytic intermediates that serve as pathway substrates), and immediate downstream-effector proteins, and found that all eight isoprenoid biosynthesis-related pB-mutants included in the pooled screen were indeed HS-Sensitive (Fig. 4D, Table S4C).

However, Table S4C has 24 proteins listed, not 8? There is no description of what is shown in Table S4C. What subset is this? Although I could eventually figure out what was in the tables, legends are a bit confusing. The “read me” are not particularly descriptive.

I don't understand what the Wilcoxon test in Figure 1H actually proves. The authors classify and then prove that their classes are different?

Supplemental Figure s2B is rather useless without common names.

Reviewer #4 (Remarks to the Author):

Overall

This first of its kind study leverages the high-throughput mutagenesis capability of the PiggyBac transposon combined with quantitative barcode-sequencing to screen a large pool of mutant *P. falciparum* parasites for differential tolerance to heat shock stress. The authors should be commended for this first large-scale forward-genetic screen. This is a grandiose achievement and is an exciting use of a library of mutants that the team has been developing for some years. Of course, the drawback of such a screen is that it can only highlight genes that are not essential to the blood-stage. For the assayed heatshock phenotype this is tolerated by the parasites, but there are likely essential genes that are missed since they are by definition not in the library.

Surprisingly, they found that upregulation of transcripts encoding enzymes needed for the isoprenoid biosynthesis pathway is significantly correlated with parasite survival of temperature and chemical stress. Other findings include downregulation of invasion and protein translation and trafficking pathways in response to stressors, with a concurrent upregulation in transcripts encoding damaged protein clearance.

The link between a response to high temperature stress that likely results in changes in cellular protein homeostasis and artemisinin sensitivity is perhaps not surprising. This is also manifest in the field data that is used to compare ART parasite clearance and mRNA abundance associated with the heatshock genes identified in this study. However it is not clear what the direct link between these responses might be. Even though the parasite may have overlapping response mechanisms, surely

they are not sensed in the same manner. Figure 5 (and the model therein) makes no indication as to what the sensor would be that integrates these disparate signals. Is it unfolded proteins? How would the response work? Mechanistic details are lacking and the overall conclusions are often unsupported by the work presented.

Comments:

The title is too bold and should be changed to: A possible apicoplast link to fever-survival and the artemisinin-resistance in the malaria parasite.

The Background section has a single sentence on ACTs and then a sentence on a screen for febrile temperatures. No clear link is provided.

Why was 8 hours chosen as the heat shock duration? Is this representative of normal malaria fever spikes? Are Plasmodium parasites generally viable after this prolonged treatment?

It is not very apparent how the 1K pB-mutant library is generated. Does it contain the 128 from Figure 1? If so, what is the variance in the results? Also, if it contains the original 128 then Figure 1 should be a Supplementary Figure. In general the pool-based experiments are not replicated. Why not?

The manuscript makes bold statements that are unsupported by the data presented. On line 214 is an early example, where the authors state that genes that are upregulated in HS are "drivers of the HS-response". How can this be known? These could be either direct or indirect effects of HS. In fact, a recent pre-print describing a transcriptional regulator of heat shock (<https://www.biorxiv.org/content/10.1101/2021.03.15.435375v1>) states that the direct response of heatshock is very focused through an small set of response genes downstream of a highly specific transcription factor. This is in direct contrast to the many changes reported here and in previous works by Oakley et al. cited in this work. Also, the recent bioRxiv submission describes a single driver of an early heat shock response. Do you see any consistent mutations or differential expression of this gene that may account for heat shock sensitivity/tolerance in your study?

Another vague statement comes in line 350: "We noted similarities between processes we identified to be driving the parasite HS-response and those implicated in parasite-resistance to artemisinin." Although three papers are cited, neither the processes nor how they are implicated in resistance to artemisinin are described or listed.

The justification for using DHC and LRR5 as exemplars of heat shock response is not clear. These are unlikely to be regulatory proteins, but rather downstream response proteins at best. Why were these selected then? What is the model for how mutations in LRR5 and DHC, two unrelated gene products that are unlikely to play a role in transcript regulation, can cause dysregulation of an overlapping gene set? Do you believe there is a feedback loop at play? Are any transcription associated genes consistently dysregulated in your sensitive vs. resistant pB lines? Furthermore, on line 235 the authors state that these are "...presumed loss-of-function mutants...". This is not very reassuring, and there are many ways to ensure a clean knockout – they could generate parasite lines with knockouts of the full genes in a clean genetic background or demonstrate a lack of protein product. Otherwise, these lines could have additional genetic abnormalities due to the selection process incurred in generating the pB libraries.

It is unclear to me whether the 3 parasite lines chosen for in-depth RNA seq analysis with and without heat stress were whole genome sequenced. If not, this critical control must be done to establish that the pB induced mutation is the primary driver of tolerance vs. sensitivity, even if the result was reproducible across two seemingly unrelated mutations.

Line 265 states that organellar targeting to the mitochondria and apicoplast are enriched. By how much? Relative to what? How comprehensive is the list of known nuclear genes whose protein product

is targeted to the mitochondrion or apicoplast?

Lines 318 the authors write that they found "all 8 isoprenoid-related pB-mutants ... to be HS-Sensitive". What other genes are relevant to isoprenoid biosynthesis in Plasmodium? And how many of these are present in other pB-mutants available to this group? I would suggest re-making a new pool that contains all isoprenoid-related genes and re-testing for HS-sensitivity. This could help to emphasize the currently weakly supported statement on line 334 that "These data taken together strongly implicated isoprenoid biosynthesis in the HS-response."

Engineered ART resistant parasite lines have been generated in the laboratory of David Fidock (<https://pubmed.ncbi.nlm.nih.gov/25502314/> and others). The authors could directly test whether kelch13 plays a key role in this sensitization to temperature or not?

A recent study (<https://pubmed.ncbi.nlm.nih.gov/33084568/>) has demonstrated that trophozoite stage parasites exposed to artemisinin produce significantly higher numbers of gametocytes. If this is true, are the changes being measured in this current study perhaps also attributable to a switch in a developmental program? If so, how is this separable from a HS-like response? Does HS result in increased gametocyte formation?

The authors should demonstrate that parasites lacking an apicoplast (see: <https://pubmed.ncbi.nlm.nih.gov/21912516/>) are more sensitive to temperature fluctuations. This would provide a definitive link between the proposed model put forth in this work. Alternatively, do you believe you could 'rescue' heat shock sensitivity of certain isoprenoid pathway mutants by spiking isoprenoid precursors or isoprenoids into the culture medium? This could potentially provide a proof of concept that links mutations in the isoprenoid biosynthesis pathway to the expected consequence of decreased protein prenylation. While intriguing, RNA seq data alone is not sufficient to conclusively demonstrate this link.

The reasoning behind the heat shock times chosen and the parasite staging compositions during all RNA seq experiments should be described slightly more in the main text. For example, understanding the staging and survival rate of parasites in the sample populations for both QI-seq and RNA seq experiments would help to deconvolute whether the observed decrease in transcripts encoding invasion genes is due to cell death, or a regulated stress response.

Along this same line of thought, did the parasite flasks that you harvested RNA for RNA seq from become asynchronous, or differently staged, during the 3 cycles that you grew them prior to harvest? This could help deconvolute whether certain transcripts are advantageous for heat shock survival, or are simply a result of changing parasite staging? On this note: Do the mutants with increased vs. decreased HS sensitivity have shorter vs. longer ring stages, since rings are the only HS resistant stage?

The cutoffs fold RNA abundance changes used are somewhat permissive. If you consider a more stringent cutoff (for example, FC >2 for increase or decrease in abundance) do you recover a very different list of differentially abundant genes?

It seems like most, or all, of the apicoplast related genes identified as essential for heat shock tolerance are also important for growth under normal conditions. Is this true, or are there exceptions? How do you interpret this trend given that sick parasites are likely more susceptible to all stressors?

The authors link plant thermotolerance to what is observed in their heat shock response of Plasmodium. Is heatshock in plants thought to be mediated by the plastid? Otherwise, this comparison doesn't make much sense. It is obvious that protein damage needs to be minimized, but I fail to see the connection. Similarly, the link to extremophilic algae is weak since it is well-established that there is broad-scale protein evolution that has led to hundreds to thousands of proteins being more

thermotolerant and active at higher temperatures. This is not a heatshock response, but rather an ability to proliferate and reproduce at higher temperatures.

Additional Comments:

Why is there a change in the concordance between genes shown to be HS-Sensitive (red dots) and both HS- and Growth-Sensitive (yellow dots) between Figure 1E and 1G?

Line 72: Does the claim that 90% of genes are untouched include the saturating mutagenesis results?

88: please cite this statement

91: please cite the statement regarding fevers driving parasite synchrony. Multiple publications have recently implicated an intrinsic circadian feedback loop as a likely driver of synchrony in natural infections, which should be mentioned as well: An intrinsic oscillator drives the blood stage cycle of the malaria parasite *Plasmodium falciparum*; The malaria parasite has an intrinsic clock.

91: what duration of heat shock is lethal to the non-ring stages, and what temperature?

148: What does extensively characterized mean? Were they clonal prior to combining them into libraries?

153: How long and at what temperature was the heat shock? How did you choose this?

163: How much of a growth advantage in terms of multiplication rate do HS tolerant mutants have? Is it all or nothing, or a slight difference in survival post heat stress? It is unclear what level of parasitemia an 8 fold increase in QI seq reads corresponds to.

185-195: What does deficient mean in this context? Do these genes have frameshifts in the CDS? Do the insertions demonstrably disrupt the functional domains of the proteins of interest identified?

225: Please describe the time point or points that were assayed for RNA transcript abundance in response to heat shock.

225: An alternative hypothesis for decrease in abundance of late stage expressed genes is the death of later stage parasites in response to heat shock, or failure of heat shocked parasites to progress.

225: Since nascent mRNA transcription is not the measured output, please change the descriptor from 'downregulated' to 'decreased in abundance'.

319: Do the insertions in these 8 IP biosynthesis genes cause likely loss of function? Specifically, do they completely disrupt translation upstream of predicted protein domains?

369: This line should be rewritten as it is currently incorrect: "...we found that genes mRNA levels of HS-Sensitive genes..."

Line 392: "endocytotic" should be "endocytic"

401: Are the DV resident proteins mentioned also important for asexual parasite growth in non-HS conditions?

421-445: This is a good model, however I feel it should be moved to the discussion section.

460: In terms of the glucose availability within *P. falciparum* culture media vs. in a natural infection, does it make sense that the parasite needs to make more glucose? Should glucose ever become a

limiting reagent for parasite growth at the parasitemia and timescale on which you carried out your experiments?

499: Is the DXS homolog Pf3D7_1337200 essential for normal growth?

Figures:

In some figures (1, for example) highlighted data points indicate pB insertion numbers. It would be more informative to indicate the gene ID disrupted.

Figures 1C and 1D are redundant with 1E. They could be moved to the supplementary materials if necessary to include.

Figure 2A: It looks like the majority of pB insertions recovered that confer heat shock tolerance also cause a higher growth rate under basal conditions. Can you highlight on this graph where LRR5 and DHC are for the purpose of showing us that these lines do not inherently grow faster than WT NF54, as was stated in the text?

Figure 3: Adding a note or diagram describing the RNA collection process including parasite staging and passage number would make it easier to interpret.

Figure 4: Panel D would be more informative if the normal condition growth differences were superimposed over the HS tolerance data as in figure 1.

Figure 5: the figure legend in panel A does not explain what the purple Apicoplast data represents.

REVIEWER COMMENTS

Reviewer #1 (Remarks to the Author):

The authors have addressed all of the points I raised appropriately.

We appreciate the reviewer considered our responses satisfactory.

One error remains in the modified figure 4E. The PPT and TPT proteins PF3D7_1218400 and PF3D7_0530200 are drawn as two different apicoplast transporters that import different metabolites, where in fact they are separate transporters in the outer and inner membranes of the apicoplast, which must import the same metabolites (see <https://pubmed.ncbi.nlm.nih.gov/16760253/>). Both PEP and DHAP are presumably imported into the apicoplast, but through concerted action of the two transporters, rather than separately.

Figure 4E has been modified to reflect the concerted action of PPT and TPT in PEP and DHAP import.

Reviewer #3 (Remarks to the Author):

In their revised manuscript the authors present a genomic study in which they perform a small scale screen of *P. falciparum* piggybac mutants to identify those that cannot tolerate “heat shock”. The authors then perform an RNAseq expression analysis of two Plasmodium piggybac mutants that show enhanced sensitivity to heat shock. The authors have thoughtfully responded to comments and the authors’ logic is easier to follow, although this remains rather difficult to review because of the cross-checking that is often

needed. The data seems to be robust even if this reviewer might not entirely agree with interpretation.

Response:

We appreciate the reviewer's comments and have revised the manuscript further to provide alternate interpretations and added some of the important caveats that our conclusions rely upon.

The manuscript might be more useful to the reader, especially the reader not so familiar with genomics, if the authors present the limitations of their study or discuss alternative interpretations. For example, the authors conclusions about the role of the apicoplast in heat stress are largely built on enrichment analysis, which is a reasonable approach. However, the authors' gene enrichment calculations may not have been corrected for multiple hypothesis testing. Finding 2 of an expected 10 in a set size of X (not sure what X is) may give a p-value of 0.008, but could be found by chance when the entire genome is examined because of multiple hypothesis testing. So, it is not entirely clear if some of the patterns that the authors discuss really are statistically significant. I would recommend that they make this clearer to the reader and provide both corrected and non-corrected p-values in the discussion of the results (if not in figures).

Response:

The composition and total count of the genes comprising the background-set and phenotype of interest-set for each enrichment analysis are provided in Supplementary Table S3C – D. The "Annotated" column reflects the total number of genes annotated to a given GO-term included in the analysis for all HS response-categories, while the "Significant" column reflects the number of genes annotated to a given GO-term in the HS response-category being tested for enrichment. We have added the full table-legends to the "ReadMe" tab of Supplementary Table 3 to aid comprehension.

We used a weighted Fisher/elimination hybrid-method to assess statistical significance of enrichment. As this method takes the entire structure of the GO-term hierarchy and the relationship between terms into consideration as opposed to simply evaluating the frequency of a single GO-term in a heat shock phenotype-category vs. the frequency of that GO term in the background-set, tests violate the expectation of independence, and correction for multiple testing is inappropriate (see Alexa et al. 2006, ref #59 cited in the manuscript).

The authors might also consider the alternative explanation for their RNA seq data—that there are minor differences in progression through the cell cycle in the mutants and thus after 3 cycles of heat shock for 8 hours each, the cycles are no longer in sync between the mutants and wildtype and the authors just identify cell-cycle regulated genes. Ribosomal protein genes are under strong cell cycle control and they do show strong differences after one parasite culture has been exposed to heat cycling. There are many ribosomal protein genes so p-values are impressive. Proteasome genes are under cell cycle control. Genes with a role in invasion also change in the authors' study (entry into host, enrichment by chance $p = 1 \times 10^{-23}$). Apicoplast genes are also under cell cycle control but there are fewer, so p-values are less impressive. Such phasing problems also complicate gene expression profiling of synchronized artemisinin resistant mutants as well. It is very difficult to compare gene expression patterns of different mutant lines because of the cell cycle differences that inevitably exist between two different mutant lines. Just changing temperature could hasten the cell cycle if you do it for 24 total hours. This possibility might be discussed.

Response:

Yes, we completely agree that this is an important concern. However, as we have shown in a previous study for the K13 mutant (Gibbons et al., Altered expression of K13 disrupts DNA replication and repair in *Plasmodium falciparum*. *BMC Genomics* 19, 849 (2018); reference #24 in the manuscript), which is one of the mutants in the pilot library, aphasical gene expression can be differentiated from simple changes in cycle progression and cycle length. Nonetheless, in anticipation of the potential impact of small cycle differences in these different clones, the parasites were highly synchronized (3 times) at the start of the experiment and during the experiment correct development stages were confirmed by direct observation on Giemsa-stained thin blood smears, not just based on timepoint. In addition, the gene hits identified in the forward screen as linked to the parasite's heat shock response matched the GO pathways and many of the same genes identified as differentially expressed by RNAseq analysis of the LRR5 and DHC mutant clones.

Minor points:

The manuscript would be more impressive and stronger if the authors added more p-values to the text. What is the probability of finding 10 of 24 mutants mapping to k13 interacting proteins are heat sensitive by chance? This could be just mentioned in the text (no additional figure needed).

Response: These analyses have been revised to indicate p-values where appropriate, as well as an additional supplemental figure S3D to bolster statistical support.

The authors could explain that what they call heat shock is not what is classically considered a heat shock. Classical heat shock causes gene expression changes within minutes of changing the temperature in many species—here the authors are studying the effects 2 days afterwards.

Response:

A sentence is added to the Introduction indicating the methodology mimics the previous gene expression study of Oakley and the *in vitro* experimental conditions represent the likely extreme limits of the malarial fever. Our previous study on individual mutant clones (Thomas et al. et al. Phenotypic Screens Identify Parasite Genetic Factors Associated with Malarial Fever Response in *Plasmodium falciparum* piggyBac Mutants. *mSphere* 1, (2016), reference #10 cited in the manuscript) analyzed by traditional Giemsa-stained thin blood smears confirmed the validity of the approach. While our experimental conditions do not represent “classical” heat shock, they are reflective of the prolonged febrile episodes experienced by individuals with malaria.

The authors should also mention that some of the piggybac insertion events were not in the actual gene that is named in the figure (e.g. in S3), but rather in the intergenic region near the gene.

Response:

Indeed, in most *piggyBac* mutants the transposon has inserted within untranslated flanking regions and not directly within the CDS. Gene assignments are based on the nearest CDS or within a known transcriptional unit. In highly essential genes such as K13 the transposon has inserted in or around the promoter region. Again, using the K13 example, even though the disruption is non-CDS the mutation

dysregulates gene expression sufficiently to link genotype to phenotype.

Some points are still rather difficult to follow.

We examined our 1K mutant-library for representation of isoprenoid biosynthesis, its immediate upstream-regulators (proteins responsible for modulation and import of glycolytic intermediates that serve as pathway substrates), and immediate downstream-effector proteins, and found that all eight isoprenoid biosynthesis-related pB-mutants included in the pooled screen were indeed HS-Sensitive (Fig. 4D, Table S4C).

However, TableS4C has 24 proteins listed, not 8? There is no description of what is shown in Table S4C. What subset is this? Although I could eventually figure out what was in the tables, legends are a bit confusing. The “read me” are not particularly descriptive.

Response:

The Table S4 “ReadMe” tab and the legend in the main text has been clarified. Table S4C contains 1k-library HS-screen data for all mutants in artemisinin MOA-associated processes featured in Figures 4D-E and S3A-C. We added this table in the first revision in response to a reviewer-request to clarify the mutants included in the making of pathway-associated figures, and the source-publications from which genes were implicated in the given pathway. Yes, all isoprenoid biosynthesis-related mutants are included in this table (n = 11 mutants in 10 unique genes), as are all mutants included in the indicated figures assessing predicted k13-interacting partners, the proteome of the digestive vacuole, and targets of artemisinin alkylation. There is some overlap in these categories. Table S4C has been extended to also include every 1K-library mutant in genes experimentally implicated in the highlighted ART MOA-associated processes (n = 58 mutants representing 47 unique genes).

I don’t understand what the Wilcoxon test in Figure 1H actually proves. The authors classify and then prove that their classes are different?

Response:

We agree that Figure 1H is redundant, and it has been removed.

Supplemental Figure s2B is rather useless without common names.

Response:

All pilot-library gene IDs, fold-change data and heat shock phenotype assignments are reported in Table S1. Common names are provided in the “Gene Function” column where available. We’ve included an additional Table S5A providing final phenotype-assignments for each pilot-library screen, including the HS screen, to ease interpretation of Figure S2B. The legend for Figure S2B has been updated to reference Table S5.

Reviewer #4 (Remarks to the Author):

Overall

This first of its kind study leverages the high-throughput mutagenesis capability of the PiggyBac transposon combined with quantitative barcode-sequencing to screen a large pool of mutant *P. falciparum* parasites for differential tolerance to heat shock stress. The authors should be commended for this first large-scale forward-genetic screen. This is a grandiose achievement and is an exciting use of a library of mutants that the team has been developing for some years. Of course, the drawback of such a screen is that it can only highlight genes that are not essential to the blood-stage. For the assayed heatshock phenotype this is tolerated by the parasites, but there are likely essential genes that are missed since they are by definition not in the library.

Response:

We appreciate the reviewer's recognition of the significance of our work. The reviewer is correct that we define "essential" genes (for asexual blood-stage growth at ideal culture conditions) as genes in which we recovered no insertions in coding regions from our saturation library, and over 60% of genes are essential. However—it is critical to note that this designation does **not** imply that the saturation library of >38,000 mutants is entirely composed of coding mutants in the other 40% of genes. ***Most mutants of the saturation-library have insertions in noncoding regions—including thousands of mutants with disruptions in predicted UTRs of essential genes, or intergenic regions neighboring essential genes.*** See Zhang et al. 2018 (ref # 11 cited in manuscript) Figure 1A-F.

While CDS disruptions are considered 'cleaner' for functional interpretation, insertions within noncoding regions can still cause sufficient functional dysregulation to make the genotype-phenotype links, and the potential impacts of the transposon-insertions allow characterization of otherwise-refractory essential genes. For example, we cite our previous profiling studies of a K13 mutant, in which the transposon integrated into the promoter region, demonstrating that even a small perturbation of the wild-type expression pattern of a gene is sufficient to link a gene to a phenotype (Gibbons et al., *ibid*, reference #24; and reference #23, Pradhan A, et al. Chemogenomic profiling of *Plasmodium falciparum* as a tool to aid antimalarial drug discovery. *Sci Rep* 5, 15930 (2015)).

Surprisingly, they found that upregulation of transcripts encoding enzymes needed for the isoprenoid biosynthesis pathway is significantly correlated with parasite survival of temperature and chemical stress. Other findings include downregulation of invasion and protein translation and trafficking pathways in response to stressors, with a concurrent upregulation in transcripts encoding damaged protein clearance.

The link between a response to high temperature stress that likely results in changes in cellular protein homeostasis and artemisinin sensitivity is perhaps not surprising. This is also manifest in the field data that is used to compare ART parasite clearance and mRNA abundance associated with the heatshock genes identified in this study. However it is not clear what the direct link between these responses might be. Even though the parasite may have overlapping response mechanisms, surely they are not sensed in the same manner. Figure 5 (and the model therein) makes no indication as to what the sensor would be that integrates these disparate signals. Is it unfolded proteins? How would the response work? Mechanistic details are lacking and the overall conclusions are often unsupported by the work presented.

Response:

We thank the reviewer for noting that we have demonstrated a link between response to high temperature-stress and artemisinin sensitivity—several overlapping pathways comprise both

responses. We do not purport to have resolved a detailed molecular mechanism for activation of either response, nor was discerning the detailed molecular mechanisms a goal for our genome-scale approach. Our “big picture” approach has, however, enabled us to propose a systems-level model for the connection between component pathways of the response suggested by our data, synthesizing prevailing mechanistic understanding of artemisinin resistance and response to febrile temperature—and laying groundwork for further experimental exploration.

Comments:

The title is too bold and should be changed to: A possible apicoplast link to fever-survival and the artemisinin-resistance in the malaria parasite.

Response:

The existing title conveys the significance of the study and its novel discovery to the potential readers.

The Background section has a single sentence on ACTs and then a sentence on a screen for febrile temperatures. No clear link is provided.

Response:

The results of this study is limited to the analysis of the link between the heat shock and artemisinin responses and was not extended to ART combination therapies. Indeed, this discovery on the link between the HS and ART is based primarily upon the parallel phenotyping results with only one drug/inhibitor at a time (Figure 5). As indicated in the text (line 360), we realized the similarities between our results for the parasite’s heat shock response and various studies on the genetic basis for ART-R phenotypes. Consequently, as this represents a novel finding there are no relevant prior publications to cite in the Background and exploring drug combinations would be complicated and better left to future studies.

Why was 8 hours chosen as the heat shock duration? Is this representative of normal malaria fever spikes? Are Plasmodium parasites generally viable after this prolonged treatment?

Response:

Malarial fever associated with *P. falciparum* infection commonly persists for several hours (6 to 12, according to current CDC and NHS clinical guidelines), recurring every 48 hours. As noted above, our methodology mimics the previous in vitro gene-expression study of malarial fever (Oakley et al. 2007; ref #9 cited in the manuscript), with duration and temperature of heat shock representing what are moderate to extreme limits of malarial fever. Our previous febrile-temperature study on individual mutant clones analyzed by traditional Giemsa-stained thin blood smears confirmed the validity of the approach (Thomas et al. *ibid*; reference #10 cited in the manuscript). Parasite survival and growth-rates in response to prolonged exposure to elevated temperature (40 – 41C) are highly dependent upon developmental stage at exposure, and the only surviving parasites after exposure are ring-stage.

We have included additional citations on the importance of temperature relative to stage sensitivity for parasite growth (Long et al., *Plasmodium falciparum*: in vitro growth inhibition by febrile temperatures. *Parasitol Res* 87, 553-555 (2001); Haynes & Moch, in *Malaria Methods and Protocols*, D.

L. Doolan, Ed. (Humana Press, Inc., Totowa, NJ, 2002); Blair et al., Transcripts of developmentally regulated *Plasmodium falciparum* genes quantified by real-time RT-PCR. *Nucleic Acids Research* 30, 2224-2231 (2002).

Previously, my research group used the temperature-cycling method in analyzing control of late-stage gene expression by Taqman (Blair et al., *ibid*), which is a methodology developed by David Haynes, our co-author. It is important to note that one of the most highly cited publications on *P. falciparum* developmentally controlled gene expression subsequently used the same temperature-cycling method to synchronize parasite growth (Le Roch et al., *Science* 2003, also with Haynes as a co-author). Concurrently, traditional sorbitol synchronization was used to validate the approach.

It is not very apparent how the 1K pB-mutant library is generated. Does it contain the 128 from Figure 1? If so, what is the variance in the results? Also, if it contains the original 128 then Figure 1 should be a Supplementary Figure. In general the pool-based experiments are not replicated. Why not?

Response:

The 1K library does not contain the 128 mutants of the pilot clone-library. The pilot clone-library is a distinct and critical tool we use in assay-design and preliminary analyses to ensure quality-control and scalability of all pooled phenotypic screens. Our pipeline consists of three major steps (illustrated conceptually in Figure S1):

- 1) protocols are tested using individual *pB*-mutant clones;
- 2) methods are adapted for pooled-screening using the well-characterized pilot-library (which includes those individual *pB*-mutant clones assayed individually in the first step), and
- 3) those confirmed methods are then scaled up to screening large pools of uncharacterized mutants (the 1k library).

We provide more context about creation of the 1k library, the pilot-clone library, and the methods for phenotype-validation the characteristics of each library allow to address the Reviewer questions about biological replication.

- Pilot clone-library: Large batches of up to 100 aliquots can be generated at a time, with each aliquot a nearly identical collection of 128 *pB* mutant clones, to provide enough biological replicate samples for experimental parallel phenotypic screens as used in this study. While this may seem trivial, the approach represents a major breakthrough in experimental genetic studies of *P. falciparum* – an organism well-known for its refractoriness to genetic manipulation. In the methods, we described how we grow 128 mutant clones individually and then pool together an equal amount of each. Once prepared we make as many cryopreserved aliquots as possible. This collection of mutant clones used for the pilot library are available to the community through MR4/BEI Resources.
- In contrast, the 1K-library is created from mixing 12 **un-cloned** mixed population pools (~100 unique mutants per pool). Each time the smaller pools are thawed and grown in culture to amplify, the relative abundance of each mutant changes due to the differences in inherent growth rates. These changes in pool-composition at each thaw of the component mixed-population pools

make biological replication of the 1k-library assays unfeasible. The 1k-library *does*, however, have internal redundancy (distinct mutants with identical insertion-sites), allowing internal validation.

- We identified 15 *piggyBac* mutants that appeared multiple times (i.e., at least twice in 2 different pools comprising the 1K-library). We used these mutant replicates to check reproducibility and validate performance of the 1K- library by comparing fold-change in response to heat shock (FC-HS) of these 15 *piggyBac* mutants in different pools. Identical mutants have highly correlated phenotypes regardless of pool (see Figure S7), validating the 1K-screen.

The manuscript makes bold statements that are unsupported by the data presented. On line 214 is an early example, where the authors state that genes that are upregulated in HS are “drivers of the HS-response”. How can this be known? These could be either direct or indirect effects of HS. In fact, a recent pre-print describing a transcriptional regulator of heat shock (<https://www.biorxiv.org/content/10.1101/2021.03.15.435375v1>) states that the direct response of heatshock is very focused through an small set of response genes downstream of a highly specific transcription factor. This is in direct contrast to the many changes reported here and in previous works by Oakley et al. cited in this work. Also, the recent bioRxiv submission describes a single driver of an early heat shock response. Do you see any consistent mutations or differential expression of this gene that may account for heat shock sensitivity/tolerance in your study?

Response:

Indeed, the recent BioRxiv pre-print has reported an ApiAP2 transcription factor that is linked to parasite heat shock response. The gene PF3D7_1342900 is named “ApiAP2-HS” in the BioRxiv manuscript. We carefully examined this gene in the context of our current study. On one hand, we found that in our RNAseq studies involving parental strain and heat shock mutant strains, this gene did not exhibit a canonical heat shock response that met our rigorous statistical classification criteria. On the other hand, in our *piggyBac* mutant 1K library, there is a mutant with an upstream intergenic insertion that may have dysregulated this gene and/or the neighboring gene PMT. This mutant was indeed HS-Sensitive, which supports a role in febrile response genetically at this locus. Further characterization both at the transcriptional level and via phenotypic screening assays will be planned to characterize our mutants at this locus.

Another vague statement comes in line 350: “We noted similarities between processes we identified to be driving the parasite HS-response and those implicated in parasite-resistance to artemisinin.” Although three papers are cited, neither the processes nor how they are implicated in resistance to artemisinin are described or listed.

Response:

This sentence in the Results section has been edited to clarify the noted similarities pertained to protein-damage and oxidative stress response processes, which establishes the rationale for performing the proteasome-inhibitor, oxidative stress, and additional artemisinin-derivative screens against the pilot library described in the next sentence. Added is a reference to our prior chemogenomics profiling study that demonstrated altered artemisinin sensitivity of some of the *piggyBac* mutants, including the K13 mutant. Interpretation of those results in the context of

artemisinin resistance and the cited publications is provided in the discussion.

The justification for using DHC and LRR5 as exemplars of heat shock response is not clear. These are unlikely to be regulatory proteins, but rather downstream response proteins at best. Why were these selected then? What is the model for how mutations in LRR5 and DHC, two unrelated gene products that are unlikely to play a role in transcript regulation, can cause dysregulation of an overlapping gene set? Do you believe there is a feedback loop at play? Are any transcription associated genes consistently dysregulated in your sensitive vs. resistant pB lines?

Response:

- Indeed, these genes are not transcriptional regulators, but the phenotypes indicate clearly that their functions are important in the parasite's response to heat stress. Since the parasite's metabolic functions involve an integrated network of interactions, disrupting the functions of LRR5 and DHC is expected to destabilize this network leading to a ripple effect of altered expression profiles within interaction network. Since LRR5 and DHC are functionally unrelated, dysregulated genes common to both mutants provide strong evidence for underlying components of this heat shock interaction network.
- We selected these two HS-Sensitive mutants over other candidates for phenotypic transcriptional profiling via RNAseq following strict criteria (as described in the Results of the manuscript):
 - i. Both mutants are highly sensitive to heat shock, but under ideal culture conditions grow better than most other mutants in the pilot library.
 - ii. The protein-coding sequences for both were disrupted by a *piggyBac* insertion (see additional discussion in our response to the next point).
 - iii. Gene functions are unrelated and not essential for parasite growth under ideal culture conditions.
 - iv. The HS-Sensitive phenotype was previously validated in a heat shock assay of individual clones.
 - v. The GO classifications of the leucine-rich repeat-5 (LRR5) and dynein heavy chain (DHC) are representative of the broad functional categories associated with heat response, regulating gene expression and intracellular vesicular transport, respectively.
- Both DHC and LRR5 are involved in dynamic metabolic processes active in the heat shock stress response. In addition, these processes identified as being essential for the heat shock response were otherwise dispensable. As noted above, the selection of these mutants was partly based on the GO classification being generally representative of the type of processes we identified as important for HS survival in our earlier small screen and other published reports, but at the time we did the experiments the selection of DHC could be considered a bit speculative. However, from recent reports there is now little doubt of the importance of vesicular transport (e.g., K13, coronin) in the parasite's response to ART. In addition to this dynein (PF3D7_1122900) from the pilot library, we modified Figure 2A to highlight three more dynein mutants that were included in the 1k library-screen (two different mutants of PF3D7_1202300 and one mutant of PF3D7_1023100). All three of these additional dynein-mutants were also heat shock-sensitive, supporting the relative importance of this functional process in the parasite's stress response.
- The purpose of the differential expression analysis by RNAseq was to reveal the global effects on the linked network of interacting partners caused by disrupting the functions of these genes. These mutants were chosen because their functions are unrelated and not essential for parasite

growth under ideal culture conditions, but both were experimentally validated to be essential for parasite survival of heat shock stress. Therefore, while some of the altered expression in each mutant is unique to their respective functions, the interconnections revealed by our comparative analysis is expected to identify the shared stress-response mechanisms.

... Furthermore, on line 235 the authors state that these are “...presumed loss-of-function mutants...”. This is not very reassuring, and there are many ways to ensure a clean knockout – they could generate parasite lines with knockouts of the full genes in a clean genetic background or demonstrate a lack of protein product. Otherwise, these lines could have additional genetic abnormalities due to the selection process incurred in generating the pB libraries. It is unclear to me whether the 3 parasite lines chosen for in-depth RNA seq analysis with and without heat stress were whole genome sequenced. If not, this critical control must be done to establish that the pB induced mutation is the primary driver of tolerance vs. sensitivity, even if the result was reproducible across two seemingly unrelated mutations.

Response:

The language has been revised to indicate the mutants have “altered function”, in the absence of molecular characterization of protein-products resulting from the disrupted locus in each mutant (and see our response to the reviewer’s first comment for a related discussion on the utility and interpretation of coding vs. noncoding mutants). We previously reported whole-genome sequencing of the NF54-parent-strain and 29 of the 128 isogenic *pB* mutants of the pilot clone library, which confirmed the lack of significant genomic differences in long term culture and only a single *pB* insertion in each *pB* mutant line (Thomas et al., *ibid*). The highly reproducible and specific heat shock phenotypes we report for these mutants indicates clear functional consequences of those disruptions. Additionally, several independent insertions in the DHC gene-family within the 1k library are also HS-Sensitive (Figure 2C)—further supporting this sensitivity is specifically tied to DHC-family gene-function.

Line 265 states that organellar targeting to the mitochondria and apicoplast are enriched. By how much? Relative to what? How comprehensive is the list of known nuclear genes whose protein product is targeted to the mitochondrion or apicoplast?

Response:

Enrichment analysis here evaluates whether the number of genes annotated to a particular GO term within a given HS response-category of interest is significantly higher than the number we would expect based on the total number of genes annotated to that term detected in RNAseq. Methods for all enrichment-analyses are provided in the Methods section. The composition and total count of the genes comprising the background-set and phenotype of interest-set for each enrichment analysis are provided in Supplementary Table S3B – D. The “Annotated” column in S3C-D reflects the total number of genes annotated to a given GO-term included in the analysis for all HS response-categories, while the “Significant” column reflects the number of genes annotated to a given GO-term in the HS response-category being tested for enrichment.

As an example—of all 61 genes with the “mitochondria” cellular compartment GO term detected above threshold in all three mutants in RNAseq, 13 of those are increased in response to heat shock. Only ~5 “mitochondria” genes would have been expected to be increased in response to heat shock by random chance. The difference between expectation and reality is significant (weighted Fisher/elim hybrid-method $p = .0051$), meaning this term is enriched. The same logic can be followed for all the other terms.

Lines 318 the authors write that they found “all 8 isoprenoid-related pB-mutants ... to be HS-Sensitive”. What other genes are relevant to isoprenoid biosynthesis in Plasmodium? And how many of these are present in other pB-mutants available to this group? I would suggest re-making a new pool that contains all isoprenoid-related genes and re-testing for HS-sensitivity. This could help to emphasize the currently weakly supported statement on line 334 that :These data taken together strongly implicated isoprenoid biosynthesis in the HS-response.”

Response:

We define “isoprenoid biosynthesis-related” genes as in Figure 4E—starting with the proteins controlling availability of substrates to the pathway (HAD1, the direct upstream regulator of the pathway; the two transporters TPT and PPT, and the two enzymes PyKII and TIM), on to the enzymes of the isoprenoid biosynthesis pathway itself ($n = 7$, DXS through IspH), and any gene whose product is directly modified by isoprenoids (prenylated proteins, shown with light green or blue lightning bolts in Figure 4E. Pf has ~20 total)). Of that list of approximately ~32 genes in the genome, 10 were represented in our 1k library. Nine of these 10 genes (or 10 of 11 mutants) were sensitive to heat shock (Fisher-test $p = 0.01$), and the role of isoprenoid biosynthesis in the heat shock response is strongly supported. A detailed list of all mutants in these genes included in the 1K library, as well as references supporting their role in isoprenoid biosynthesis, is included in Table S4C.

Engineered ART resistant parasite lines have been generated in the laboratory of David Fidock (<https://pubmed.ncbi.nlm.nih.gov/25502314/> and others). The authors could directly test whether kelch13 plays a key role in this sensitization to temperature or not?

Response:

We appreciate the suggestion and indeed we will discuss this type of study with David. However, we do already include a well-characterized k13 mutant that has a HS phenotype and altered sensitivity to artemisinin. In addition, using one of his mutants will introduce additional unknown variables since those parasites will not be isogenic with the *piggyBac* mutants.

A recent study (<https://pubmed.ncbi.nlm.nih.gov/33084568/>) has demonstrated that trophozoite stage parasites exposed to artemisinin produce significantly higher numbers of gametocytes. If this is true, are the changes being measured in this current study perhaps also attributable to a switch in a developmental program? If so, how is this separable from a HS-like response? Does HS result in increased gametocyte formation?

Response:

This is a fairly open-ended comment and beyond the bounds of our study. Nonetheless, we find no evidence to suggest increased gametocyte-commitment is a confounder in our HS assays—which is not to say that increased gametocyte-commitment in response to heat shock could not itself be a “HS-like response”. Indeed, increased gametocyte formation is associated with exposure to several disparate stressors and is not unexpected to observe in heat stress. That said, our data do not suggest a direct relationship between HS-Sensitivity and sexual-stage commitment (nor does significantly increased gametocyte-commitment characterize the wildtype parasite response). Several genes associated with gametocyte commitment are represented in our 1k library (including AP2G, GDV1, MSRP1, PMT, and DBLMSP2). None of the mutants in these genes were characterized as HS-Sensitive. Similarly, our RNASeq data do not suggest increased mRNA levels of genes associated with increased gametocyte commitment.

The authors should demonstrate that parasites lacking an apicoplast (see: <https://pubmed.ncbi.nlm.nih.gov/21912516/>) are more sensitive to temperature fluctuations. This would provide a definitive link between the proposed model put forth in this work. Alternatively, do you believe you could ‘rescue’ heat shock sensitivity of certain isoprenoid pathway mutants by spiking isoprenoid precursors or isoprenoids into the culture medium? This could potentially provide a proof of concept that links mutations in the isoprenoid biosynthesis pathway to the expected consequence of decreased protein prenylation. While intriguing, RNA seq data alone is not sufficient to conclusively demonstrate this link.

Response:

Indeed, RNAseq data alone are not what we rely on to demonstrate the link between isoprenoid biosynthesis and heat shock. The strongest connections are from the pooled screens of the 1k library. We refer the reviewer to figures 4D-4E and S3A-S3C along with the associated tables.

Importantly, 10 of the 11 isoprenoid biosynthesis-related mutants represented in the 1k screen are sensitive to HS (Fisher test, $p = 0.01$).

The reasoning behind the heat shock times chosen and the parasite staging compositions during all RNA seq experiments should be described slightly more in the main text. For example, understanding the staging and survival rate of parasites in the sample populations for both QI-seq and RNA seq experiments would help to deconvolute whether the observed decrease in transcripts encoding invasion genes is due to cell death, or a regulated stress response.

Response:

More details supporting the rationale for the selected heat shock parameters have been added to the manuscript, and we have clarified that parasite cultures were highly synchronized prior to exposure to HS. As already noted, all cultures were staged via Giemsa-stained thin blood smears at every collected timepoint and we did not simply rely on the time passed.

Along this same line of thought, did the parasite flasks that you harvested RNA for RNA seq from become asynchronous, or differently staged, during the 3 cycles that you grew them prior to harvest? This could help deconvolute whether certain transcripts are advantageous for heat shock survival, or are simply a result of changing parasite staging? On this note: Do the mutants with increased vs. decreased HS sensitivity have shorter vs. longer ring stages, since rings are the only HS resistant stage?

Response:

As indicated above, the parasite cultures were triple synchronized at the beginning of the experiments to control for developmental 'wobble'. We would expect some degree of asynchrony in the mutants with increased sensitivity; however, as also noted above we developed methods to parse out the aphasical gene expression patterns (Gibbons et al., *ibid*). In addition, the earlier studies on temperature cycling methods demonstrated that even slightly elevated temperature (e.g., $\geq 38.5^\circ\text{C}$) are deleterious to late-stage parasites (schizonts) and would serve to help maintain a relatively synchronous population through the experiment.

Based on the in-depth analysis of the k13 mutant, increased sensitivity to artemisinin is due to a shortened ring stage and logically this would extend to increased HS sensitivity.

The cutoffs fold RNA abundance changes used are somewhat permissive. If you consider a more stringent cutoff (for example, $\text{FC} > 2$ for increase or decrease in abundance) do you recover a very different list of differentially abundant genes?

Response:

As shown in the histograms figure, the majority of increased (orange)-and decreased (navy)-abundance genes have a fold change higher or lower than 2 in response to heat shock across all three parasite lines. Therefore, setting $\text{FC} > 2$ ($\log_2(\text{FC}) > 1$) does not appreciably change the list of differentially expressed genes.

It seems like most, or all, of the apicoplast related genes identified as essential for heat shock tolerance are also important for growth under normal conditions. Is this true, or are there exceptions? How do you interpret this trend given that sick parasites are likely more susceptible to all stressors?

Response:

We agree that there is a need for nuanced interpretation of “complicated” phenotypes where a Growth-Sensitive mutant is also sensitive to a selection of interest. We developed the Phenotypic Fitness Score (PFS) to distinguish mutants with phenotypes specifically in the condition under selection (HS) vs. those with inherently compromised growth in ideal conditions—inherent growth-phenotypes are accounted for by normalizing mutant performance under selection to its performance in ideal growth-conditions.

Double-sensitive mutants were classified into the distinct “HS- and Growth-Sensitive” phenotype-category and are not included in our “HS-Sensitive” classification to avoid overinterpretation of possibly confounding phenotypes (Figures 1 and 2).

The authors link plant thermotolerance to what is observed in their heat shock response of Plasmodium. Is heatshock in plants thought to be mediated by the plastid? Otherwise, this comparison doesn't make much sense. It is obvious that protein damage needs to be minimized, but I fail to see the connection. Similarly, the link to extremophilic algae is weak since it is well-established that there is broad-scale protein evolution that has led to hundreds to thousands of proteins being more thermotolerant and active at higher temperatures. This is not a heatshock response, but rather an ability to proliferate and reproduce at higher temperatures.

Response:

This comment appears to be similar to a comment from Reviewer #2 in the initial critique and our previous response is pasted below with bullets. Our response is primarily about the apparent mechanism that the isoprenoid metabolic pathway is important for in the heat shock response, prenylation of HSP40.

- We agree that protein prenylation is a ubiquitous eukaryotic post-translational protein modification process. However, the enormous functional diversity of prenylated proteins gives rise to a wide range of roles in cell signaling, homeostasis and coping with stress. Our study showed the important aspect of isopentenyl diphosphate in the parasite for dealing with cyclical febrile stress encountered during the malaria paroxysm.
- We are not suggesting that parasites invented prenylation to survive heat stress, particularly not to the exclusion of other eukaryotes. As the reviewer has noted, however, we and others have demonstrated that prenylation is critical for parasite survival of febrile temperatures—and phyletic analyses have long suggested the pathway through which these parasites produce isoprenoid precursors necessary for that prenylation was acquired via the secondary endosymbiotic event giving rise to the apicoplast.
- While all eukaryotes do employ prenylation—plants/algae, bacteria, and apicomplexan parasites are the only ones that do so via the non-mevalonate isoprenoid biosynthesis pathway. Stress responses, including those induced in response to heat stress, have been thoroughly studied in these lineages. It is also notable that prenylation (which includes geranyl-geranylation as well as farnesylation) in *Plasmodium* during the blood stage is restricted to an uncharacteristically small list of proteins (~15 to 20), the vast majority of which are members of the Rab-family proteins

with key roles in vesicular trafficking. The lone farnesylated protein during the parasite blood-stage is HSP40—and incidentally, this farnesylation of HSP40 is absolutely critical for parasite survival of thermal shock (see Mathews et al., ref #45).

Additional Comments:

Why is there a change in the concordance between genes shown to be HS-Sensitive (red dots) and both HS- and Growth-Sensitive (yellow dots) between Figure 1E and 1G?

Response:

Figure 1 has been reduced to remove redundancy. Figures 1C, 1D and 1G were intended to highlight/clarify features of Figure 1E, and all panels should agree with Figure 1E.

Line 72: Does the claim that 90% of genes are untouched include the saturating mutagenesis results?

Response:

The passage has been edited to clarify that 90% of genes have not been explored experimentally through classical (e.g., targeted) genetic manipulation.

88: please cite this statement [Host fever is triggered by a Type I shock-like response of the innate immune-system exposure to extracellular parasite debris released when infected RBCs are lysed during parasite egress]

Response: We've added a citation for Kwiatkowski 1989 (ref #4).

91: please cite the statement regarding fevers driving parasite synchrony. Multiple publications have recently implicated an intrinsic circadian feedback loop as a likely driver of synchrony in natural infections, which should be mentioned as well: An intrinsic oscillator drives the blood stage cycle of the malaria parasite Plasmodium falciparum; The malaria parasite has an intrinsic clock.

Response:

We do not think that the circadian feedback-loop is directly relevant since our study is analyzing how elevated temperature alters normal development and the parasite's response. As noted above, additional citations have been added to the revision to support the role of heat/temperature cycling in driving parasite synchrony.

91: what duration of heat shock is lethal to the non-ring stages, and what temperature?

Response:

This varies with by stage of parasite development. Our study is deciphering the parasite's ability to survive febrile temperatures during ring stage. Previous systematic characterization of the relationship between temperature (tested at a range of 37C to 40C), length of exposure (from 6 to 12 hours), and parasite stage at time of exposure indicated that starting at 40C, there is an exponential increase in growth-inhibition as a function of parasite age independent of incubation time, apparent for parasites starting around late ring-stage (12 hrs post invasion). The authors in Long et alia (ibid) fitted a model to their observations predicting >90% reduction in parasitemia at 40C, rising to 98% reduction at 41C—with early ring-stage parasites the only survivors. Our observations of our cultures post 8-hour exposure to 41C are consistent with these data.

148: What does extensively characterized mean? Were they clonal prior to combining them into libraries?

Response:

Yes, the pilot-library is comprised of 128 clonal mutant lines pooled together. Extensive characterization includes:

- Repeated characterization and confirmation of each mutant's growth-rate both as a clone (Balu et al., PLoS ONE 5, e13282 (2010)) and in pools of varied composition, including the specific context of the pilot-library (Bronner et al., Genome Research, (2016)). Growth for all 128 clones was confirmed in subsequent growth screens across 12 asexual intra-erythrocytic development cycles (24 days, with samples collected in biological duplicate at cycles 3, 6, 9 and 12). Growth rates of individual *pB*-mutant clones were highly reproducible between biological replicates, as well as between pools with different compositions.
- Approximately half of the mutants comprising the pilot-library were previously evaluated for heat-shock phenotypes as clones. Phenotypes were again very reproducible in biological triplicate.
- Whole-genome sequencing of the NF54 parent-strain and 29 of the 128 *pB*-mutant-clones in the pilot-library confirmed the lack of significant genomic differences in long term culture and only a single *pB* insertion in each *pB* mutant line was identified, supporting that detected phenotypes are attributable to the single disruption (Thomas et al. ibid).

153: How long and at what temperature was the heat shock? How did you choose this?

Response:

See our response to the previous reviewer above. Briefly, pools of *pB*-mutant parasites were exposed to three rounds of temperature-cycling (one growth-cycle at 37C until parasites reached ring-stage;

then 8 hours at 41C, then returned to 37C for the remainder of the 48-hour developmental cycle until they reached ring-stage again) to simulate the cyclical pattern of malarial fever (Figure 1A). Our methodology mimics the previous in vitro gene-expression study of malarial fever (Oakley et al., *ibid*), with duration and temperature of heat shock representing what are likely moderate to extreme limits of malarial fever. Our previous febrile-temperature study on individual mutant clones analyzed by traditional Giemsa-stained thin blood smears confirmed the validity of the approach (Thomas et al. *ibid*).

163: How much of a growth advantage in terms of multiplication rate do HS tolerant mutants have? Is it all or nothing, or a slight difference in survival post heat stress? It is unclear what level of parasitemia an 8 fold increase in QI seq reads corresponds to.

Response:

Our initial study applying heat shock to individual clones (Thomas et al., *ibid*) analyzed parasite growth by flow cytometry and traditional Giemsa-stained blood smears. This was possible because each clone was analyzed independently for growth differences relative to the WT parent NF54. However, in the current study all the different mutant parasites were cultured together in pools and growth analysis was accomplished by QIseq. Consequently, there is no “parasitemia” in the traditional sense and growth of each mutant is relative to the other. However, 8-fold increase by QIseq can be considered to similar fold-change determined by qRT-PCR methods and other quantitative lab assays.

185-195: What does deficient mean in this context? Do these genes have frameshifts in the CDS? Do the insertions demonstrably disrupt the functional domains of the proteins of interest identified?

Response:

We are unsure of the question(s). However, mutations are not known to cause frame shifts. Typically, insertions in CDS severely diminish expression levels and either largely KO function or severely attenuate function.

225: Please describe the time point or points that were assayed for RNA transcript abundance in response to heat shock.

Response:

RNAseq experimental design is outlined in Fig. S11A. Briefly, highly synchronized ring-stage cultures of wildtype NF54 and HS-Sensitive mutants *LRR5* and *DHC* were split equally into four T75 flasks each. All parasites were grown at the normal human body temperature (37C) to early ring-stage. Three flasks of each pool were then exposed to febrile temperatures (41C) for 8 hours, while the remaining two flasks were allowed to continue to grow at 37C for 8 hours without exposure to heat-stress. This temperature-cycling was repeated three times, just as we allowed for the pooled HS-Screen. After the third round of heat-shock (Time 1, T¹), RNA was harvested simultaneously from both conditions for

RNAseq as in (Gibbons et al., 2018, *ibid*).

225: An alternative hypothesis for decrease in abundance of late stage expressed genes is the death of later stage parasites in response to heat shock, or failure of heat shocked parasites to progress.

Response:

This seems to us a restatement of the same hypothesis that we presented and perhaps our wording is confusing. We considered genes “increased in abundance” (i.e., “upregulated”) to be “drivers” of the response.

225: Since nascent mRNA transcription is not the measured output, please change the descriptor from ‘downregulated’ to ‘decreased in abundance’.

Response:

We’ve added a clarification in the Results that all references to “upregulated” and “downregulated” in the context of RNAseq are shorthand for “increased abundance” and “decreased abundance” transcripts, respectively.

319: Do the insertions in these 8 IP biosynthesis genes cause likely loss of function? Specifically, do they completely disrupt translation upstream of predicted protein domains?

Response:

All but one of the isoprenoid biosynthesis-related genes are noncoding insertions (see Table S4C).

369: This line should be rewritten as it is currently incorrect: “...we found that gnes mRNA levels of HS-Sensitive genes...”

Response: The typo has been corrected.

Line 392: “endocytotic” should be “endocytic”

Response: The line has been edited.

401: Are the DV resident proteins mentioned also important for asexual parasite growth in non-HS conditions?

Response:

All DV proteome-associated genes represented in the screen (10 unique genes, including M1APP and plasmepsin I) are essential (non-mutable in the coding region); therefore, all mutations in these genes are noncoding. The MIS and MFS scores for each gene—which relate relative essentiality (i.e., mutability) and relative abundance/growth (i.e., fitness) in the saturation screen of asexual blood-stage parasites under ideal in vitro culture conditions—are also included in Table S4C.

421-445: This is a good model, however I feel it should be moved to the discussion section.

Response:

We thank the reviewer for the suggestion. The figure is a summation of the data reported in our Results, and we believe its placement at the end of the Results section is appropriate.

460: In terms of the glucose availability within *P. falciparum* culture media vs. in a natural infection, does it make sense that the parasite needs to make more glucose? Should glucose ever become a limiting reagent for parasite growth at the parasitemia and timescale on which you carried out your experiments?

Response:

Complete RPMI 1640 with serum or albumax, which is the standard culture medium used for *P. falciparum*, is a 'rich' culture medium and glucose is not considered a limiting factor.

499: Is the DXS homolog Pf3D7_1337200 essential for normal growth?

Response:

Yes, Pf3D7_1337200 is highly essential for ideal blood-stage growth with a Mutagenesis Index Score (MIS, the measure of essentiality on a scale from 0 to 1 where 0 indicates highest essentiality, reported in our original saturation mutagenesis paper) of 0.123. The DXS mutant in our 1K library has the mutation in the noncoding region; however, the mutant is classified as HS-Sensitive—meaning it is sensitive to heat but its growth phenotype in ideal culture conditions is not severely attenuated. See Table S4C.

Figures:

In some figures (1, for example) highlighted data points indicate pB insertion numbers. It would be more informative to indicate the gene ID disrupted.

Response:

We've updated the figure-legend to direct readers to the precise insertion-sites for each indicated pB-mutant.

Figures 1C and 1D are redundant with 1E. They could be moved to the supplementary materials if necessary to include.

Response:

Figure 1 has been reduced to remove redundancy.

Figure 2A: It looks like the majority of pB insertions recovered that confer heat shock tolerance also cause a higher growth rate under basal conditions. Can you highlight on this graph where LRR5 and DHC are for the purpose of showing us that these lines do not inherently grow faster than WT NF54, as was stated in the text?

Response:

Qlseq is a type of nested PCR initially amplifying on *piggyBac cis* elements and therefore only detects mutants. In the current study, the parent WT NF54 line is never included in pooled screening. In an earlier study analyzing mutant clones, almost no mutants grew significantly faster than the WT parent clone (Balu et al., 2010, *ibid*). The bar charts in figure 1E and 2A show relative fold-change of the mutant in response to heat shock normalized against performance of that same mutant under ideal culture conditions – not against the NF54 parent. The LRR5 and DHC clonal lines were screened in the pilot library (and are included in figure 1E). Figure 2 shows the results from the larger 1k library (which does not include the clonal lines).

Figure 3: Adding a note or diagram describing the RNA collection process including parasite staging and passage number would make it easier to interpret.

Response:

A detailed methods-figure describing RNAseq experimental design is provided in Figure S11. We have also added further clarifications to the relevant subsection of the Methods.

Figure 4: Panel D would be more informative if the normal condition growth differences were superimposed over the HS tolerance data as in figure 1.

Response:

These figures are relatively cluttered already and we think including additional information would make it unreadable. For particular genes of interest readers will be able find this information in the related table.

Figure 5: the figure legend in panel A does not explain what the purple Apicoplast data represents.

Response: The legend has been clarified.

REVIEWERS' COMMENTS

Reviewer #3 (Remarks to the Author):

Overall the reviewers have addressed my concerns, but I do think the authors are doing the community a disservice by not addressing the potential "cherry-picking" of the isoprenoid data with a few lines in the discussion. The fact that others have done the same thing doesn't make it right. The authors state, rather emphatically, that there are no cell cycle difference (which could give rise to differences in the gene set enrichment changes) between the mutant and WT in their RNA-seq data because they examined their mutants by Giemsa stain (not particularly quantitative). But haven't the authors already stated that their mutants grow differently in heat shock? In fact, this is how they were isolated in the Piggybac screen.

Instead of arguing, it would be simple for them to write at line 479, something like the following "A caveat of our gene expression study is that we performed a lengthy RNAseq heatshock timecourse experiment with highly synchronized mutant and wildtype parasites. Cell cycle differences could theoretically and artifactually contribute to some of our observed isoprenoid biosynthesis gene expression changes. We consider this unlikely because..."

Reviewer #4 (Remarks to the Author):

The authors have done a comprehensive job of incorporating many of the suggestions into a highly revised manuscript that reads well and embodies a significant contribution to the field.

At this point, I would merely request that the following points be added to the text:

1) This statement should still be toned down:

On line 219 the authors state that genes that are upregulated in HS are "drivers of the HSresponse". As I asked before, How can this be known?

2) The fact that the Pb insertions are in noncoding regions in "all but one" isoprenoid biosynthesis-related genes needs to be added explicitly to the text:

Original Question: Do the insertions in these 8 IP biosynthesis genes cause likely loss of function?

Specifically, do they completely disrupt translation upstream of predicted protein domains?

Response: All but one of the isoprenoid biosynthesis-related genes are noncoding insertions (see Table S4C).

3) For the final results section, which is largely described in figure 5, I do not fully agree with the interpretation of wildtype parasite regulation in response to heat shock, as in line 406: "We found that K13-defined endocytosis is also downregulated in response to HS."

Interpreting the large differences in parasite transcript abundances in Figure 5e as specific gene regulation is overstating the result. This result could just as easily be a combined output resulting from a specific transcriptional response coupled with a number of indirect effects of heat shock. Further complicating the interpretation of this result is the culturing method for this experiment (Under Methods as: Comparative RNAseq between wild-type NF54 and two HS-Sensitive mutant parasite). The culture method was to heat shock 3 independent times before harvesting RNA, which is effectively a selection. At that point it is completely fair to say that parasites that survive heat shock are likely to overexpress protein clearance-related genes, but not that they specifically upregulate them as a heat shock response.

REVIEWERS' COMMENTS

Reviewer #3 (Remarks to the Author):

Overall the reviewers have addressed my concerns, but I do think the authors are doing the community a disservice by not addressing the potential "cherry-picking" of the isoprenoid data with a few lines in the discussion. The fact that others have done the same thing doesn't make it right. The authors state, rather emphatically, that there are no cell cycle difference (which could give rise to differences in the gene set enrichment changes) between the mutant and WT in their RNA-seq data because they examined their mutants by Giemsa stain (not particularly quantitative). But haven't the authors already stated that their mutants grow differently in heat shock? In fact, this is how they were isolated in the Piggybac screen.

Instead of arguing, it would be simple for them to write at line 479, something like the following "A caveat of our gene expression study is that we performed a lengthy RNAseq heatshock timecourse experiment with highly synchronized mutant and wildtype parasites. Cell cycle differences could theoretically and artifactually contribute to some of our observed isoprenoid biosynthesis gene expression changes. We consider this unlikely because..."

The reviewer is correct that the HS-Sensitive mutants by definition grow differently in response to heat shock—and critically, these mutants *do not* grow differently in response to ideal growth conditions. We have added a paragraph summarizing the controls included/steps taken in each set of experiments (RNAseq and pooled screening) to minimize the potential effects of differences in cell-cycle length that led us to conclude the role of isoprenoid biosynthesis in the HS response is not an artefact of general growth defects (lines 486-498 of the revised manuscript).

Reviewer #4 (Remarks to the Author):

The authors have done a comprehensive job of incorporating many of the suggestions into a highly revised manuscript that reads well and embodies a significant contribution to the field.

At this point, I would merely request that the following points be added to the text:

1) This statement should still be toned down:

On line 219 the authors state that genes that are upregulated in HS are “drivers of the HSresponse”. As I asked before, How can this be known?

We’ve removed the offending sentence in the revised manuscript.

2) The fact that the Pb insertions are in noncoding regions in “all but one” isoprenoid biosynthesis-related genes needs to be added explicitly to the text:

Original Question: Do the insertions in these 8 IP biosynthesis genes cause likely loss of function?

Specifically, do they completely disrupt translation upstream of predicted protein domains?

Response: All but one of the isoprenoid biosynthesis-related genes are noncoding insertions (see Table S4C).

The discussion has been updated in the revised manuscript (lines 486 – 498).

3) For the final results section, which is largely described in figure 5, I do not fully agree with the interpretation of wildtype parasite regulation in response to heat shock, as in line 406: “We found that K13-defined endocytosis is also downregulated in response to HS.”

Interpreting the large differences in parasite transcript abundances in Figure 5e as specific gene regulation is overstating the result. This result could just as easily be a combined output resulting from a specific transcriptional response coupled with a number of indirect effects of heat shock. Further complicating the interpretation of this result is the culturing method for this experiment (Under Methods as: Comparative RNAseq between wild-type NF54 and two HS-Sensitive mutant parasite). The culture method was to heat shock 3 independent times before harvesting RNA, which is effectively a selection. At that point it is completely fair to say that parasites that survive heat shock are likely to overexpress protein clearance-related genes, but not that they specifically upregulate them as a heat shock response.

We agree with the reviewer that these phenotypes are complicated—the complete catalogue of epistatic interactions that ultimately results in survival of any stress-condition cannot be detailed mechanistically (including differentiating direct from indirect effects) through RNAseq and genome-scale approaches alone. However—all observations have been reported in heat-shocked parasites vs. carefully-considered, non-heat-shocked controls. We have added additional discussion to address this point in the revised manuscript (lines 473 – 479).

To the reviewer’s second point, culture methods were designed to approximate conditions of cyclical malarial fever; three rounds of heat shock is indeed a selection, as is cyclical malarial fever. Our observation that protein clearance-related genes are reproducibly upregulated in response to heat shock— and that those genes are *not* upregulated in ideal culture-condition controls—is consistent with our interpretation that upregulation of those genes characterizes the heat shock response. We have clarified the clinical basis for our heat shock assay-parameters in the results section.